# Spatially explicit analysis identifies significant potential for bioenergy with carbon capture and storage in China

Xiaofan Xing[1], Rong Wang [1,2,3,4,5,6✉], Nico Bauer [7], Philippe Ciais [8,9], Junji Cao[10], Jianmin Chen [1,2,3], Xu Tang[1,2,3], Lin Wang [1], Xin Yang[1], Olivier Boucher[11], Daniel Goll [12], Josep Peñuelas [13,14], Ivan A. Janssens [15], Yves Balkanski [8], James Clark [16], Jianmin Ma [17], Bo Pan[18], Shicheng Zhang [1], Xingnan Ye[1], Yutao Wang [1], Qing Li [1], Gang Luo[1], Guofeng Shen [17], Wei Li [19], Yechen Yang[1] & Siqing Xu[1]

As China ramped-up coal power capacities rapidly while $CO_2$ emissions need to decline, these capacities would turn into stranded assets. To deal with this risk, a promising option is to retrofit these capacities to co-fire with biomass and eventually upgrade to CCS operation (BECCS), but the feasibility is debated with respect to negative impacts on broader sustainability issues. Here we present a data-rich spatially explicit approach to estimate the marginal cost curve for decarbonizing the power sector in China with BECCS. We identify a potential of 222 GW of power capacities in 2836 counties generated by co-firing 0.9 Gt of biomass from the same county, with half being agricultural residues. Our spatially explicit method helps to reduce uncertainty in the economic costs and emissions of BECCS, identify the best opportunities for bioenergy and show the limitations by logistical challenges to achieve carbon neutrality in the power sector with large-scale BECCS in China.

[1] Shanghai Key Laboratory of Atmospheric Particle Pollution and Prevention, Department of Environmental Science and Engineering, Fudan University, Shanghai, China. [2] IRDR International Center of Excellence on Risk Interconnectivity and Governance on Weather/Climate Extremes Impact and Public Health (WECEIPHE), Fudan University, Shanghai, China. [3] Institute of Atmospheric Sciences, Fudan University, Shanghai, China. [4] Center for Urban Eco-Planning and Design, Fudan University, Shanghai, China. [5] Big Data Institute for Carbon Emission and Environmental Pollution, Fudan University, Shanghai, China. [6] Shanghai Institute of Pollution Control and Ecological Security, Shanghai, China. [7] Potsdam Institute for Climate Impact Research (PIK), Member of the Leibniz Association, Potsdam, Germany. [8] Laboratoire des Sciences du Climat et de l'Environnement, CEA CNRS UVSQ, Gif-sur-Yvette, France. [9] Climate and Atmosphere Research Center (CARE-C) The Cyprus Institute 20 Konstantinou Kavafi Street, 2121 Nicosia, Cyprus. [10] Institute of Atmospheric Physics, Chinese Academy of Sciences, Beijing, China. [11] Institut Pierre-Simon Laplace, Sorbonne Université/CNRS, Paris, France. [12] Lehrstuhl für Physische Geographie mit Schwerpunkt Klimaforschung, Universität Augsburg, Augsburg, Germany. [13] CREAF, Cerdanyola del Vallès, Catalonia, Spain. [14] CSIC, Global Ecology Unit CREAF-CSIC-UAB, Catalonia, Spain. [15] Department of Biology, University of Antwerp, Wilrijk, Belgium. [16] Department of Chemistry, Green Chemistry Centre of Excellence, The University of York, York, UK. [17] College of Urban and Environmental Sciences, Laboratory for Earth Surface Processes, Peking University, Beijing, China. [18] Faculty of Environmental Science and Engineering, Kunming University of Science and Technology, Kunming, China. [19] Department of Earth System Science, Tsinghua University, Beijing, China. ✉email: rongwang@fudan.edu.cn

Biomass utilization in combination with carbon capture and storage (CCS) is a crucial option to produce energy and remove carbon from the atmosphere, thus complying with the climate change stabilization targets of the Paris Agreement[1–6]. The atmospheric $CO_2$ concentration in 2019 reached 415 ppm[7]. If we are to limit warming to 2 (1.5) °C, a cap of $1400^{+530}_{-330}$ $(480^{+260}_{-160})$ Gt $CO_2$ is allowable starting in 2011[5]. This carbon budget is currently depleted[3] at a rate of 42 Gt $CO_2$ year$^{-1}$. China is responsible for 10% of historical climate warming[8] and is today's largest emitter—it is a key player in the forthcoming global climate actions. Coal-fired power plants provide 70% of electricity in China and the young age of this network implies a significant commitment to future $CO_2$ emissions[9]. China is one of the world's largest agricultural producers with a substantial forestry sector[10]. An attractive proposition is to turn agricultural and forestry residues into electricity rather than burnt them for cooking and heating, or as waste in the field. This path would open the way for rural areas to benefit from new sources of income and reduced air pollution[11].

Developing the value chain for BECCS implies addressing the following challenges: land and biomass availability[12], costs of biomass acquisition (including soil remediation), and pretreatment[11], requirements for water and fertilizer[13], associated GHG emissions[14], investment to make power plants suitable for biomass co-firing with coal in the case of China[15], CCS[13], $CO_2$ transport from power plants to repositories for storage[16], reduced electricity generation efficiency in power plants due to CCS and co-firing[17], inertia of energy system[9], and public perception of carbon removal[18]. The potential and marginal costs of BECCS have been estimated at a regional scale (Table S1). The spatially explicit method has been applied for several countries[19,20], but there is, to our knowledge, only one study that estimates the marginal cost curve of BECCS over western North America[4]. A recent study suggests that the cost of $CO_2$ capture is affordable for China's power plants[21], but the cost of biomass and $CO_2$ transportation has not been fully assessed. It requires a sufficient understanding of the cost and associated difficulties of implementation, else, insufficient preparatory studies will cause delays in mitigation and lock-in with fossil-fuel-based energy system jeopardizing the climate targets of the Paris Agreement[22].

We aim at analyzing the potential and barriers towards a carbon emission-negative power system[23] by BECCS. Our central hypothesis is that BECCS can be harnessed by (1) biomass utilization, including collection and pretreatment of agricultural and forestry residues, energy crop production, biomass handling, and transport to power plants, (2) retrofitting of coal-fired power plants to be suitable for biomass co-firing (90% weight) and CCS, and (3) utilization of pipelines to transport $CO_2$ to geological storage sites. We estimated the supply potential of 19 ligno-cellulosic biomass feedstocks over China from agricultural and forestry residues (excluding grains) and dedicated energy crops (*Miscanthus* or high-yield crops[24]) on available marginal lands or grasslands. We optimized the consumption of biomass for electricity generation in 2836 counties to achieve a target of national emission reduction, based on: (1) a life-cycle analysis of emissions and costs and (2) spatially explicit constraints on the supply of biomass feedstock, capacity for electricity generation, and capacity for geological carbon storage. Geographical logistical constraints in a large country like China have not been investigated at a county level for the full supply chain[1–3,11,23,25]. It allows us to compare the marginal costs of decarbonizing the power system by BECCS with bioenergy or CCS alone and to identify the major routes for biomass transportation. Such an integrated strategy taking into account the specificities of China can be viewed as an early entry point to address global carbon emission mitigation and national energy security challenges —kick-starting an innovative cycle.

## Results

**Feedstocks of biomass in China.** We estimated that 3.04 Gt dry matter year$^{-1}$ of ligno-cellulosic biomass could be harvested theoretically from agricultural residues (0.79 Gt year$^{-1}$), forestry residues (0.31 Gt year$^{-1}$), and potential energy crops grown on marginal lands (0.32 Gt year$^{-1}$) and grasslands (1.62 Gt year$^{-1}$). The production of this amount of biomass equates to 5.24 Gt $CO_2$ year$^{-1}$ of carbon sequestration and 58 EJ year$^{-1}$ of primary energy. Figure 1 puts the estimates of bioenergy in the context of previous studies[26–38]. Our 2015 estimate for 11 types of agricultural residues is close to the high end of previous estimates (14.7–16.8 EJ year$^{-1}$). For example, Li et al. account for five types of agricultural residues[26], while Li et al.[27] and Yang et al.[28] derived their estimates for earlier years than ours. For forestry residues, our estimate based on the statistic of wood products is lower than that by Yang et al.[28], who estimated the wood feedstocks based on the forestry area, biomass resource yield and a constant collectable rate, but close to other estimates[34,37,38]. For energy crops, our estimate, based on the 2015 satellite land-use data[39] and the theoretical yield of *Miscanthus*[33], is close to the median of previous estimates[30,33,35–37].

With an electricity generation efficiency of 25.1% in 90% biomass co-firing plants[40], co-firing all biomass can produce 4.03 PWh year$^{-1}$ compared to 0.59 PWh year$^{-1}$ from the co-fired coal. This supply of electricity represents 80% of China's electricity demand[11] in 2015 or 49% of the projected electricity demand[41] in 2030. With a 90% efficiency of $CO_2$ capture[11], 4.72 Gt $CO_2$ year$^{-1}$ captured from the burnt biomass would take 600 years to fill the theoretical capacity of carbon storage in land repositories (2104 Gt $CO_2$) and off-shore territories (719 Gt $CO_2$) in China[42] (Fig. S1), if no logistical constraints existed. The electricity generation efficiency decreases due to biomass co-firing[43] by consuming more energy to break the oxygenated chemical bonds in the gasification of biomass[11]. This requires to increase the capacity of loading more fuels into power plants by

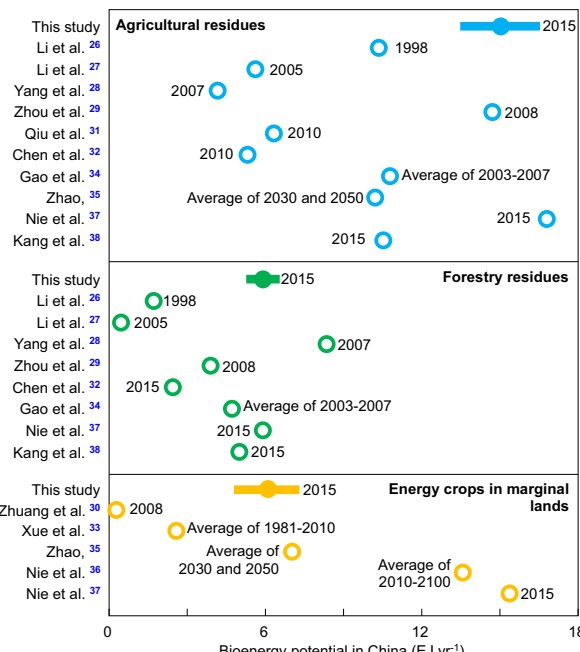

**Fig. 1 Comparison of the biomass feedstocks in this study with previous estimates.** The bioenergy is converted from the ton of standard coal equivalent (tce)[26,28,32] using a constant ratio of 29.3 GJ tce$^{-1}$ (from ref. [26]) or from the weight of biomass[30–33,37] using a constant heat content of 19 GJ (t biomass$^{-1}$) (from ref. [92]). Error bars show uncertainties in the estimation from this study.

**Table 1 Costs and $CO_2$-equivalent emissions in the scenarios of B90-2015-PC, noBiomass-2015-PC, and B90-2015-PC-noCCS.**

| Scenarios | B90-2015-PC | | | noBiomass-2015-PC | | | B90-2015-PC-noCCS | | |
|---|---|---|---|---|---|---|---|---|---|
| Abatement target (Gt $CO_{2\text{-eq}}$ $yr^{-1}$) | 0.1 | 1 | 1.9 | 0.1 | 1 | 1.9 | 0.1 | 1 | 1.9 |
| Biomass consumption (Gt $yr^{-1}$) | 0.04 | 0.4 | 0.8 | - | - | - | 0.1 | 0.8 | 1.8 |
| Total costs (billion $ $yr^{-1}$) | 5.2 | 64.4 | 131.4 | 11 | 110.1 | 209.8 | −0.7 | 23.0 | 264.6 |
| Items | Marginal cost ($ (t $CO_{2\text{-eq}}$)$^{-1}$) | | | | | | | | |
| Coal substitution (lower cost) | −31.8 | −33.7 | −34.2 | - | - | - | −82.0 | −91.0 | −220.6 |
| Retrofitting plants for co-firing and CCS | 27.1 | 28.7 | 29.1 | 43.7 | 43.7 | 43.7 | 34.7 | 38.5 | 93.4 |
| Biomass acquisition | 12.4 | 20.4 | 21.2 | - | - | - | 22.2 | 74.7 | 259.3 |
| Biomass pretreatment | 5.5 | 6.4 | 6.6 | - | - | - | 9.9 | 11.8 | 38.1 |
| Biomass transport | 1.1 | 2.3 | 13.2 | - | - | - | 1.9 | 32.6 | 274.0 |
| $CO_2$ capture and storage (CCS) | 33.0 | 34.9 | 35.4 | 52.4 | 52.4 | 52.4 | - | - | - |
| $CO_2$ transport | 2.2 | 2.7 | 3.5 | 6.9 | 7.8 | 8.4 | - | - | - |
| Fertilizer usage | 0.0 | 3.5 | 3.9 | - | - | - | 0.0 | 5.2 | 37.0 |
| Water consumption | 2.9 | 3.6 | 3.6 | 6.6 | 6.6 | 6.6 | 6.3 | 7.7 | 89.7 |
| Total | 52.3 | 68.8 | 82.3 | 109.6 | 110.5 | 111.1 | -7.0 | 79.5 | 570.8 |
| Sources | Marginal emission (t $CO_{2\text{-eq}}$ (t $CO_{2\text{-eq}}$)$^{-1}$) | | | | | | | | |
| Emission in land-use change | 0.00 | 0.00 | 0.00 | - | - | - | 0.00 | 0.00 | 0.86 |
| Emission in fertilizer production | 0.00 | 0.03 | 0.03 | - | - | - | 0.00 | 0.05 | 0.35 |
| Emission in fertilizer application | 0.00 | 0.02 | 0.02 | - | - | - | 0.00 | 0.02 | 0.15 |
| Emission in biomass treatment | 0.08 | 0.10 | 0.10 | - | - | - | 0.14 | 0.18 | 0.50 |
| Emission in biomass transport | 0.00 | 0.00 | 0.01 | - | - | - | 0.00 | 0.03 | 0.21 |
| Emission in retrofitting power plants | 0.00 | 0.00 | 0.00 | 0.00 | 0.00 | 0.00 | - | - | - |
| Emission in substituted coal | −0.44 | −0.47 | −0.47 | - | - | - | −1.14 | −1.27 | −3.08 |
| Carbon sequestration in CCS | −0.64 | −0.68 | −0.69 | −1.00 | −1.00 | −1.00 | - | - | - |
| Total | −1.00 | −1.00 | −1.00 | −1.00 | −1.00 | −1.00 | −1.00 | −1.00 | −1.00 |

increasing the rated evaporation, boiler thermal load, and the fuel flow rate[11,44,45]. These operational items are considered in the costs of retrofitting power plants (Table 1).

**Spatial distributions of biomass feedstocks, power plants, and carbon storages in China**. One of the challenges in harnessing BECCS into the energy mix of China is the spatial mismatch between biomass production areas, power plant locations, and geological storage sites. We estimated the emission reduction by considering physical flows across administrative units of provinces and counties. Agricultural and forestry residues as relatively low-cost biomass sources are mainly growing in the rural central, south, and northeast of China (Fig. 2a), with suitable marginal lands (20 Mha) and grasslands (83 Mha) for energy crops being distributed in the sparsely populated southwest and northeast of China (Fig. 2b). Coal-fired power plants are concentrated in the densely populated central and southeast parts of China (Fig. 2c). In another mismatch, most geological repositories suitable for storing carbon, such as deep saline aquifer basins (dominant), depleted oil and gas basins, and deep reserves of coal[42], are situated in the west and northeast of China (Fig. 2d). Based on the ratio of bioenergy to electricity generation in power plants and the ratio of $CO_2$ captured from biomass to carbon storage in proven repositories, there are a few counties in China with these three factors matching perfectly (Fig. S2).

We identified 0.3 Gt $year^{-1}$ of biomass that can be burnt in power plants with $CO_2$ captured and stored in the same counties, thus removing any requirements of transportation (Fig. 2e). By contrast, 0.6 or 1.3 Gt $year^{-1}$ of the biomass to be burnt in power plants requires transportation within a province or between provinces, respectively. Storing the post-combustion carbon requires transportation of 2.0 and 0.8 Gt $CO_2$ $year^{-1}$ respectively, within a given province and between provinces, probably using the existing West-East pipeline corridors constructed for natural gas[46] (Fig. 2d). In total, there are 222 GW of existing power plant capacities, which can be fuelled by co-firing 0.9 Gt $year^{-1}$ of biomass within the same county. This ultimately permits to reduce 1.0 ± 0.1 Gt $CO_2$ $year^{-1}$ from coal combustion and removes 1.4 ± 0.1 Gt $CO_2$ $year^{-1}$ from the atmosphere by transporting over different distances the captured $CO_2$.

**Marginal cost curves of BECCS in China**. We estimated the marginal costs of BECCS to reduce net $CO_2$ emissions by solving a cost-minimization problem with costs and life-cycle emission sources. We started from analyzing a scenario "B90-2015-PC", where "B90" stands for 90% biomass co-firing, "2015" stands for generating electricity by coal-fired plants in 2015 (4.24 PWh $year^{-1}$), and "PC" stands for using pulverized-coal (PC) plants. In this scenario, the marginal abatement cost increases from $49$^{+18}_{-15}$ to $103$^{+93}_{-55}$ (t $CO_{2\text{-eq}}$)$^{-1}$ when the target is to abate net emissions from 0 to 2 Gt $CO_{2\text{-eq}}$ $year^{-1}$ (Fig. 3a). The marginal cost increases to $180$^{+220}_{-78}$, $292$^{+350}_{-180}$, and $309$^{+310}_{-180}$ (t $CO_{2\text{-eq}}$)$^{-1}$ to abate, respectively, 3.0, 4.0, and 5.0 Gt $CO_{2\text{-eq}}$ $year^{-1}$. Lu et al.[11] suggested that BECCS is cost-competitive with coal in integrated gasification combined cycle (IGCC) plants at a cost of $42–52 (t $CO_{2\text{-eq}}$)$^{-1}$, comparing to $68 (t $CO_{2\text{-eq}}$)$^{-1}$ in our study. This difference is mainly due to the fact that Lu et al.[11] did not consider the dependence of transportation costs on the target of emission reduction, in the absence of logistics system for biomass and the captured $CO_2$. For CCS as a decarbonization option, the marginal cost increases from $70 (t $CO_{2\text{-eq}}$)$^{-1}$ to $150 (t $CO_{2\text{-eq}}$)$^{-1}$ due to a high cost of carbon separation and compression[47,48]. The marginal cost of BECCS in our study is close to wind or solar power, but higher than hydropower or nuclear power[49–51] (Fig. 3a). The potential of nuclear electricity is limited by safety risk[49], and the expansion of hydropower is limited by uncertainty in water resource[50].

We evaluated the impact of parameters on the marginal costs by considering alternative scenarios: "B30-2015-PC" for 30% biomass co-firing, "B90-2015-IGCC" for transferring PC to IGCC plants, "B30-2015-PC-EneCrop" for using dedicated energy crops only, "B90-2030-PC" for generating the projected electricity[41] for 2030 (9.5 PWh $year^{-1}$), "B90-2015-PC-BestCrop" for using the best-yield crops[24], "noBiomass-2015-PC" for using coal in power plants equipped with CCS, "B90-2015-PC-noCCS" for biomass co-firing without CCS and "B90-2015-PC-routes" for considering

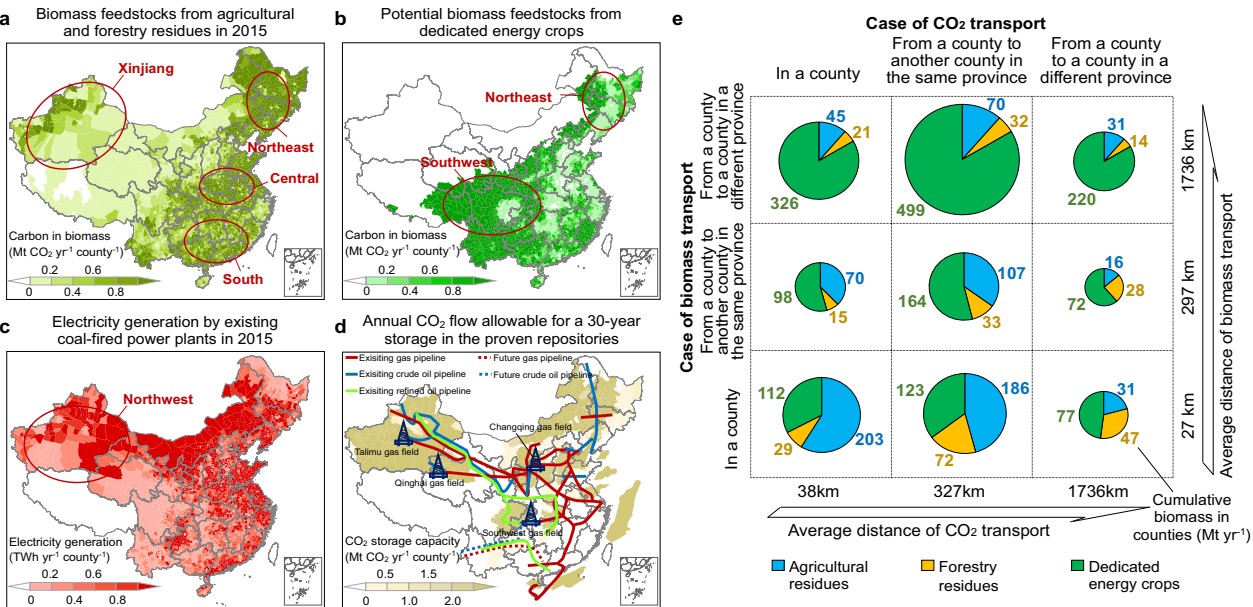

**Fig. 2 Spatial distributions of biomass feedstocks, power plants, and capacity of carbon storage in China. a**, **b** Feedstocks of ligno-cellulosic biomass from 11 types of agricultural residues and 8 types of forestry residues in 2015 and from dedicated energy crops grown on marginal lands and grasslands by county. **c** Electricity generation in coal-fired power plants by county in 2015. **d** Capacity of $CO_2$ storage for 30 years in the proven land repositories and offshore territories. The West-East pipelines constructed for natural gas are shown as lines. **e** Amounts of biomass burnt in power plants in each county based on three types of biomass transportation and three types of $CO_2$ storages.

the routes of biomass transportation between each county and the nearest ten counties (Figs. 3 and S3). For example, to abate 0.88 Gt $CO_{2-eq}$ year$^{-1}$ in China[11], if the ratio of biomass in co-firing is 30% rather than 90%, the marginal cost decreases slightly from $68 in the scenario of "B90-2015-PC" to $62 (t $CO_{2-eq}$)$^{-1}$, but the potential of emission reduction is largely weakened from 5.3 to 1.8 Gt $CO_{2-eq}$ year$^{-1}$. The marginal cost increases to $153 (t $CO_{2-eq}$)$^{-1}$, if we consider biomass co-firing in an IGCC system as in Lu et al.[11], due to high costs by shifting PC plants to IGCC[52,53]. In the absence of agricultural and forestry residues, the marginal cost would increase to $204 (t $CO_{2-eq}$)$^{-1}$; by contrast, shifting the energy crops from *Miscanthus* to the best-yield crops[24] affects the marginal cost slightly.

Because BECCS combines bioenergy and CCS, we investigated the effects by using one technology of them. Relative to the scenario of "B90-2015-PC", the marginal cost to abate 1 Gt $CO_{2-eq}$ year$^{-1}$ increases from $70 to $110 (t $CO_{2-eq}$)$^{-1}$ if only CCS is implemented on coal-fired power plants in the absence of biomass, or alternatively $80 (t $CO_{2-eq}$)$^{-1}$ in a scenario of biomass co-firing without CCS. The marginal cost in the scenario of biomass co-firing without CCS decreases to zero when abating 0.1 Gt $CO_{2-eq}$ year$^{-1}$, because the value of the substituted coal offsets the cost of utilizing biomass without requirements of long-distance transportation and retrofitting power plants for CCS (Table 1). This is in line with previous studies that bioenergy alone could be an option for a shallow decarbonization in China[11,40]. For a deeper decarbonization of >1.0 Gt $CO_{2-eq}$ year$^{-1}$, the marginal cost increases sharply without CCS due to high costs in the acquisition and logistics of biomass, where CCS becomes a cheaper option.

**Components of costs and emissions in the deployment of BECCS.** We obtained the composition of biomass feedstocks and the components of marginal costs and emissions for BECCS in China (Fig. 4). In the scenario of "B90-2015-PC", total emissions for China[11] of 9.6 Gt $CO_2$ year$^{-1}$ in 2015 can be abated by 1, 2,

and 5 Gt $CO_2$ year$^{-1}$ at a marginal cost of $69$^{+32}_{-23}$, $103$^{+93}_{-55}$, and $309$^{+310}_{-180}$ (t $CO_{2-eq}$)$^{-1}$, by investing, respectively, $64, 142, and 819 billion year$^{-1}$ (that is 0.6%, 1.3%, and 7.3% of China's GDP in 2015) into BECCS deployment. To abate 2 Gt $CO_2$ year$^{-1}$ of net emissions, 0.93$^{+0.02}_{-0.02}$ Gt $CO_2$ year$^{-1}$ of emissions from coal will be avoided by generating 1.12 PWh year$^{-1}$ in electricity, 0.92 PWh year$^{-1}$ from agricultural residues, and 0.18 PWh year$^{-1}$ from fuelwood. It substitutes 26% of coal in China's power plants and sequestrates 1.35$^{+0.09}_{-0.08}$ Gt year$^{-1}$ of atmospheric $CO_2$ into geological repositories; however, the above two emission reductions will be partly offset by 0.28$^{+0.11}_{-0.11}$ Gt $CO_{2-eq}$ year$^{-1}$ due to life-cycle emissions from retrofitting power plants for biomass co-firing and CCS (338$^{+84}_{-68}$ kt $CO_{2-eq}$ year$^{-1}$), pretreatment and logistics of biomass transportation (200$^{+60}_{-60}$ Mt $CO_{2-eq}$ year$^{-1}$), production and application of fertilizers (80$^{+50}_{-50}$ Mt $CO_{2-eq}$ year$^{-1}$), and land-use change if growing energy crops on grasslands (0$^{+0}_{-0}$ Gt $CO_{2-eq}$ year$^{-1}$). Correspondingly, the marginal cost ($103 (t $CO_{2-eq}$)$^{-1}$) decomposes into $48 (t $CO_{2-eq}$)$^{-1}$ by biomass acquisition and pretreatment, $35 (t $CO_{2-eq}$)$^{-1}$ by CCS, $29 (t $CO_{2-eq}$)$^{-1}$ by retrofitting power plants to be suitable for biomass co-firing, and CCS, $15 (t $CO_{2-eq}$)$^{-1}$ by logistics of biomass transportation, $4 (t $CO_{2-eq}$)$^{-1}$ by water consumption for irrigation in agriculture and for CCS in power plants, $3 (t $CO_{2-eq}$)$^{-1}$ by fertilizers usage, and $3 (t $CO_{2-eq}$)$^{-1}$ by $CO_2$ transportation, which is offset by $34 (t $CO_{2-eq}$)$^{-1}$ by the value of the substituted coal.

Through optimizing biomass usage at lower acquisition costs and on shorter distances for the transportation of biomass and $CO_2$, the marginal cost is minimized to achieve a target of national emission reduction. The economic cost increases in a concave shape to abate more net $CO_2$ emissions, because dedicated energy crops take up a larger fraction in bioenergy with higher costs in biomass acquisition and larger carbon emissions from land-use change[17] (Fig. 4). The price of electricity generation increases to abate more net $CO_2$ emissions. For example, to abate 1, 3, and 5 Gt $CO_{2-eq}$ year$^{-1}$ by BECCS in

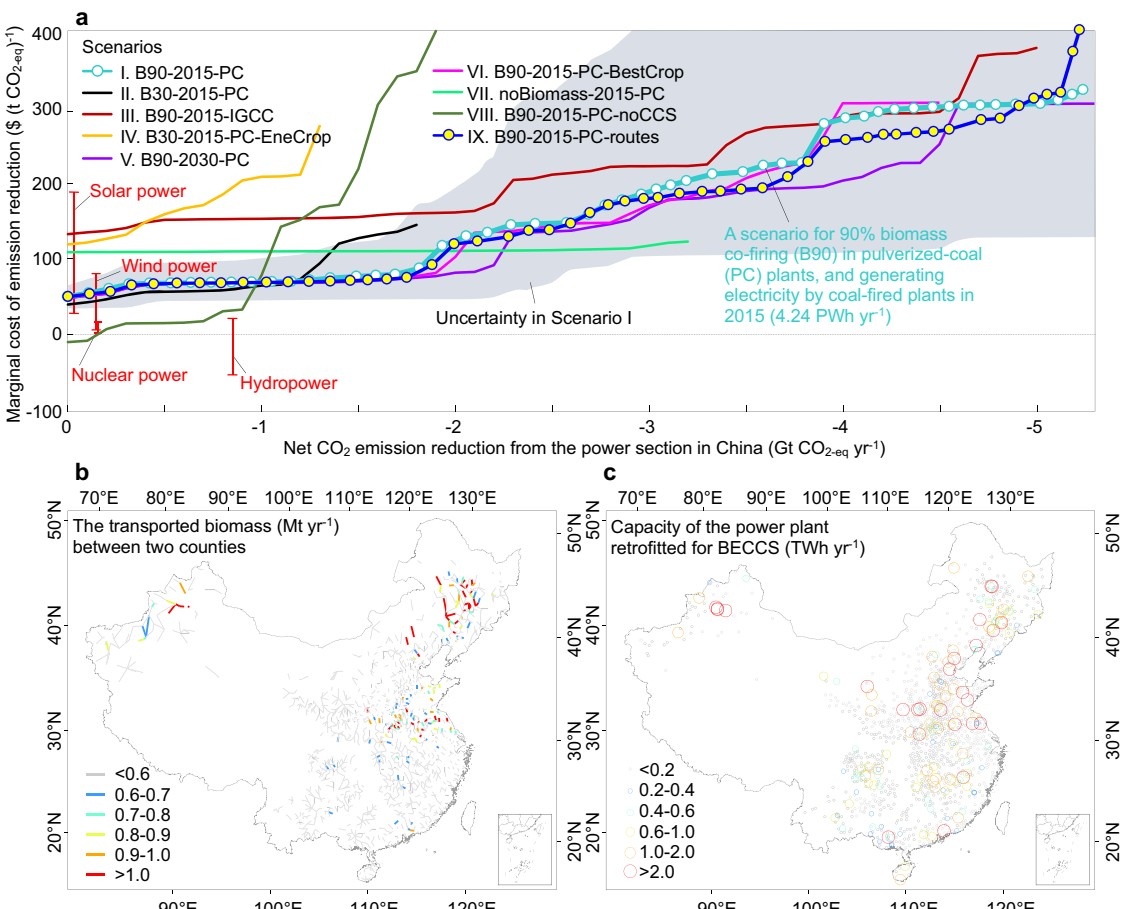

**Fig. 3 Marginal cost curves of BECCS in China. a** Marginal cost curves of BECCS in scenarios include: "B90-2015-PC" for retrofitting pulverized-coal (PC) plants under 90% biomass co-firing (B90) to generate electricity in 2015, "B30-2015-PC" for 30% biomass co-firing (B30), "B90-2015-IGCC" for transferring PC to integrated gasification combined cycle (IGCC) plants, "B30-2015-PC-EneCrop" for using dedicated energy crops (EneCrop) only, "B90-2030-PC" for generating the projected electricity in 2030, "B90-2015-PC-BestCrop" for using the best-yield crops (BestCrop), "noBiomass-2015-PC" for using coal in power plants equipped with CCS, "B90-2015-PC-noCCS" for biomass co-firing without CCS and "B90-2015-PC-routes" for considering the routes of biomass transportation between each county and the nearest ten counties. $CO_2$ emission reduction by hydropower, wind, solar, and nuclear power in China in 2015 and the marginal costs are shown as red lines. Uncertainties in the marginal costs in the scenario of "B90-2015-PC" estimated from Monte Carlo simulations are shown as the shaded area. The marginal costs of retrofitting 300, 600, ..., 15,000 of 15,244 power plants in the scenario of "B90-2015-PC" or retrofitting 300, 600, ..., 15,000 of 15,159 power plants in "B90-2015-PC-routes" are shown by circles. **b** Routes of the biomass transportation between counties to abate 2 Gt $CO_{2-eq}$ year$^{-1}$ in "B90-2015-PC-routes". **c** Spatial distribution of the retrofitted power plants for BECCS to abate 2 Gt $CO_{2-eq}$ year$^{-1}$ in "B90-2015-PC".

China, the electricity generation efficiency is reduced from 39.3% (coal-fired plants) to 25.1% (90% biomass co-firing) to generate 0.55, 1.74, and 3.34 PWh year$^{-1}$ of electricity from bioenergy, which increases the electricity price from $0.060 kWh$^{-1}$ to $0.075, 0.129, and 0.253 kWh$^{-1}$, respectively.

**Spatial distributions of bioenergy production.** Policymakers need to know where the potential for bioenergy is high and where the marginal cost is low. Using our spatially explicit network, we identified the major sources of bioenergy (Fig. 5), routes for biomass transportation (Fig. 3b), and amounts of the burnt biomass that is transported from the nearest ten counties (Fig. S3). Major bioenergy producers (>0.02 EJ county$^{-1}$ year$^{-1}$) are identified in the rural northeast and center of China to abate 1 Gt $CO_{2-eq}$ year$^{-1}$ of net emissions, and expand to the northwest, south and southwest of China to abate 3 and 5 Gt $CO_{2-eq}$ year$^{-1}$ (Fig. 5a–c). As the highest portion of economic activity is concentrated in the southeast and north of China, total costs take up a larger share of GDP in the rural northwest and northeast of

China (>5%) than the south and north of China (<1%) (Fig. 5d–f). The distributions of bioenergy production and economic costs overlap in the scenario of "B90-2015-PC" to abate 1 Gt $CO_{2-eq}$ year$^{-1}$, with 11% of counties generating 50% of bioenergy (3.97 EJ year$^{-1}$) and taking up 50% of economic costs ($32.8 billion year$^{-1}$) (Fig. 5g). By contrast, to abate 3 and 5 Gt $CO_{2-eq}$ year$^{-1}$, 50% of bioenergy will be produced by 18% and 13% of counties, which is associated with 51% and 59% of economic cost, respectively (Fig. 5h, i).

By retrofitting an integral number of power plants for BECCS, we identified the largest power plants that could be retrofitted for biomass co-firing in each county (Fig. 3c). At the provincial level, Shandong province is the largest contributor to abate net emissions of 1 Gt $CO_{2-eq}$ year$^{-1}$, providing bioenergy of 1.12 EJ year$^{-1}$ at a cost of $9.2 billion year$^{-1}$, followed by Henan province (0.99 EJ year$^{-1}$), Jiangsu province (0.76 EJ year$^{-1}$), and Hebei province (0.69 EJ year$^{-1}$) (Table S2). In contrast, Sichuan province is the largest producer of bioenergy (5.41 EJ year$^{-1}$) when the goal is to abate net emissions of 5 Gt $CO_{2-eq}$ year$^{-1}$, followed by Yunnan province (3.49 EJ year$^{-1}$), Qinghai province

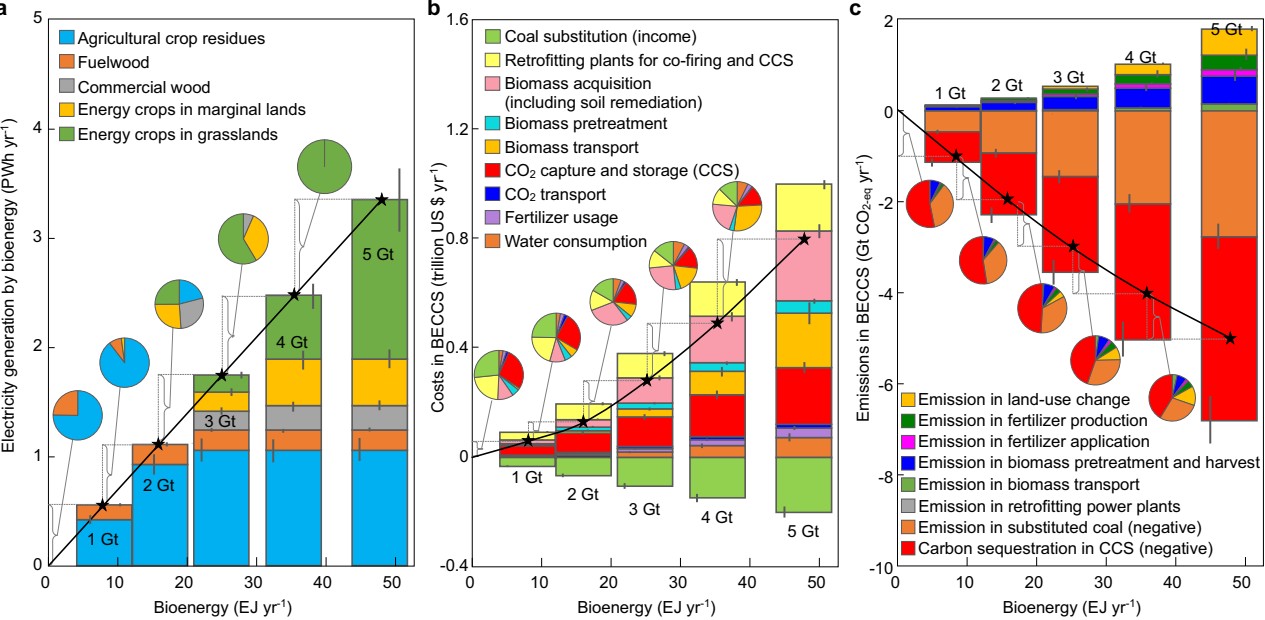

**Fig. 4 Composition of biomass feedstocks, economic costs, and carbon emissions by employing BECCS in China.** Bioenergy in electricity generation (**a**), economic costs (**b**), and life-cycle emissions (**c**) for BECCS are calculated from the scenario "B90-2015-PC" with 90% biomass co-firing in pulverized-coal plants. The components of electricity, costs, and emissions as differences between two adjacent bars under net emission reductions of 1, 2, 3, 4, and 5 Gt $CO_{2\text{-eq}}$ year$^{-1}$ are shown as pie charts. Uncertainty in the calculated electricity, costs, and emissions to abate 1, 2, 3, 4, and 5 Gt $CO_{2\text{-eq}}$ year$^{-1}$ in our Monte Carlo simulations are shown by error bars.

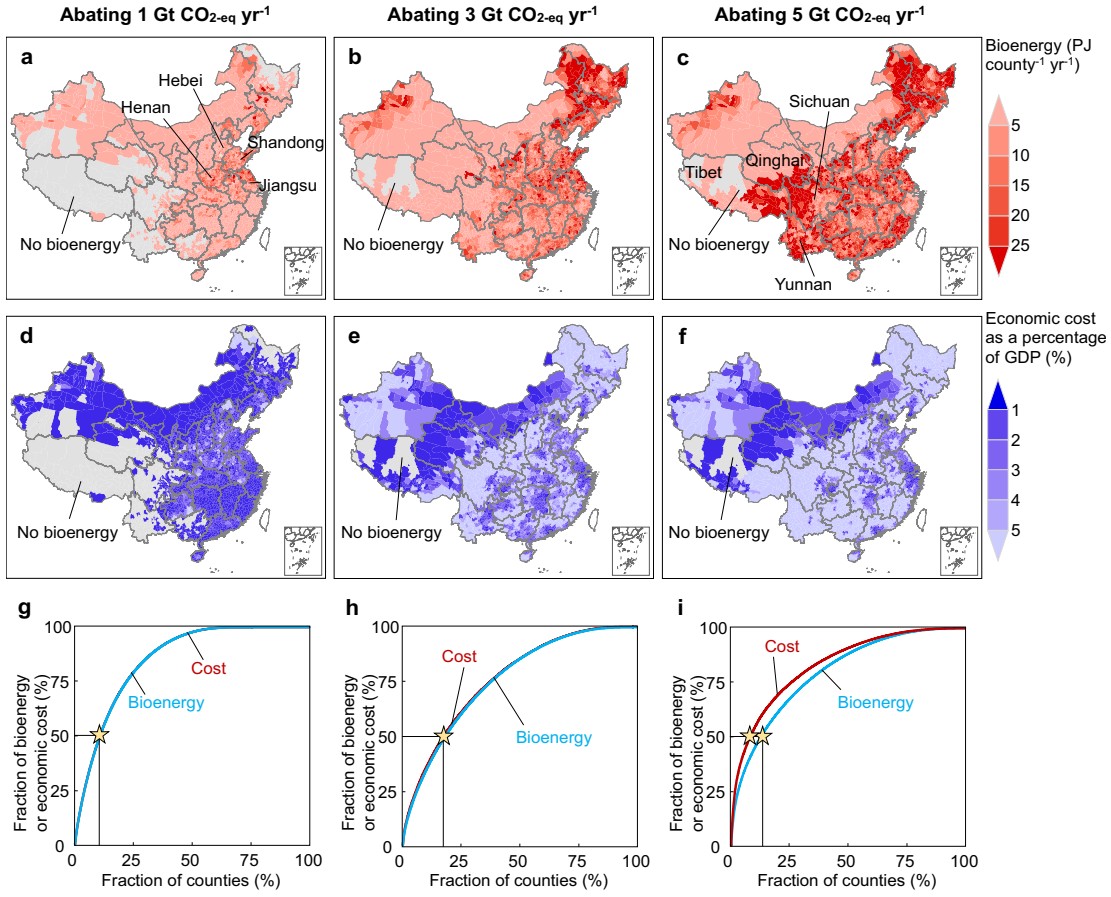

**Fig. 5 Spatial distributions of bioenergy production and economic costs of BECCS. a–c** Bioenergy production or **d–f** economic costs as a percentage of GDP in 2015 by county to abate net emissions of 1, 3, and 5 Gt $CO_{2\text{-eq}}$ year$^{-1}$ by BECCS in China, respectively. **g–i** Distributions of the bioenergy (blue) and economic costs (red) across 2836 counties in China. The stars show where 50% of bioenergy is produced.

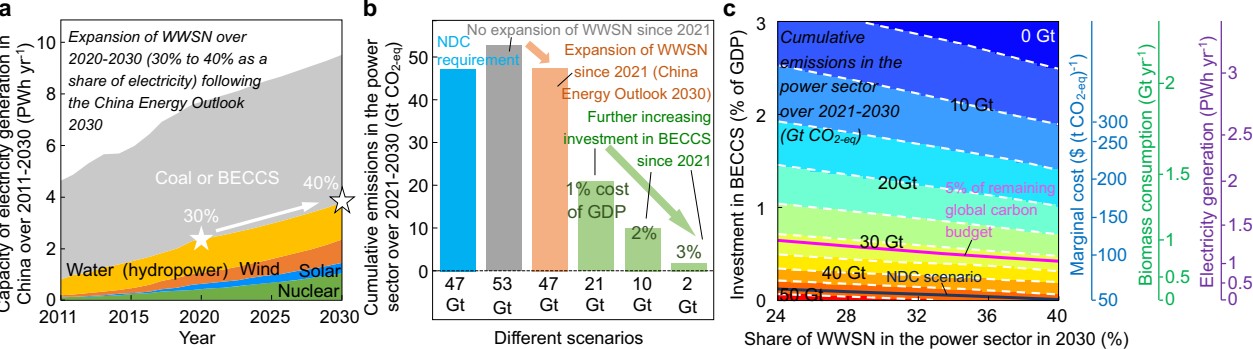

**Fig. 6 Contribution of BECCS to CO₂ emission reduction in the power sector in China. a** Path of total electricity generation and the electricity generated by water, wind, solar, and nuclear (WWSN) power in China from 2011 to 2030. The data are projected from 2021 to 2030 following the implementation of National Bureau of Statistics of China and China Energy Outlook. **b** Total $CO_2$ emissions from 2021 to 2030 under different scenarios of WWSN and BECCS in China. **c** Dependence of total $CO_2$ emissions from 2021 to 2030 on the projected share of WWSN in 2030 and the investment in BECCS from 2021 to 2030 as a percentage to GDP. The path of total electricity generation is identical to **a**. To keep total emissions from 2011 to 2030 within 5% of the remaining global carbon budget for a 2 °C limit (1400 Gt $CO_{2\text{-}eq}$), the allowable emissions have been depleted by 38.6 Gt $CO_{2\text{-}eq}$ from 2011 to 2020, and the remaining total emissions for 2021–2030 ($1400 \times 5\% - 38.6 = 31.4$ Gt $CO_{2\text{-}eq}$) are shown by the pink line. The emission permits in nationally determined contribution (NDC) are shown by a black line. The marginal costs, biomass demand, and electricity generation by BECCS in 2030 are shown on the right axes.

(3.19 EJ year$^{-1}$), and Tibet province (2.74 EJ year$^{-1}$). In China, economic costs to abate 1, 3, and 5 Gt $CO_{2\text{-}eq}$ year$^{-1}$ are estimated to be \$64, \$293, and \$819 billion year$^{-1}$, of which 29%, 37%, and 34% are spent on biomass acquisition and 8%, 9%, and 12% cover labor costs, respectively; these potential spendings provide new sources of income to farmers over rural areas in China (Table S3).

While agricultural residues can be used as cheap bioenergy, attentions should be paid to the adverse impacts on soil erosion[54] and carbon content[55]. We further considered two scenarios, where an average fraction of sustainable agricultural residues (50%) could be taken away from the land for bioenergy (Table S4) or additional technologies are applied for soil remediation at an average cost of \$76 ha$^{-1}$ year$^{-1}$ (Tables S5 and S6). Relative to the scenario of "B90-2015-PC" that does not consider these issues, the marginal cost curve moves leftward if 50% of agricultural residues can be used for bioenergy, but changes slightly if soil remediation technologies such as the plastic film mulch can be taken to protect soil from erosion and prevent soil carbon loss (Fig. S4). More studies are needed to monitor the impacts of biomass removal on soil degradation[54,55] and the effects of these soil remediation technologies[56] to ensure a sustainable development of lands.

**Role of BECCS in the decarbonization of the power sector in China.** We evaluated the contribution of BECCS to $CO_2$ emission reduction within a low-carbon energy portfolio of water (i.e., hydropower), wind, solar, and nuclear (WWSN) energy[41,57] in the power sector in China. Cumulative emissions from 2021 to 2030 in China's power sector will reach 53 Gt $CO_{2\text{-}eq}$ in a baseline scenario without expansion of WWSN and development of BECCS, and decline to 47 Gt $CO_{2\text{-}eq}$ under the current legislation[41,57] on WWSN. Based on the marginal cost curve of BECCS in the scenario of "B90-2015-PC", investing \$2.3 trillion from 2021 to 2030 (that is 1% of China's GDP) into BECCS can reduce total emissions of 21 Gt $CO_{2\text{-}eq}$; further increasing this investment to \$7.4 trillion permits to reach a carbon emission neutral power system by 2030 (Fig. 6a, b). To be consistent with the nationally determined contribution (NDC)[6], that is reducing the carbon intensity by 70% from 2005 to 2030, a scenario has to reach total emissions of 47 Gt $CO_{2\text{-}eq}$ from China's power sector from 2021 to 2030. This cap can be achieved without BECCS

(Fig. 6b). However, the choice of BECCS allows for China to reach even lower emissions than the actual NDC target. In a scenario limiting emissions from the China's power sector within 5% of the remaining global carbon budget[5] for 2 °C, an investment of \$1.1 trillion from 2021 to 2030 (that is 0.50% of GDP) is required into generating electricity of 1.1 PWh year$^{-1}$ with BECCS at a marginal cost of $82^{+96}_{-35}$ (t $CO_{2\text{-}eq}$)$^{-1}$ (Fig. 6c). Therefore, given the limits in WWSN[58], investment in BECCS is necessary to reach carbon neutrality in China's power sector.

**Policy implications.** The pressing climate change issues create incentives for near-term negative-emission technologies[59]. China's government has targeted at reaching a carbon neutrality by 2060, but the path to this target remains unclear[60]. A major challenge in emission reduction for the growing China's economy is that existing coal power plants become stranded assets combined with the need to ramp-up new generation capacities[9]. Our spatially explicit method shows that retrofitting existing coal-fired plants allows China to face these challenges in a win–win solution by 2030. First, retrofitting power plants to co-fire biomass and equipping them with CCS will form an early entry point to abate 2 Gt $CO_{2\text{-}eq}$ year$^{-1}$ of emissions. Second, taking a longer-term perspective, the stock of biomass, the growing electricity demand and the potential for geological storage needs to be matched more flexibly and efficiently in space[61,62]. If the option of retrofitting power plants for biomass co-firing and CCS is implemented, China can achieve carbon emission abatement beyond the announced NDC. This study clarifies the value of BECCS in meeting near- and long-term emission reduction targets, informs the government on the requirements for investment in BECCS, and clarifies the environmental, socioeconomic, and technological challenges of decarbonizing the power system in China.

## Methods
**Data for biomass feedstocks, power plants, and carbon storages in China.** We estimated the production of biomass from eight sources of forestry residues (commercial fuelwood, residential fuelwood, commercial roundwood, residential wood discarded by farmers, commercial bamboo, imported roundwood, imported wood pulp, imported sawn wood), agricultural residues (excluding grains) from 11 crops (rice, potato, soybean, peanut, hemp, cotton, sugar beet, wheat, maize, sesame, and rapeseed) and dedicated energy crops grown on former marginal lands and grasslands. We combined agricultural and industrial statistics for China in 2000–2015 (see data sources in Table S7). We compared data for the production of

wood and crop residues from the China Statistics Yearbook[63] and the China Forestry Statistics Yearbook[64] with data from the Food and Agriculture Organization (FAO)[65]. Because some wood products (commercial roundwood, bamboo, wood discarded by farmers, and commercial fuelwood) are not included in the FAO data, the total feedstock of biomass in the China Statistics Yearbook and the China Forestry Statistics Yearbook (adopted for this study) is higher than the FAO-based estimate (Fig. S5). Twenty-one biomass feedstocks are allocated to 2836 counties in 31 provinces over China (Fig. S6).

Our method estimating the quantity of agricultural residue is different from previous studies basing on net primary productivity of crops[28,35], which adopted the fraction of straw and grain that can be collected in that field. We considered that the fraction of straw growing in the field is equal to that for grain. The amount of agricultural residue was estimated from the quantity of harvested grain and a crop-specified straw-to-grain ratio for above-ground biomass (excluding difficult-to-obtain biomass like roots) (Table S7). To estimate the quantity of agricultural residues, 100% of agricultural residue associated with the harvested grain, rather than 100% of agricultural residue growing in the field, were considered for BECCS.

To estimate the capacities of electricity generation in coal-fired power plants that could be retrofitting for co-firing biomass and coal (e.g., 90% biomass by weight)[17], we estimated the capacities (kWh year$^{-1}$) of all coal-fired power plants as the amount of annual $CO_2$ emissions in existing coal-fired power plants (Mt $CO_2$ year$^{-1}$) divided by the $CO_2$ emissions to generate 1 kWh of electricity (0.85 t $CO_2$ MWh$^{-1}$)[66]. The $CO_2$ emissions from coal-fired power plants are taken as an average of emissions in 2014 and 2016 in the Multi-resolution Emission Inventory for China at a resolution of 0.5° × 0.5°, developed at Tsinghua University[67]. The gridded electricity generation capacities were allocated to 2836 counties over China.

Major reservoirs suitable for carbon storage in the territories of China include deep saline aquifer basins, depleted oil and gas basins, and deep reserves of coal. We compiled a bottom-up inventory of carbon storage in 2836 counties over land and as a total over land and off-shore territories based on the spatial distribution of these reservoirs[42,68,69]. The national capacity for storing $CO_2$ (2823 Gt $CO_2$) is close to the higher bound in the range of previous estimates (403–2830 Gt $CO_2$)[70,71]. The distributions of carbon storage capacity in different reservoirs are shown in Fig. S7. We assumed a lifetime of 30 years for reservoirs to match medium-term mitigation target and avoid uncertainties in the longer future—the choice of this parameter slightly affects the constraints on carbon storage in counties where bioenergy is produced (Fig. S8). We considered that these repositories are no longer used for further storage if it reaches the full capacity of storage after 30 years, and the stored carbon cannot be released. Increasing the lifetime of these reservoirs from 10 to 100 years reduces the amount of $CO_2$ stored within those counties where $CO_2$ can be captured from 1.10 to 0.96 Gt year$^{-1}$, as the capacity for carbon storage in each county is reduced by year.

Data for biomass feedstocks, gross domestic product (GDP), capacities of the electricity generation by power plants in 2015, and capacity for carbon storage are given in the supplementary data 1.

## Potential of growing dedicated energy crops.

We considered energy crops grown on marginal lands (including sparsely vegetated grassland, permanent glaciers and snow, beaches, sandy land, saline and alkaline land, and bare land), and grasslands in 2015 (Table S7). In China, the total area is 173 Mha for marginal lands, and 208 Mha for medium and high vegetated grasslands. We considered only lands with minimum of monthly mean temperature above −23 °C and annual precipitation above 400 mm as suitable for growing energy crops[33]. This filter reduces the total area where energy crops can be grown to 20 Mha for marginal lands and 83 Mha for grasslands. In previous studies, the suitable area is estimated to be 34 Mha for winter-fallow paddy lands and uncultivated marginal lands[72], 35–75 Mha for marginal uncultivated lands[73], 82 Mha for marginal crop lands, and 79 Mha for marginal grass, savanna, and shrub lands[74].

We estimated the yield of dedicated energy crops in two approaches. First, we estimated the theoretical yield of *Miscanthus* based on minimum of monthly average temperature, annual average precipitation rate, and annual average active sunshine hours (see their spatial distributions in Fig. S9). The yield ranges from 10 to 60 t dry matter ha$^{-1}$ year$^{-1}$ for 2836 counties in China, with an average of 21.9 t dry matter ha$^{-1}$ year$^{-1}$ over the counties. Our estimate is comparable to estimates[33] of 18–45 t dry matter ha$^{-1}$ year$^{-1}$ or an estimate[17] of 24.3 t dry matter ha$^{-1}$ year$^{-1}$, slightly higher than estimates[24] of 1–20 t dry matter ha$^{-1}$ year$^{-1}$ for mixed crops, and lower than an estimate[75] of 35.76 t dry matter ha$^{-1}$ year$^{-1}$ in Illinois. Second, we adopted a global map of yield based on the best-yield crops produced in a recent study[24]. The cost of growing energy crops as well as the life-cycle emissions of GHGs were estimated in our model. Parameters used to calculate the unit costs and emissions are listed in Tables S8–S11.

## Optimization of the biomass burnt in power plans retrofitted for BECCS.

For power plants in a given county $i$, we considered biomass grown in the same county $i$ and transported within the county (case $x = 1$), biomass grown in other counties of the same province and transported to county $i$ (case $x = 2$) and biomass grown and transported from counties in other provinces (case $x = 3$). We considered $CO_2$ captured from power plants in county $i$ and transported to storage site in county $i$ (case $y = 1$), $CO_2$ captured from power plants in county $i$ and transported to storage site in other counties of the same province (case $y = 2$), and $CO_2$ captured

from power plants in county $i$ and transported to storage site in other provinces (case $y = 3$).

Following a cost-minimization approach, we optimized the consumption of biomass in power plants retrofitted for biomass co-firing in 2836 counties ($T_{ixyh}$) to minimize the economic cost under a target of net emission reduction ($F$) as:

$$\min_{T_{ixyh}} C = \sum_{i=1}^{2836} \sum_{h=1}^{21} \sum_{x=1}^{3} \sum_{y=1}^{3} \mu_{ixyh} T_{ixyh} - \frac{\kappa E}{\upsilon \lambda_c}, \forall \frac{\varphi E}{\upsilon \lambda_c} - N = F \quad (1)$$

$$E = \eta \sum_{i=1}^{2836} \sum_{h=1}^{21} \sum_{x=1}^{3} \sum_{y=1}^{3} \lambda_h T_{ixyh} \quad (2)$$

$$N = \sum_{i=1}^{2836} \sum_{h=1}^{21} \sum_{x=1}^{3} \sum_{y=1}^{3} r_{ixyh} T_{ixyh} \quad (3)$$

where $C$ is total economic cost; $h$ is a type of biomass; $F$ is a target of net emission reduction; $E$ is electricity generation of coal that is replaced by biomass; $N$ is life-cycle emission; $T_{ixyh}$ is consumption of biomass; $\mu_{ixyh}$ and $r_{ixyh}$ are unit cost and emission in utilization of biomass $h$; $\upsilon$ and $\eta$ denote the electricity generation efficiency of coal and co-firing of biomass with coal, respectively; $\kappa$ is the price of coal; $\lambda_c$ and $\lambda_h$ is heat content of coal and biomass $h$, respectively; and $\varphi$ is $CO_2$ emission factor of coal.

Then, the additional cost per ton carbon reduced ($\xi$), equivalent to the marginal cost of BECCS, is expressed as:

$$\xi = \frac{dC}{dF} \quad (4)$$

where $dC$ is the increase in total costs when the target of net emission reduction increases from $F$ to $(F + dF)$.

We considered that an integral number of power plants can be retrofitted for biomass co-firing to achieve a target of $CO_2$ emission reduction, based on the capacity of 15,462 power plants (i.e., the minimal capacity is 215,000 MWh) in 2836 counties over China[76]. First, we calculated the electricity generated by the power plants retrofitted for biomass co-firing from the optimized consumption of biomass in each county ($T_{ixyh}$). Second, we ranked the power plants in this county in order of the capacity and found the last power plant that is retrofitted to ensure that the electricity generated by retrofitted plants is closest to the target in this county. The marginal cost of BECCS and $CO_2$ emission reduction are re-calculated when an integral number of power plants are retrofitted in all counties.

## Electricity generation efficiency of the power plants retrofitted for BECCS.

Transformation of the power sector to BECCS operation is a massive project for China[40]. To reach a 90% biomass co-firing ratio for a deep decarbonization target, it requires boiler adjustments and biomass pretreatment technologies, such as drying, grinding, milling, torrefaction, pelletization[77–79]. Following the practice in the literature[11], we considered the fraction of biomass co-firing to vary from 30 to 90%. The electricity generation efficiency in pulverized-coal power plants is 39.3% for coal combustion, 36.2% for 90% biomass co-firing without CCS, 27.3% for 30% of biomass co-firing with CCS, and 25.1% for 90% of biomass co-firing with CCS[40,80]. For IGCC plants, we adopted an electricity generation efficiency of 35.8% in China[11]. More studies are needed to examine the feasibility of a 90% biomass co-firing ratio and enhance the electricity generation efficiency for biomass co-firing.

## Unit costs and emissions of burning biomass in power plans retrofitted for BECCS.

Based on a life-cycle analysis, the unit cost ($ spent to utilize 1 t biomass) of BECCS includes:

$$\mu_{ixyh} = \mu_h^a + \mu_h^f + \mu_{ix}^b + \mu_{iy}^t + \mu_h^p + \mu_h^g + \mu_h^w + \mu_h^d + \mu_h^{cap} \quad (5)$$

where $\mu_h^a$ is unit cost of biomass acquisition, including costs of seeding, pesticide, cultivation, sowing, harvesting, land, labor inputs, and soil remediation technologies; $\mu_h^f$ is unit cost of fertilizer used, based on the demand of N, $P_2O_5$, and $K_2O$ fertilizers to produce 1 t biomass, and the corresponding fertilizer prices; $\mu_{ix}^b$ is unit cost of biomass transport, as a function of the distance transported and the cost to transport one ton of biomass over 1 km by vehicles (using diesel); $\mu_{iy}^t$ is unit cost of $CO_2$ transport, based on the length of pipeline, $CO_2$ mass flow rate, location factor, terrain factor, capital recovery factor, and annualized constant ratio of operational and maintenance costs relative to capital investments (see Eq. (16) below); $\mu_h^p$ is unit cost of biomass pretreatment, based on the consumptions of diesel and electricity in pretreatment, and the corresponding energy price; $\mu_h^g$ and $\mu_h^w$ are unit cost of water used for irrigation to grow crops and CCS and cooling tower in power plants, respectively, based on the consumption of water, and the corresponding water prices; $\mu_h^d$ is unit cost of retrofitting facilities in power plants to be suitable for co-firing and CCS, based on the capacity of power plants, capacity factor, investment costs, fixed and variable operation and maintenance costs, full-load running times, and capital recovery factor (see Eq. (17) below); $\mu_h^{cap}$ is unit cost of $CO_2$ capture and storage in power plants[81,82]. Acquisition requires fixed field operations that lead to dependence of the harvest cost on the yield. We assumed that the harvest cost for low-yielding residues (soy, potato, and sugar beet) is two times of

that for other residues[83], which has a minor impact on our result. The parameters used to calculate these cost items are listed in Table S8.

The unit emission (t CO₂-eq emitted to utilize 1 t biomass) is calculated as:

$$r_{ixyh} = r_h^p + r_h^{zp} + r_h^{za} + r_{ix}^b + r_h^d + r_h^{uc} - r_h^s \quad (6)$$

where $r_h^p$ is unit emission of GHGs (converted to equivalent amounts of CO₂) in biomass pretreatment and harvesting, based on the consumption of diesel and electricity, and the corresponding equivalent CO₂ emission factors; $r_h^{zp}$ and $r_h^{za}$ are unit emission from production and application of fertilizers, respectively, based on the demand of N, P₂O₅, and K₂O fertilizers, equivalent CO₂ emission factor in fertilizer production, and equivalent CO₂ emission factor in fertilizer application; $r_{ix}^b$ is unit emission of equivalent CO₂ from biomass transport, based on the distance transported, the equivalent CO₂ emission factor to transport one ton of biomass over 1 km by vehicles (using diesel); $r_h^d$ is unit emission of equivalent CO₂ from retrofitting power plants, based on electricity consumption in the retrofitting and equivalent CO₂ emission factors to produce electricity by coal; $r_h^{uc}$ is unit emission of CO₂ from land-use change to grow dedicated energy crops on marginal lands (zero emissions) and grasslands (30 ± 3% of CO₂ emissions relative to CO₂ in biomass)[17]; and $r_h^s$ is unit sequestration of CO₂ from biomass, based on carbon content in biomass and the fraction of CO₂ capture. The parameters used to calculate these emission items are listed in Table S9.

**Constraints on biomass supply, electricity generation, and carbon storage by county.** For power plants with a co-firing of biomass and coal (90% by weight biomass), the constraints on the capacity of electricity generation are expressed as:

$$\sum_{h=1}^{21}\sum_{x=1}^{3}\sum_{y=1}^{3} \eta T_{ixyh}(\lambda_h + \frac{\lambda_c}{9})/3.6 \leq P_i, \text{ for } i = 1 \text{ to } 2836 \quad (7)$$

where $\lambda_h$ and $\lambda_c$ are the heat content of biomass $h$ and coal, respectively; $\eta$ is electricity generation efficiency; $P_i$ is the electricity generation by power plants in county $i$ in 2015; and the factor 3.6 converts heat content (GJ) to electricity (MWh).

For county $i$ in province $j$, we considered $m_j$ and $(m_j + n_j)$ as the first and last county in this province $j$ ($j = 1$ to 31 for 31 provinces in China). Thus, the constraints on the supply of biomass grown in the same county $i$ and transported within the county (case $x = 1$), biomass grown in other counties of the same province and transported to county $i$ (case $x = 2$) and biomass grown and transported from counties in other provinces (case $x = 3$) are, respectively, expressed as:

$$\sum_{y=1}^{3} T_{ixyh} \leq B_{ih}, i = 1 \text{ to } 2836, x = 1 \text{ and } h = 1 \text{ to } 21 \quad (8)$$

$$\sum_{i=m_j}^{m_j+n_j}\sum_{x=1}^{2}\sum_{y=1}^{3} T_{ixyh} \leq \sum_{i=m_j}^{m_j+n_j} B_{ih}, j = 1 \text{ to } 31 \text{ and } h = 1 \text{ to } 21 \quad (9)$$

$$\sum_{i=1}^{2836}\sum_{x=1}^{3}\sum_{y=1}^{3} T_{ixyh} \leq \sum_{i=1}^{2836} B_{ih}, h = 1 \text{ to } 21 \quad (10)$$

where $B_{ih}$ is feedstock of biomass $h$ in county $i$.

Similarly, the constraints on CO₂ captured from power plants in county $i$ and transported to storage site in county $i$ (case $y = 1$), CO₂ captured from power plants in county $i$ and transported to storage site in other counties of the same province (case $y = 2$), and CO₂ captured from power plants in county $i$ and transported to storage site in other provinces (case $y = 3$) are, respectively, expressed as:

$$\sum_{h=1}^{21}\sum_{x=1}^{3} r^s T_{ixyh} \leq S_i, y = 1 \text{ and } i = 1 \text{ to } 2836 \quad (11)$$

$$\sum_{i=m_j}^{m_j+n_j}\sum_{h=1}^{21}\sum_{x=1}^{3}\sum_{y=1}^{2} r^s T_{ixyh} \leq \sum_{i=m_j}^{m_j+n_j} S_i, j = 1 \text{ to } 31 \quad (12)$$

$$\sum_{i=1}^{2836}\sum_{h=1}^{21}\sum_{x=1}^{3}\sum_{y=1}^{3} r^s T_{ixyh} \leq \sum_{i=1}^{2836} S_i \quad (13)$$

where $S_i$ is capacity of carbon storage in county $i$.

$T_{ixyh}$ is numerically solved as a linear-programing problem[84] using the MATLAB function "LINPROG" (https://ww2.mathworks.cn/help/optim/ug/linprog.html) to derive 536,004 independent variables from 65,998 inequality function.

The approximation of $x$ and $y$ allows us to consider the priority of biomass and CO₂ transportation in routes over shorter distances. Because uncertainties in transportation costs increase from $x, y = 1$ to $x, y = 3$, we considered four sensitivity tests, where the distance of transportation is changed by ±50% for the scenarios of $x, y = 3$ and $x, y = 2, 3$ (Fig. S4). We considered an additional uncertainty of ±50% in the distance for biomass and CO₂ transportation in the scenarios of $x, y = 2, 3$ in our Monte Carlo simulations, and explicitly modeled the transportation of biomass between each county and the nearest ten counties in the scenario of "B90-2015-PC-routes".

**Transportation of biomass between a county and the nearest ten counties.** Johnson et al. considered the routes of biomass transportation in a model by optimizing the amount of biomass transported between two counties to achieve the lowest cost for bioenergy in power plants in the USA[85]. The computational loads are too heavy to consider the transportation between any two of 2836 counties in China. Because the unit cost of the transportation of biomass is higher than CO₂, we used a similar method as Johnson et al.[85] to consider the transportation of biomass from the nearest ten counties to power plants in each county. For each route of the biomass transportation, we take the center of non-urban pixels in a county as the starting point of the route, and take the center of power plants in a county as the destination of the route. The distance of transportation and the unit costs of biomass transportation are calculated for each county correspondingly (Table S10). For power plants in each county, we take the amount of the burnt biomass transported from the nearest ten counties, other counties in this province, or counties in other provinces as the variables for optimization in our model to minimize the total economic cost under a target of CO₂ emission reduction. For the retrofitted power plants in a county, the burnt biomass can be harvested from the same county without long-distance transportation or transported from the nearest counties or other counties at a longer distance. By varying the source of biomass ($x$) from 1 to 12 in this scenario, the optimization function (similar to Eq. (1)) is expressed as:

$$\min_{T_{ixyh}} C = \sum_{i=1}^{2836}\sum_{h=1}^{21}\sum_{x=1}^{12}\sum_{y=1}^{3} \mu_{ixyh} T_{ixyh} - \frac{\kappa E}{v\lambda_c}, \forall \frac{\varphi E}{v\lambda_c} - N = F \quad (14)$$

where $T_{ixyh}$ is the biomass burnt in county $i$ from a specific source $x$ (1 to 10 for the nearest ten cities including county $i$, 11 for other counties in this province and 12 for counties in other provinces). While we determine the distance of biomass transportation between any two counties for $x = 1-10$, we use the average distance between county $i$ and other counties for $x = 11, 12$ to estimate the unit cost and emission of biomass transportation. For each county, we identify that the biomass is transported from county $i$ to county $i_j$ in a route $x$, and then the constraint on biomass availability in county $i$ is expressed as:

$$\sum_{y=1}^{3} T_{i1yh} + \sum_{y=1}^{3}\sum_{j=1}^{n} T_{i_jxyh} \leq B_{ih}, i = 1 \text{ to } 2836 \text{ and } h = 1 \text{ to } 21 \quad (15)$$

where $n$ is the number of counties, to which the biomass harvested in county $i$ has been transported.

**Parameterization of the cost of CO₂ transport by pipelines.** Pipelines are recommended by the Intergovernmental Panel on Climate Change for CO₂ transport[86]. We determined unit cost ($\mu_{iy}^t$, $ (t biomass)$^{-1}$) of pipeline constructed to transport CO₂ captured from biomass burnt in power plants using a function developed by McCollum et al.[87] as:

$$\mu_{iy}^t = 1.31 \cdot 9770 \cdot L_{iy}^{0.13} \cdot (\chi_{iy}/365)^{0.35} \cdot F^L \cdot F^T \cdot (\text{CRF} + \text{OM}) \cdot \frac{L_{iy}}{\chi_{iy}} \cdot \text{CC} \cdot 3.67 \cdot \text{EC} \quad (16)$$

where 1.31 is a deflation rate to convert the 2005 US dollar to the 2015 US dollar, 365 is days in 1 year, 9770 is a calibrated constant[87], $\chi_{iy}$ is CO₂ mass flow rate (t yr$^{-1}$), $L_{iy}$ is the length of pipeline (km), $F^L$ is a location factor, $F^T$ is a terrain factor, CRF is capital recovery factor (year$^{-1}$, see Eq. (17)), OM is an annualized constant ratio of operational and maintenance costs relative to capital investments (0.025 year$^{-1}$)[87], CC is carbon content in biomass (47%)[17,88,89], 3.67 converts C to CO₂, and EC is efficiency of CO₂ capture (90%). We adopt $F^L = F^T = 1$, which can be improved when we can obtain more information on the routes of pipelines. Using a constant discount rate ($r$) of 7% per year[11], the capital recovery factor (CRF) is derived as:

$$\text{CRF} = \frac{r(1+r)^n}{(1+r)^n - 1}(1+r)^t \quad (17)$$

where $n$ is the facility's operational lifetime (35 years)[11] and $t$ is an average construction time (3 years)[11].

The unit cost is extremely high when the CO₂ flow rate is low (unit cost is infinite if $\chi$ is close to zero). To estimate the potential of BECCS, we considered that pipelines are constructed to transport a given amount of CO₂, that is 40% of total emissions from all power plants in 2030 according to a recent study for China[90], which does not depend on the deployment of BECCS. We considered CO₂ transported in a network with pipes converging from counties to one provincial node and from 31 provinces to the national West-East pipeline[46,91]. The transport network is conceptually mapped in Fig. S10. We approximated the distance of CO₂ transported in county $i$ ($D_i$), province $j$ ($D_j$) or the country ($D_0$) as the radii of circle with area equal to the area of the county, province or the country, $D = (A/\pi)^{1/2}$, where $A$ is the area. This approximation allows us to qualitatively calculate the unit cost in three cases using Eq. (16) as: (1) $\mu_i^t$ for unit cost of transport in county $i$ with $L = D_i$ and $\chi$ equal to 40% of CO₂ emissions from all power plants in county $i$;

(2) $\mu_i^t + \mu_j^t$, where $\mu_j^t$ is unit cost of transport from county $i$ in province $j$ to other counties in province $j$ with $L = D_j - D_i$ and $\chi$ equal to 40% of $CO_2$ emissions from all power plants in province $j$; and (3) $\mu_i^t + \mu_j^t + \mu_0^t$, where $\mu_0^t$ is unit cost of transport from province $j$ to other provinces with $L = D_0 - D_j$ and $\chi$ equal to 40% of $CO_2$ emissions from all power plants in the country.

Sanchez et al. suggested that the flow rate of $CO_2$ transported by pipelines is limited by the nominal pipeline size[16], and we adopted the minimal ($\chi^{MIN} = 89$ kt year$^{-1}$) and maximal ($\chi^{MAX} = 26{,}698$ kt year$^{-1}$) flow rate from that study due to lack of standards in China. Therefore, the capacity of CCS is set to be zero for a county if $\chi$ is below 89 kt year$^{-1}$, so that there is no pipeline to transport $CO_2$ captured in power plants in this county; $\chi$ is set to 26,698 kt year$^{-1}$ if it is above 26,698 kt year$^{-1}$, so that unit cost does not decrease when $\chi$ exceeds this threshold. We assumed that, if the $CO_2$ emissions exceed 40% of the $CO_2$ capacity of the county, another pipeline of the same size at the same unit cost is constructed, because the flux of $CO_2$ transported in pipeline size[16] is limited to <26,698 kt year$^{-1}$. The unit cost of $CO_2$ transport by county is mapped in Fig. S11.

**Electricity generation by hydropower, wind, solar, and nuclear power.** The total electricity generation and capacities of WWSN in 31 provinces were compiled from 2011 to 2019 using data from the National Bureau of Statistics of China[57], predicted for 2020 using data from the China Energy Outlook 2030[41], and predicted for 2021–2030 based on a constant growth rate of total electricity generation (2.3% per year), hydropower (1.1% per year), wind (8.4% per year), solar (7.7% per year), and nuclear (10.2% per year) power according to the China Energy Outlook 2030 for the development of renewable energy[41]. We considered GDP to be growing at a constant rate of 5.5% per year from 2020 to 2025 and 4.5% per year from 2026 to 2030, according to the China Energy Outlook 2030[41]. The electricity not produced by WWSN will be generated by coal or BECCS, and total $CO_2$ emissions were calculated using the emission factor of coal (0.85 t $CO_2$ to produce 1 MWh)[66] or WWSN (0 t $CO_2$ to produce 1 MWh) and the negative emissions of BECCS.

To investigate the impact of expanding WWSN on cumulative carbon emissions from 2021 to 2030, the rates of growth in WWSN power from 2021 to 2030 were reduced proportionally to achieve a given fraction of total WWSN power in the total electricity generation[41,57] in 2030. According to the China's NDC target (that is to reduce the carbon intensity in the power sector from 0.58 to 0.17 kg $CO_2$ \$$^{-1}$ from 2005 to 2030)[6], the rates of growth in WWSN are required to increase by a factor of 0.013 relative to the rates projected by the China Energy Outlook 2030[41,57]. Meanwhile, the share of WWSN in electricity generation in 2030 will reach 40.3% (compared to 39.8% in the China Energy Outlook 2030), resulting in total emissions of 47.2 Gt $CO_2$ from China's power sector from 2021 to 2030 without considering BECCS. We consider that China contributes 10% to historical global radiative forcing[8], and we assume that China agrees to keep this figure as its quota for the future, as a static perspective on the composition of future emissions. Because the power sector contributes 50% to total $CO_2$ emissions in China[11], the World may expect that total emissions from the China's power sector stay within 5% of the remaining global carbon budget[5] to limit warming to 2 °C. The marginal costs of WWSN are estimated based on the prices of hydropower, wind, solar, and nuclear power[51] in China in 2016. We adopted the projected capacities from the plan of the Chinese government[41]. This allows us to focus on analyzing the potential and costs of BECCS to decarbonize the electricity system.

**Uncertainty analyses by Monte Carlo simulations.** To assess the uncertainty, we adopted the range of parameters in (1) the cost calculations, including unit cost of biomass acquisition, unit cost of fertilizer usage, unit cost of biomass and $CO_2$ pipeline transport, unit cost of biomass pretreatment, unit cost of water consumption for irrigation to grow crops or in power plants for CCS, unit cost of retrofitting power plants, and unit cost of capturing and storing of $CO_2$, (2) the life-cycle emissions, including unit emission of equivalent $CO_2$ from biomass pretreatment, unit emission from production and application of fertilizers, unit emission from diesel consumption in biomass transport, unit emission from diesel used in retrofitting power plants, and unit emission of equivalent $CO_2$ from land-use change, (3) the carbon content of biomass, and (4) an additional uncertainty of ±50% in the scenarios of $x$, $y = 2$, 3 for the distance of biomass and $CO_2$ transportation. We adopted random values for these parameters from the uniform distributions (see their ranges in Tables S8 and S9) in the Monte Carlo simulations to estimate uncertainties in the marginal costs of BECCS.

**Reporting summary.** Further information on research design is available in the Nature Research Reporting Summary linked to this article.

## Data availability
Data for biomass feedstocks, gross domestic product (GDP), capacities of the electricity generation by power plants in 2015, and capacity for carbon storage are provided in Supplementary data 1. Further data are available from the corresponding author on request.

## Code availability
The code used to optimize the retrofitting of power plants by county for biomass co-firing with CCS, and to estimate the capacity of electricity generation in China over 2011–2030 are provided in Supplementary software 1. Further data are available from the corresponding author on request.

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

## Acknowledgements
This work was funded by the National Natural Science Foundation of China (41877506, 21925601, 41922057), National Key Research and Development Program of China (No. 2017YFC0212200), the Chinese Thousand Youth Talents Program, Chinese Academy of Science (XDA23010100), the ERC-Synergy IMBALANCE-P, and the ANR CLAND Convergence Institute 16-CONV-0003. The research leading to these results was also supported by the PEGASOS (01LA1826C) project funded by the German Federal Ministry of Education and Research (BMBF). We gratefully acknowledge the useful comments offered by three anonymous referees which are helpful to improve the manuscript.

## Author contributions
R.W. designed the research and wrote the paper. X.X. analyzed data and prepared graphs. N.B. provided expertise in economics. O.B., P.C., J.P., D.G., I.A.J., Y.B., J.Clark, J.Cao, J.M., B.P., L.W., and X.Yang provided expertise in biogeochemistry. S.Z., X.Ye, Y.W., Q.L., G.L., L.W., X.T., J.Clark, Y.Y., and S.X. provided data and tools. Q.L., J.Chen, G.S., W.L., and J.Chen provided expertise in bioenergy. All coauthors interpreted results and provided comments on the manuscript.

## Competing interests
The authors declare no competing interests.
