## [Peer Review File · Nature Communications]

Editorial Note: Parts of this Peer Review File have been amended to remove third-party material where no permission to publish could be obtained.

REVIEWER COMMENTS

Reviewer #1 (Remarks to the Author):

This paper is an absolute treasure trove of valuable data, and presents strong and robust analysis, therefore I strongly support its publication, essentially as is.

If possible, I would like to see the graphics in the main paper enlarged so as to be easier to read, and I'd also like to draw the attention of the authors to the work of Singh et al, "Large-Scale Affordable CO₂ Capture Is Possible by 2030", published in Joule, 3(9), 2154-2164, 2019 - the approach is somewhat similar.

Thank you for what I anticipate being a landmark paper.

Spatially explicit analysis identifies significant potential for bioenergy with carbon capture and storage in China

Summary

The paper presents a linear program where co-firing of biomass with coal in existing power plant fleet is optimized, when the CO₂ emissions of the combusted biomass is captured and stored. The model applies county level data from China. The results suggest quite promising BECCS scenarios for China.

General impression

The paper presents a set of important research questions that contribute to the solution of the global climate change problem. Unfortunately, for me, the manuscript was quite laborious to read. The results offered a lot of numbers, which I had difficulty to put in perspective. It was not clear, where does these number come from and what are the underlying assumption.

Hence, for a person like me, the reading of results and numbers is meaningless, before I can see the methodology behind the numbers. So, I focused on the Methods section and the Supplement. While the data set was impressive, there were also issues that I raise below. The most worrisome aspect from my perspective was the contents (and the presentation) of the linear programming model. I raise these issues below.

All in all, if my worries on the data and the model prove unwarranted, there is a feeling that the results are quite optimistic. Is it realistic to assume that it is possible to extract all the biomass with a constant unit cost from the whole county? If not, the modeling approach is in jeopardy. If it is OK, then the paper a promising future.

In addition, there were severe issues in overall presentation. I mention some issues in my “Comments and suggestions” and in “Smaller comments”.

Given the high requirements of Nature Communications, I fear that the manuscript will not have enough scientific impact to warrant publication in this journal. Yet, the manuscript has a high reader potential given the nature of the data and importance of the research question. Before that, however, the manuscript should be polished, and the modeling revised.

Comments and suggestion

1. Optimization problem

Optimization problem (Eq. (1)) is a little bit confusing. The results of the manuscript suggest that the authors solve a cost minimization problem with a given emission reduction target, but instead the formulation is a profit maximization problem with a given carbon emission price. However, row 427 suggests that there is an emission reduction target together with a given carbon price. In addition, Eq (1) suggests that decision variable was E , which seems misleading as the real decision variables should, to my understanding, be T_{ixy}^h .

A major issue is the formulation. I list my concerns below.

The biofuel-based electricity automatically cancels an equal amount of coal use. But should it not cancel an equal amount of coal-based electricity. Hence, the carbon price, κ , should be divided by the efficiency η in

Eq. (1). Am I utterly confused here? (I suppose, it would be easier to operate with biofuel feedstocks than the generated electricity. However, it is a matter of taste.)

Why some of the indices in Eqs. (4) and (5) are in parenthesis as arguments and not subscripts as the other. It may confuse someone to think that there are non-constant unit costs and unit emissions in the model.

If co-firing share is fixed to, say, 90%, what is the decision in the end. If the total electricity generation Z is fixed in Eq. (2), and the co-firing share in (6), the different levels of biofuel use should break either of these relations. Of course, the constraint (6) holds only when capacity is in full use and then 90% share holds. But when the capacity is not in full use, the power generation efficiency (26.6%) is based on 90% share but, in fact, it should be higher. Have I misunderstood something?

In addition, I suppose, there is a problem with constraints (7A-C), since on the left-hand side there is a sum over h but in the right-hand side there is not. I suppose, there should not be summation on the left-hand side. Or perhaps there should only be one constraint, namely,

$$\sum_{x=1}^3 \sum_{y=1}^3 T_{ixy}^h \leq B_i^h.$$

I do not understand constraints in Eqs. (8A-C). Why the emission from coal combustion are not captured when the transport capacity allows it? This can be seen as a major flaw in the modeling. This raises a serious question, how is the investment decision modelled? It seems that it is linked to biofuel use through unit costs. So, do the CCS unit costs depend on the biofuel use (and the utilization rate)?

Note: B_i^h 's and S_i 's are not defined in Eqs. (7A-C) and (8A-C).

Comparison between fuel capacities and electricity capacities: The paper utilizes the current level of electricity generation by the power plants (Supplementary spreadsheet). With CCS and co-firing, the plant's efficiency is lowered. In the model the plants can still produce the same amount of electricity and just needs more feedstock. Instead, it would be possible to think that the plants can digest the same amount of fuel (in Joules) but the electricity generation is lowered due to co-firing (and CCS). This can have a notable effect on outcomes as the electricity production capacity is lower in the latter approach. I suppose, the latter case is closer to the reality as the capacity of the boiler (i.e., the feedstock/thermal capacity) limits the scale of production and CCS and co-firing just make the steam going to the turbines less potent, reducing the efficiency of electricity generation.

Should not we allow for utilization of hydro and nuclear power generation also. They seem to be (Fig. 2) cheaper yet capacity constrained options.

2. Presentation

Abstract is confusing and hastily written. The language should be checked.

The manuscript relies quite heavily on Supplementary material. There are a lot of references to Supplementary Tables of Figures. So, reading the paper without the supplement is difficult. And in many cases, the Supplementary material does not support the text conveniently or the connection is a bit unclear (e.g. Table S1 and Figure S2).

Figure 2 is difficult! Before it can be presented, you should clearly state all the cases/scenarios you consider. Only 90% co-firing share of biomass is mentioned (e.g., on row 93), but the reader should know that you consider also 30% co-firing shares (with and without dedicated energy crops) and IGCC-retrofitting too. (The idea behind these additional scenarios should be discussed somewhere.)

Retrofitting is central in your paper, but it is not really explained what is meant by that. I suppose, there are separate decision to retrofit biomass feeding and CCS. What does biomass retrofitting mean: biomass handling, storages, pretreatment etc.? IGCC approach seems to be a bigger task than just retrofitting the pulverized fuel digestion. The CCS has separate aspects. These could be explained shortly.

The paper examines an all-in approach, where biomass co-firing and CCS (i.e., BECCS) are jointly introduced. It would be interesting to see, what would happen if (i) biomass co-firing without CCS and (ii) not using biomass but implementing CCS on coal-fired power plants were allowed too. I suppose, it could lower the marginal costs, if these decisions could be done separately. Linear program could optimize these options.

As a minor thing, in many places, the word "biofuel" could be replaced with "biomass".

3. Previous assessments

The paper would benefit from proper comparison and discussion of the results in the light of a largish literature on bioenergy potential of China. A quick search provided several related studies (see references below), which try to assess the biomass potential. The reliability of the current results would be improved, if they were contrasted with previous work.

In addition, with an omitted discussion, I cannot be certain, whether the spatial approach is such novelty after all. (There are spatial analyses in the papers listed in the references.)

Similarly, a discussion on marginal costs of emission abatement is lacking. Are these costs high or low compared to other studies or other emission abatement solutions?

4. BECCS

Paper would benefit from a clear exposition of BECCS's effects. How much efficiency of power generation is lowered and what are the resulting investment costs (biomass use increases marginal costs, but the fixed costs need to be considered too.) How much does the electricity price increase? (I do not know how electricity price is determined in China.)

Comparison without BECCS would be very useful! Scenarios with separate biomass retrofitting and CCS implementation and after that the full BECCS scenario. Or if the decisions to use biomass and CCS could be made separately, the linear program would find an optimal mix of solutions.

The co-firing shares of biomass are very high (up to 90%). Is this feasible? What kind of preprocessing is needed? Pelletization, torrefication, etc.? I suppose, biomass has notably lower energy density than coal. Therefore, an ordinary pulverized fuel burner cannot be fed fuel at its optimal/maximal rate with biomass in energy terms. Also, the amount of volatiles and moisture in biomass differs from coal, which distorts the optimized combustion process of a pulverized coal boiler. Hence, a heavy preprocessing of biomass can be needed. I suppose, this is taken into account at some level by low efficiency coefficient (26.6%) and preprocessing costs? They could be discussed more explicitly.

Why is the biomass share so low (30%) in IGCC plants? Should not it be easier to get high biomass shares with IGCC than with pulverized combustion?

5. Data set

The biomass data set is impressive and to my understanding the allocation of provincial figures between counties seems reasonable. However, I have some reservations about technological data.

The calculation of unit emission factors is not explicit. Notes of Table S4 tell that these are functions of specific parameters. The functions should be explicitly stated.

Why do not you use China data for power plants? It seems that most of the values are case studies (e.g. Fajardy & MacDowell 2017). I suppose, China has a fleet of coal power plants with variety of efficiencies. This is something that affects the results notably and should be discussed, at least.

Investment costs? In Black & Veatch the numbers seem to be higher than yours. Is this just because of inflation or didn't I find correct numbers? There is no investment decision in your optimization problem. How is the changing utilization rate taken into account in the unit costs?

On row 104 you apply 90% efficiency for CCS but on row 520 you use 40%. Which is a correct number?

6. The scale of the operation

To me the BECCS operation seems to be a massive national project. I was wondering what the province level economic effects would be. How much labor would be needed? For example, the labor costs of energy crops are five times those of agricultural crops (Table S5). Does this mean that five times more labor is needed?

Smaller comments

R75: List could include costs (fuel costs and reduced efficiency of power plants due to CCS and co-firing).

R78: I suppose, the link between Table S1 and the preceding text could be made clearer.

R124: This 0.9 Gt/yr is little surprising after the fact that only 0.36 Gt/yr can be combusted and storage in a same county. Reader could be helped by mentioning that now you do not consider the CO₂ transportation limit.

Figure 1, panel D: Storage for 30 years? Is it then released? Why not longer timescales? Based on Figure S7, this should not make such a difference, but 30 years looks quite bad from the climate point of view.

R143: I do not understand the reference to carbon price.

R 146: "... abatement could be offset by ..." I do not understand. Do they offset? What is the idea here?

R151: Perhaps, the CCS does not "dominate", when the marginal biofuel acquisition and pretreatment costs are almost the same...

R151 onwards: These marginal cost components sum up over one hundred that was mentioned earlier in the paragraph. You lost me here.

R155: "To reduce the marginal costs while making the compromise to abate less CO₂ emissions, it is possible to focus on biofuel with lower production costs, and on shorter distances for the transport of biofuel and captured CO₂." To me the causality goes to other direction: By focusing on biofuel with lower production costs and on shorter distances for the transport of biofuel and captured CO₂, the marginal costs but also the CO₂ emission abatement are reduced.

R158: I am sorry to say that the example does not provide a "clear illustration". Economic efficiency is unclear for me here. Yes, there are increasing marginal costs related to BECCS. Maybe that is what you try to convey here? (Additional note: I would like to see Fig. 3S in the main text.)

R167: This 30% possibility comes from out of the blue. It should be mentioned in the introductory paragraphs. Also, you could explain the reader why the efficiency depends on the co-fire share of biomass.

R168: Can you really retrofit a coal power plant into IGCC or do you have to build a completely new power plant?

R170: “The marginal costs are less sensitive to the capacity of electricity generation and the yield of energy crops when the target is to abate less than 3 Gt CO₂-eq yr⁻¹.” I cannot understand the reference here. The previous sentences are disconnected (0.88 Gt abatement).

Figure 2: What determines the limit for nuclear energy capacity?

R203: Consider changing “water” to “hydro”.

R207: 2.3 is 19% of 12.2 not 30%. Please, correct something.

R212: Are these payments made once or are they annual payments until 2030? How much will the WWSN-project cost in terms of GDP? Combined capacity of hydro and nuclear is quite high (Fig. 2). What would be the cost to abate 2 Gt CO₂eq /yr with those two technologies? That is, why you consider only BECCS and not the hydro and nuclear too.

R221 “we want to keep” Do you mean that you keep in the calculation or the World is expecting that China’s power sector keeps, or that you as researcher want...

R247: “retrofitting coal plants to co-fire with biofuel” and CCS? Or is this separate biomass case without CCS. I suppose not, as on row 250 CO₂ is stored.

R266: “the latter is missing in most global integrated assessment models” I fear that this topic is missing also in this study, except for couple of citations to previous literature...

Eqs (3A-C): Consider removing unnecessary parentheses.

R514: How are the CO₂ transport capacities (89 and 26698 kt yr⁻¹) calculated?

Eqs 9A-C and row 529: Where do these equations come from? What is 1660? Capital Fs should be χ s? Why the exponents are 0.65 and 1.13? What kind of values does the distances L get?

Comments on Supplement

Figure S1: The Figure is rather difficult to read. There are two T pillars and the middle pillar are continuing beyond the cut-off, the level of which is not clear to me.

Figure S2: How can the “BECCS electricity generation” be greater than the “current fossil electricity generation”? Is there new capacity or is the capacity utilization rate very low currently? Or do you mean “biofuel-based generation potential” vs. electricity generation capacity? Then I suggest rephrasing the terms. (Note: It would be easier to compare the “amount of bioenergy” to the “maximum fuel capacity of power plants”.)

Figure S7: Difficult to read. What is going on? Time development is unnoticeable. Title of the vertical axis is confusing: How can carbon sequestration happen across counties?

Figure S8: Panel A shows the “monthly minimum temperature”. Should not there be 12 values for each county? Is this the minimum of monthly minimum temperatures or a minimum monthly temperature in a specific month? Unit must have some error too, or is “monthly minimum temperature” an annual mean of these minimum temperatures? In panels E and F, the measurement of biofuel/biomass in tons of CO₂ is little confusing. At least it needs clarification that you want to operate with “carbon sink” units.

Figure S9: The text is slightly confusing. Do you mean that national pipeline has diameter of 76.2 cm whereas county pipeline has diameter of 7.62 cm? Where does the “a function of CO₂ flow rate” refer to? Title of Sanchez et al. (2018) implies that their data is from US. Is it applicable to China too?

Figure S10: Panels C & F: Is this cost to neighboring province or is there one cost for transports between all the provinces? Typically, transport costs between “n” different areas implies “(n-1) x (n-1)” transport unit costs...

References

- Chen, X. (2016). Economic potential of biomass supply from crop residues in China. *Applied Energy*, 166, 141-149.
- Jingjing, L., Xing, Z., DeLaquil, P., & Larson, E. D. (2001). Biomass energy in China and its potential. *Energy for Sustainable Development*, 5(4), 66-80.
- Junfeng, L., & Runqing, H. (2003). Sustainable biomass production for energy in China. *Biomass and Bioenergy*, 25(5), 483-499.
- Junfeng, L., Runqing, H., Yanqin, S., Jingli, S., Bhattacharya, S. C., & Salam, P. A. (2005). Assessment of sustainable energy potential of non-plantation biomass resources in China. *Biomass and Bioenergy*, 29(3), 167-177.
- Liu, H., Jiang, G. M., Zhuang, H. Y., & Wang, K. J. (2008). Distribution, utilization structure and potential of biomass resources in rural China: with special references of crop residues. *Renewable and Sustainable Energy Reviews*, 12(5), 1402-1418.
- Nie, Y., Chang, S., Cai, W., Wang, C., Fu, J., Hui, J., ... & Guo, W. (2020). Spatial distribution of usable biomass feedstock and technical bioenergy potential in China. *GCB Bioenergy*, 12(1), 54-70.
- Qin, Z., Zhuang, Q., Cai, X., He, Y., Huang, Y., Jiang, D., ... & Wang, M. Q. (2018). Biomass and biofuels in China: Toward bioenergy resource potentials and their impacts on the environment. *Renewable and Sustainable Energy Reviews*, 82, 2387-2400.
- Shi, Y., Ge, Y., Chang, J., Shao, H., & Tang, Y. (2013). Garden waste biomass for renewable and sustainable energy production in China: Potential, challenges and development. *Renewable and Sustainable Energy Reviews*, 22, 432-437.
- Yanli, Y., Peidong, Z., Wenlong, Z., Yongsheng, T., Yonghong, Z., & Lisheng, W. (2010). Quantitative appraisal and potential analysis for primary biomass resources for energy utilization in China. *Renewable and Sustainable Energy Reviews*, 14(9), 3050-3058.
- Zhao, G. (2018). Assessment of potential biomass energy production in China towards 2030 and 2050. *International Journal of Sustainable Energy*, 37(1), 47-66.
- Zhou, X., Wang, F., Hu, H., Yang, L., Guo, P., & Xiao, B. (2011). Assessment of sustainable biomass resource for energy use in China. *Biomass and Bioenergy*, 35(1), 1-11.

Referee #1**Comment A1**

This paper is an absolute treasure trove of valuable data, and presents strong and robust analysis, therefore I strongly support its publication, essentially as is.

Response

Thank you for the positive comment. Our motivation is to provide basic data to help decision makers determine the potential contribution of BECCS to decarbonize the power system in a near term, and present a conceptual method for how these data can be used to develop an optimal model by retrofitting existing power plants for biomass utilization and CCS together. We hope that our paper can motivate more studies to estimate the cost curves of similar technological options in China, which are required to re-visit the nationally determined contributions to the Paris Agreement. We have improved the manuscript and justified the methodology. Please refer to a track version for detailed changes.

Comment A2

If possible, I would like to see the graphics in the main paper enlarged so as to be easier to read.

Response

We thank Reviewer 1 for this good suggestion. All graphics in our paper have been enlarged to enhance the readability.

Comment A3

And I'd also like to draw the attention of the authors to the work of Singh et al, "Large-Scale Affordable CO₂ Capture Is Possible by 2030", published in Joule, 3(9), 2154-2164, 2019 - the approach is somewhat similar.

Response

We thank Reviewer 1 for bringing this literature to our notice. Singh et al. (2019) have estimated the cost curve of CO₂ capture in China, while the limitation by uncertainties in the transport and storage systems is acknowledged:

Conclusions

The cost of CO₂ capture across power plants in China has been shown to have a wide distribution due to a combination of technical and market factors. In China, several avenues in the existing policy and market regime can be applied reduce the avoided cost of CO₂ to as low as \$25/tCO₂. Extrapolation of our results from our case study fleet of 25 power plants across the entire Chinese power sector suggests that significant volumes of CO₂ could be captured using CCS retrofits at coal-fired power plants for costs on par with other options for low-carbon energy. The actual rate of deployment of CCS in China will likely be limited by maturation of transport and storage capabilities rather than the cost of capture and could coincide with other developments in the transformation of the energy system such as renewables deployment. Globally, this approach could be used to identify lower cost opportunities to accelerate CCS deployment in other countries but could have the most impact through its contribution to maturation of the CCS supply chain in China.

Reprinted from Joule, VOLUME 3, ISSUE 9, Singh et al, Large-Scale Affordable CO₂ Capture Is Possible by 2030, p2154-2164., Copyright 2019, with permission from Elsevier

To highlight the importance of modeling the transport and storage systems, we added the following sentence on lines 92-96 as: “*A recent study indicates that the cost of CO₂ capture is affordable for China’s power plants²¹, but up to now, the costs of biomass transport and CO₂ storage had not been assessed.*”.

Reference:

Singh, S. P. et al. Large-Scale Affordable CO₂ Capture Is Possible by 2030. *Joule* **3**, 2154-2164 (2019).

Referee #2

Comment B1

Summary The paper presents a linear program where co-firing of biomass with coal in existing power plant fleet is optimized, when the CO₂ emissions of the combusted biomass is captured and stored. The model applies county level data from China. The results suggest quite promising BECCS scenarios for China.

Response

Thank you for the positive comments.

Comment B2

General impression: The paper presents a set of important research questions that contribute to the solution of the global climate change problem. Unfortunately, for me, the manuscript was quite laborious to read. The results offered a lot of numbers, which I had difficulty to put in perspective. It was not clear, where does these number come from and what are the underlying assumption.

Response

We appreciate Reviewer 2 for all good suggestions, which are helpful to improve the manuscript. According to the comments, we have revised our paper by 1) presenting all numbers more clearly in our paper to improve the readability (please see our response to **Comments B17, B24, B31, B32, B33, B45, B55, B56, B61, B62**, a new **Table 1** added to the main text, and new **Tables S2-S6** added to the Supporting Information); 2) comparing our estimate of China’s bioenergy potential with previous studies (see a new **Fig 1** added to the main text and our response to **Comment B21**); and 3) clearly explaining the sources of data, clarifying the underlying assumptions, and clarifying the limitations (see our response to **Comments B9, B17, B32, B56, B61, B62** and new **Tables S3-S5** added to the Supporting Information). Please refer to a point-by-point response to all comments below.

Comment B3

Hence, for a person like me, the reading of results and numbers is meaningless, before I can see the methodology behind the numbers. So, I focused on the Methods section and the Supplement. While the data set was impressive, there were also issues that I raise below. The most worrisome aspect from my perspective was the contents (and the presentation) of the linear programming model. I raise these issues below. All in all, if my worries on the data and the model prove unwarranted, there is a feeling that the results are quite optimistic. Is it realistic to assume that it is possible to extract all the biomass with a constant unit cost from

the whole county? If not, the modeling approach is in jeopardy. If it is OK, then the paper a promising future.

Response

Thank you for the insightful comment. We do not assume that “*it is possible to extract all the biomass with a constant unit cost from the whole county*”. As a major novelty in our present paper, we consider that the unit cost of utilizing biomass for BECCS depends on the target of national emission abatement using our optimization model: 1) a low target allows us to utilize the cheapest biomass (agricultural residues) at a low cost of biomass/CO₂ transport in a county; and 2) a higher target requires us to utilize more expensive biomass (dedicated energy crops) at a larger cost of biomass/CO₂ transport from a county in a province to a county in another province. For the first time, we present the marginal cost curves for BECCS. We realized that the formulation of our optimization model is not correctly given in the previously submitted version of manuscript. Please see a revision to the formulation, equations 1-4, and the **Method** section in our responses to **Comment B6**, and a point-by-point response to all technical comments below.

Comment B4

In addition, there were severe issues in overall presentation. I mention some issues in my “Comments and suggestions” and in “Smaller comments”.

Response

We have improved the presentation of the manuscript (see our response to **Comments B9, B15, B16, B17, B23, B29** and the language is polished by a native speaker) and the link between Supporting materials and the main text (see our response to **Comment B16**). Please refer to a point-by-point response to all comments of Reviewer 2 below.

Comment B5

Given the high requirements of Nature Communications, I fear that the manuscript will not have enough scientific impact to warrant publication in this journal. Yet, the manuscript as a high reader potential given the nature of the data and importance of the research question. Before that, however, the manuscript should be polished, and the modeling revised.

Response

Thank you for the insightful comment. We have revised the model (please see a new Fig 3 and our response to **Comments B6, B17, B19, B31**), compared our estimate of China’s bioenergy potential with previous estimates (see a new Fig 1 added to the main text and our response to **Comment B21**), evaluated the effects of BECCS relative to options of biomass utilization or CCS alone, justified the data sources (please see new **Tables S2-S4** and our response to **Comments B9, B29, B32, B56, B61, B62**), and improved the presentation (please see our response to **Comments B15, B16, B23, B29**). We hope that Reviewer 2 will find that our revised manuscript can meet the requirements of Nature Communications.

Comment B6

Optimization problem: Optimization problem (Eq. (1)) is a little bit confusing. The results of the manuscript suggest that the authors solve a cost minimization problem with a given emission reduction target, but instead the formulation is a profit maximization problem with a given carbon emission price. However, row 427 suggests that there is an emission reduction target together with a given carbon price. In addition, Eq (1) suggests that decision variable was E , which seems misleading as the real decision variables should, to my understanding, be T_{ixy}^h .

Response

Thank you for pointing out this problem in the formulation of our model. We realized that, while our model and calculations are correct, the formulation is not correctly described in the previous manuscript. The decision variable is T_{ixyh} , in line with the suggestion of Reviewer 2.

In this model, we have followed two steps: (1) the consumption (T_{ixyh}) of 21 types of biomass (h) in power plants co-fired with coal in 2836 counties (i) at three levels of biomass transport (x) and three levels of CO₂ transport (y) is optimized to minimize the total cost (C) to meet a national target of emission reduction (F), and (2) the carbon price per ton of net carbon emissions reduced (ζ) is calculated as $\zeta = dC/dF$, where dC is the increase in total cost (C) when the target of emission reduction increases from F to $(F+dF)$. To clarify it,

we revised Eqs. 1-2 into a cost-minimization problem (please see equations 1,2). We hope that it helps to understand the relationship between the decision variable and the corresponding carbon price.

Calculations behind the results of the previous manuscript were done according to equations 1,2, thus there is no change in our results due to this correction.

According to these procedures, the following paragraphs on lines 760-781 in the **Methods** section were revised as: “To achieve a target of national emission mitigation without changing total electricity generation, we estimated the total costs of BECCS deployment accounting for the saving made from the substitution of coal. For power plants in a given county i , we considered biomass grown in the same county i and transported within the county (case $x=1$), biomass grown in other counties of the same province and transported to county i (case $x=2$) and biomass grown and transported from counties in other provinces (case $x=3$). We then considered CO_2 captured from power plants in county i and transported to storage site in county i (case $y=1$), CO_2 captured from power plants in county i and transported to storage site in other counties of the same province (case $y=2$), and CO_2 captured from power plants in county i and transported to storage site in other provinces (case $y=3$). We followed a cost-minimization approach to optimize the consumption of biomass for co-firing with coal in 2836 counties (T_{ixyh}) to achieve a given target of national emission reduction (F) as:

$$\min_{T_{ixyh}} C = \sum_{i=1}^{2836} \sum_{h=1}^{21} \sum_{x=1}^3 \sum_{y=1}^3 \mu_{ixyh} T_{ixyh} - \frac{\kappa E}{v\lambda_c}, \quad \forall \frac{\phi E}{v\lambda_c} - N = F \quad (1A)$$

$$E = \eta \sum_{i=1}^{2836} \sum_{h=1}^{21} \sum_{x=1}^3 \sum_{y=1}^3 \lambda_h T_{ixyh} \quad (1B)$$

$$N = \sum_{i=1}^{2836} \sum_{h=1}^{21} \sum_{x=1}^3 \sum_{y=1}^3 r_{ixyh} T_{ixyh} \quad (1C)$$

where C is total costs; h is the type of biomass; F is the target of net emission reduction; E is electricity generation of coal that is replaced by biomass; N is the lifecycle emissions in deploying BECCS; T_{ixyh} is consumption of biomass; μ_{ixyh} and r_{ixyh} are unit cost and emission in utilization of biomass h ; v and η denote the power generation efficiency of coal and co-firing of biomass with coal, respectively; κ is the price of coal; λ_c and λ_h is heat content of coal and biomass h , respectively; and ϕ is the CO_2 emission factor of coal.

Then, the carbon price per ton carbon reduced (ξ), equivalent to the marginal cost of BECCS as shown in Fig. 3, is calculated as:

$$\xi = \frac{dC}{dF} \quad (2)$$

where dC is the increase in total costs when the target of national emission reduction increases from F to $(F+dF)$.”

Comment B7

A major issue is the formulation. I list my concerns below.

The biofuel-based electricity automatically cancels an equal amount of coal use. But should it not cancel an equal amount of coal-based electricity. Hence, the carbon price, κ , should be divided by the efficiency η in Eq. (1). Am I utterly confused here? (I suppose, it would be easier to operate with biofuel feedstocks than the generated electricity. However, it is a matter of taste.)

Response

Thank you for pointing out this problem in the model description: the coal price (κ) has been divided by the power generation efficiency of coal-fired power plants (v) and heat content of coal (λ_c) in our model. Please see a new equation 1 in our response to **Comment B6**. Calculations behind the results of the previous manuscript were done according to equation 1, thus there is no changes in our results due to this correction.

Comment B8

Why some of the indices in Eqs. (4) and (5) are in parenthesis as arguments and not subscripts as the other. It may confuse someone to think that there are non-constant unit costs and unit emissions in the model.

Response

Thank you for the comment, and the indices in the equations 3,4 (corresponding to Eqs. 4,5 in the last version) are now written as subscripts, rather than in parenthesis. The new equations 3,4 reads as:

$$\mu_{ixyh} = \mu_h^a + \mu_h^f + \mu_{ix}^b + \mu_{iy}^t + \mu_h^p + \mu_h^s + \mu_h^w + \mu_h^d + \mu_h^{cap} \quad (3)$$

$$r_{ixyh} = r_h^p + r_h^{zp} + r_h^{za} + r_{ix}^b + r_h^d + r_h^{uc} - r_h^s \quad (4)$$

The following sentence was added on lines 797-798 in the track version “*the subscript denotes the independent variable, and the superscript denotes the type of costs*”.

The following sentence was added on lines 817-818 in the track version “*the subscript denotes the independent variable, and the superscript denotes the category of emissions*”.

Comment B9

If co-firing share is fixed to, say, 90%, what is the decision in the end. If the total electricity generation Z is fixed in Eq. (2), and the co-firing share in (6), the different levels of biofuel use should break either of these relations. Of course, the constraint (6) holds only when capacity is in full use and then 90% share holds. But when the capacity is not in full use, the power generation efficiency (26.6%) is based on 90% share but, in fact, it should be higher. Have I misunderstood something?

Response

In the new equation 5 (Eq. 6 in the last version), the electricity generation is constrained by co-firing biomass and coal together not exceeding the current electricity generation by power plants in each county. The total electricity generation and the co-firing ratio (90%) are both fixed, while the power generation efficiency is fixed in coal-fired plants (39.3%) and biomass-coal-cofiring plants (25.1%). We have re-claimed our decision variable to be the consumption of biomass in 2836 counties (T_{ixyh}) and revised the optimization problem to a cost-minimization approach (please see our response to Comment B6).

The power generation efficiency depends on the fraction of coal plants that are shifted to biomass co-firing, which is determined by our decision variable (T_{ixyh}). When a fraction of coal-fired plants is retrofitted to biomass-coal-cofiring plants, the average power generation efficiency decreases with an expansion of the boiler capacity to consume more fuels (Joule) in all China’s power plants. In our model, we have considered the cost of consuming more fuels and the cost of retrofitting power plants (please see our new equation 3).

To clarify it, we added the following sentences on lines 173-182 in the track version as: “*We assume that BECCS plants produce the same amount of electricity to avoid the impact of reduced electricity generation on the economy. However, the power generation efficiency declines as a result of biomass co-firing, since a high moisture and a fibrous structure of biomass reduce the stability of combustion flame⁴⁴, while it consumes more energy to break the oxygenated chemical bonds in the gasification of biomass¹¹. This requires to increase the capacity of loading more fuels into power plants by increasing the rated evaporation, boiler thermal load and fuel flow rate^{11,45,46}. These operational conditions are included in the costs of retrofitting power plants in our modelling (see the costs in Table 1).*”.

To make it clearer that biomass co-firing and CCS will reduce the efficiency of power generation and increase the cost of electricity generation, the following sentence was added on lines 384-388: “*For example, to abate 1, 3 and 5 Gt CO₂-eq yr⁻¹ by BECCS in China, the power generation efficiency is reduced from 39.3% (coal-fired plants) to 25.1% (90% biomass co-firing) to generate 0.55, 1.74 and 3.33 PWh yr⁻¹ of electricity in BECCS, which will increase the electricity price from \$0.060 kWh⁻¹ to \$0.074, 0.125 and 0.248 kWh⁻¹, respectively.*”.

Comment B10

In addition, I suppose, there is a problem with constraints (7A-C), since on the left-hand side there is a sum over h but in the right-hand side there is not. I suppose, there should not be summation on the left-hand side. Or perhaps there should only be one constraint, namely,

$$\sum_{h=1}^{21} \sum_{y=1}^3 T_{i1y}^h \leq B_i^h$$

Response

Thank you for pointing out this mistake in the equation. We have double-checked all equations in our paper. The equation 6 (Eq. 7 in the last version) was corrected as: “*For county i in province j , we considered m_j and (m_j+n_j) as the first and last county in this province j ($j=1$ to 31 for 31 provinces in China). Thus, the constraints on the supply of biomass grown in the same county i and transported within the county (case $x=1$), biomass grown in other counties of the same province and transported to county i (case $x=2$) and biomass grown and transported from counties in other provinces (case $x=3$) are respectively: ”*

$$\sum_{y=1}^3 T_{ixyh} \leq B_{ih}, \quad i=1 \text{ to } 2836, x=1 \text{ and } h=1 \text{ to } 21 \quad (6A)$$

$$\sum_{i=m_j}^{m_j+n_j} \sum_{x=1}^2 \sum_{y=1}^3 T_{ixyh} \leq \sum_{i=m_j}^{m_j+n_j} B_{ih}, \quad j=1 \text{ to } 31 \text{ and } h=1 \text{ to } 21 \quad (6B)$$

$$\sum_{i=1}^{2836} \sum_{x=1}^3 \sum_{y=1}^3 T_{ixyh} \leq \sum_{i=1}^{2836} B_{ih}, \quad h=1 \text{ to } 21 \quad (6C)$$

Comment B11

I do not understand constraints in Eqs. (8A-C). Why the emission from coal combustion are not captured when the transport capacity allows it? This can be seen as a major flaw in the modeling. This raises a serious question, how is the investment decision modelled? It seems that it is linked to biofuel use through unit costs. So, do the CCS unit costs depend on the biofuel use (and the utilization rate)?

Response

Thank you for the good suggestion. The equation 7 (Eq. 8 in the last version) is constrained by the carbon capture from BECCS in county i , province j and the country not exceeding the capacity of carbon storage in county i , province j and the country, respectively. When the transport capacity allows it, the capture of CO₂ from coal combustion is additional to the contribution from BECCS. The abated emissions and costs by capturing CO₂ from coal emissions are not considered as a contribution of BECCS to emission reduction in China. However, we have followed Comment B19 by considering a scenario of implementing CCS to coal-fired plants (please refer to our response to Comment B19).

Therefore, we believe that this is not a major flaw in our modeling approach.

For the decision variable, we have re-claimed it to be the consumption of biomass in 2836 counties (please see our response to Comment B6).

For CCS, unit cost (μ_h^{cap} , that is $\$82.5$ (t biomass)⁻¹ for agricultural residues and energy crops, and $\$82.3$ (t biomass)⁻¹ for wood products) is calculated as a product of the cost to capture one ton CO₂ (PR_CCS) and the amount of CO₂ emitted from one ton of biomass ($CC_h \cdot EC \cdot 3.67$), which is explained in a new Table S2 added to the Supporting Information:

		Unit cost of CCS in power plants is calculated as: [⚡]
		$\mu_h^{cap} = PR_CCS \cdot CC_h \cdot EC \cdot 3.67^{\leftarrow}$
CO ₂ capture and storage (μ_h^{cap}) [⚡]	82.5 (±10%, agricultural residues and energy crops) [⚡]	where h is the type of biomass; PR_CCS is the price of CO ₂ capture and storage; CC_h is carbon content in biomass; EC is the efficiency of CO ₂ capture; 3.67 is the parameter convert C to CO ₂ . [⚡]
	82.3 (±10%, wood products) [⚡]	For PR_CCS , we adopt a value of $\$53 \pm 7$ (t CO ₂) ⁻¹ (Koornneef et al., 2012 ; Rubin et al., 2015). [⚡]
		For CC_h , we adopt values of 47.1% for agricultural residues and energy crops (Zhang et al., 2015 ; Fajardy & Mac Dowell, 2017), and 47% for wood products (IPCC, 2006).
		For EC , we adopt a constant value of 90% (Anderson & Peters, 2016 ; Lu et al., 2019).

Comment B12

Note: B_{hi} and S_i are not defined in Eqs. (7A-C) and (8A-C).

Response

In our previous manuscript, B_{ih} and S_i were defined at the beginning of this section. In the new version, we now define B_{ih} on line 850: “*where B_{ih} is feedstock of biomass h in county i .*”, and define S_i on line 856: “*where S_i is capacity of carbon storage in county i .*”.

Comment B13

Comparison between fuel capacities and electricity capacities: The paper utilizes the current level of electricity generation by the power plants (Supplementary spreadsheet). With CCS and co-firing, the plant’s efficiency is lowered. In the model the plants can still produce the same amount of electricity and just needs more feedstock. Instead, it would be possible to think that the plants can digest the same amount of fuel (in Joules) but the electricity generation is lowered due to co-firing (and CCS). This can have a notable effect on outcomes as the electricity production capacity is lower in the latter approach. I suppose, the latter case is closer to the reality as the capacity of the boiler (i.e., the feedstock/thermal capacity) limits the scale of production and CCS and co-firing just make the steam going to the turbines less potent, reducing the efficiency of electricity generation.

Response

Thank you for the comment. We do not assume that the plants are digesting the same amount of fuel (in Joules), because the reduced electricity generation (in PWh) will produce a damage to the economy or require more renewable energy to retain the electricity production. Then, the costs and potential of emission reduction cannot be fully attributed to BECCS, and this does not meet the purpose of our study to estimate the potential and costs of BECCS. Instead, we have estimated the potential and costs of BECCS when the power system can produce the same electricity (in PWh) as that in a case without BECCS. After a careful consideration, we wish to retain to our present approach by considering that the same amount of electricity (in PWh) is generated in China’s power plants.

Following the suggestion of Reviewer 2, we have examined the requirements to increase the capacity of boilers (Lu et al., 2019; Liu et al., 2019; Shi et al., 2019). Accordingly, we added the following sentences on lines 173-182 in the track version as: “*We assume that BECCS plants produce the same amount of electricity to avoid the impact of reduced electricity generation on the economy. However, the power generation efficiency declines as a result of biomass co-firing, since a high moisture and a fibrous structure of biomass reduce the stability of combustion flame⁴⁴, while it consumes more energy to break the oxygenated chemical bonds in the gasification of biomass¹¹. This requires to increase the capacity of loading more fuels into power plants by increasing the rated evaporation, boiler thermal load and fuel flow rate^{11,45,46}. These operational conditions are included in the costs of retrofitting power plants in our modelling (see the costs in **Table 1**).*”.

To clarify the effects of co-firing biomass and implementing CCS on reducing the efficiency of power generation and increasing the cost of retrofitting power plants, we added the following sentence on lines 384-388 as: “*For example, to abate 1, 3 and 5 Gt CO_{2-eq} yr⁻¹ by BECCS in China, the power generation efficiency is reduced from 39.3% (coal-fired plants) to 25.1% (90% biomass co-firing) to generate 0.55, 1.74 and 3.33 PWh yr⁻¹ of electricity in BECCS, which will increase the electricity price from \$0.060 kWh⁻¹ to \$0.074, 0.125 and 0.248 kWh⁻¹, respectively.*”.

References:

Lu, X. et al. Gasification of coal and biomass as a net carbon-negative power source for environment-friendly electricity generation in China. *Proc. Natl. Acad. Sci. U.S.A.* **116**, 8206-8213 (2019).

Liu, Z., Ma, S., Pan, X. & Chen, J. Experimental study on the load response rate under the dynamic combined combustion of PC coal and CFB coal in a CFB boiler. *Fuel* **236**, 445-451 (2019).

Shi, Y. et al. Soot blowing optimization for frequency in economizers to improve boiler performance in coal-fired power plant. *Energies* **12**, 2901 (2019).

Comment B14

Should not we allow for utilization of hydro and nuclear power generation also. They seem to be (Fig. 2)

cheaper yet capacity constrained options.

Response

Thank you for the good suggestion.

We agree with Reviewer 2 that it is interesting to model the utilization of hydropower and nuclear power generation. However, their marginal cost curves are not yet available to us. The marginal cost of hydropower and nuclear power as shown in the new Fig. 3 (Fig. 2 in the last version) are calculated from the prices of hydropower and nuclear power in China (NEA, 2018) and the emission factor of coal. We now adopt the projected capacity from the plan of the Chinese government (CERS, 2016), rather than modelling the competition of water (i.e., hydropower), wind, solar and nuclear (WWSN) power to BECCS (see results in the new Fig 6). This allows us to focus on discussing on the costs and potential of deploying BECCS, and qualitatively show the trade-off between expanding WWSN and investing in BECCS.

To make it clearer that the marginal cost curves for hydropower and nuclear power are not available, the following sentences were added on lines 965-973 in the main text as: “*Note that the marginal cost of WWSN as shown in Fig. 3 are estimated based on the prices of hydropower, wind, solar, and nuclear power⁵² in China in 2016. Their marginal cost curves are yet not available for China. Therefore, we adopted the projected capacity from the plan of the Chinese government⁴², rather than modelling the competition of WWSN to BECCS in our optimization model. This allows us to focus on analyzing the potential and costs of deploying BECCS to decarbonize the electricity system, and qualitatively show a trade-off between WWSN and BECCS. However, more studies are needed to estimate the marginal cost curves for hydropower, wind, solar, and nuclear power, which can better determine the contribution of BECCS relative to WWSN in China.*”.

To make it clearer how we obtain the marginal costs of for hydropower and nuclear power in 2015, we added the following sentences in the caption of Fig. 3 as: “*The national CO₂ emission reduction (calculated from the substituted coal consumption⁴² and the CO₂ emission factor of coal⁵³) by hydropower, wind, solar and nuclear power in China in 2015 as well as the range of their marginal costs (calculated from their prices of electricity generation in different provinces⁵² and the CO₂ emission factor of coal⁵³) are shown as vertical lines.*”.

References:

China Energy Research Society (CERS). *China Energy Outlook 2030* (Energy & Management Press, Beijing, 2016).

National Energy Administration of the People’s Republic of China (NEA). *Regulatory Bulletin on the State Power Price in 2017*. http://www.nea.gov.cn/137519800_15391333051221n.pdf (2018).

Comment B15

2. Presentation

Abstract is confusing and hastily written. The language should be checked.

Response

Thank you for the comment. We have polished the language in Abstract and the whole manuscript by two native speakers.

The revised **Abstract** reads as: “*As China ramped-up coal power capacities rapidly while CO₂ emissions need to decline, these capacities would turn into stranded assets. To deal with this risk, a promising option is to retrofit these capacities to co-fire with biomass and eventually upgrade to CCS operation (BECCS); yet, the feasibility is debated with respect to negative impacts on broader sustainability issues. We present a data-rich spatially explicit approach that allows to estimate marginal cost curves for decarbonizing China’s power system. We show that 222 GW of power capacities in 2836 counties can be generated by co-firing 0.9 Gt of biomass from the same county, with half being agricultural residues. Two gigatonnes of net CO₂ emissions annually can be abated by BECCS at a marginal cost of \$92⁺⁹³_{-.48} (t CO₂)⁻¹. Using spatially explicit data helps to identify the best opportunities, shows the limitations imposed by logistical challenges and spatial mismatches, and provides differentiated information for assessing emission reduction potentials.*”.

Comment B16

The manuscript relies quite heavily on Supplementary material. There are a lot of references to Supplementary Tables of Figures. So, reading the paper without the supplement is difficult. And in many cases, the Supplementary material does not support the text conveniently or the connection is a bit unclear (e.g. Table S1 and Figure S2).

Response

Thank you for the good suggestion.

Following the format of Nature Communications, we present the major method and key results in the main text, and provide detailed methods and data sources in the Supporting Information.

To improve the readability, we revised **Table S2,S3** (please refer to our response to **Comment B29**), **Table S4 (Comment B17)**, **Fig. S1 (Comment B57)**, **Fig. S2 (Comment B58)**, **Fig. S6 (Comment B59)**, and **Fig. S7 (Comment B60)**. We moved **Fig. S3** from Supporting Information to the main text as a new **Fig. 4 (Comment B43)**.

To improve the link between Supplementary Information and the main text, the sentence on lines 89-90 in the track version was revised as: “*Studies that have investigated the potential and marginal cost of BECCS at a regional scale are listed in **Table S1**.*”,

the sentence on lines 236-237 was revised as: “*see parameterization of unit costs and emissions in **Tables S2-S4***”,

the sentence on lines 815-816 was revised as: “*Methods to calculate all cost items and their data sources are given in **Table S2**.*”,

the sentence on lines 832-833 was revised as: “*Methods to calculate all emission items and their data sources are given in **Table S3**.*”

and the sentence on lines 984-986 was revised as: “*We adopted random values for these parameters from their uniform distributions (see their ranges given in **Tables S2,S3**) in the Monte Carlo simulations...*”

the sentence on lines 293-294 was revised as: “*To address the source of uncertainties, we design seven scenarios by varying parameters (see parameterization of each scenario in **Table S4**)*”,

the sentence on lines 423-424 was revised as: “*see the potential and costs of BECCS by province in **Table S5***”,

the sentence on line 432 was revised as: “*see **Table S6** for the fraction of acquisition and labor costs in total costs*”,

the sentence on line 691 was revised as: “*see all sources of statistic data in **Table S7***”,

the sentence on line 701 was revised as: “*.....biomass productivity (see all sources of spatial data in **Table S7**)*”,

the sentence on line 169 was revised as: “*The carbon budget in BECCS with their constraints is shown in **Fig. S1**.*”,

the sentence on lines 195-198 in the track version was revised as: “*The ratio of bioenergy potential to electricity generation capacity in existing power plants and the ratio of potential CO₂ captured from combustion of biomass to carbon storage capacity in proven repositories are mapped by county in **Fig. S2**.*”,

the sentence on lines 694-695 was revised as: “*As shown in **Fig. S3**, some wood products (commercial roundwood, bamboo,...)*”,

the sentence on lines 699-700 was revised as: “*(see their spatial distributions in **Fig. S4**), based on spatial distributions of socioeconomic parameters...*”,

the sentence on lines 718-719 was revised as: “*The distributions of carbon-storage capacity in different reservoirs are shown in **Fig. S5**.*”,

the sentence on lines 720-722 was revised as: “*the choice of this parameter slightly affects the constraints on carbon storage in counties where bioenergy is produced (see a sensitivity test in **Fig. S6**)*”,

the sentence on lines 744-746 was revised as: “*based on minimum of monthly average temperature, annual average precipitation rate and annual average active sunshine hours (see their spatial distributions in Fig. S7).*”;

the sentence on line 894 was revised as: “*This transport network is conceptually mapped in Fig. S8.*”;

the sentence on lines 909-910 was revised as: “*The unit cost of CO₂ transport in the three cases is mapped by county in Fig. S9.*”;

the sentence on lines 929-930 was revised as: “*As shown in Fig. S10, if the pipelines are designed to transport 20%, rather than 40%, of CO₂ emitted from all power...*”;

the sentence on lines 989-992 was revised as: “*The MATLAB codes used to optimize the consumption of biomass by a cost-minimization approach (take central scenario B90-2015-PC as an example), and the path of total electricity generation, and WWSN power generation over 2011–2030 are provided together in the Supplementary Zip file.*”.

Comment B17

Figure 2 is difficult! Before it can be presented, you should clearly state all the cases/scenarios you consider. Only 90% co-firing share of biomass is mentioned (e.g., on row 93), but the reader should know that you consider also 30% co-firing shares (with and without dedicated energy crops) and IGCC-retrofitting too. (The idea behind these additional scenarios should be discussed somewhere.)

Response

Thank you for the good suggestion.

In the **Introductory** paragraph, to introduce the “30% co-firing shares” and the “IGCC-retrofitting”, we added one sentence on lines 120-123 as: “*Furthermore, in order to improve the power generation efficiency, we follow a recent study to consider an advanced retrofit of power plants to an integrated gasification combined cycle (IGCC) system¹¹. Alternatively, we test the results for adopting a lower biomass co-firing ratio (30% weight)¹¹.*”.

To explain the motivation for all scenarios, one paragraph was added on lines 292-307 in the track version: “**Sensitivity analyses** *To address the sources of uncertainties, we design seven scenarios by varying parameters (see parameterization of each scenario in Table S4) relative to the central scenario, including B30-2015-PC (“B30” for 30% biomass co-firing with coal) and B90-2015-IGCC (“IGCC” for transferring PC to IGCC plants) to achieve a higher power generation efficiency, B30-2015-PC-EneCrop to examine the impact of removing agricultural and forestry residues (“EneCrop” for using dedicated energy crops (Miscanthus) only), B90-2030-PC to examine the impact of larger power capacities (“2030” for generating the electricity⁴² projected for 2030 (9.5 PWh yr⁻¹)), B90-2015-PC-BestCrop (“BestCrop” for shifting energy crops from Miscanthus to the best-yield crops²⁴), and noBiomass-2015-PC (“noBiomass” for using coal only in power plants with CCS) and B90-2015-PC-noCCS (“noCCS” for co-firing biomass without CCS facilities) to examine the effects of CCS and biomass alone, respectively, on marginal cost curves for emission reduction.*”.

To discuss on the results of scenario, the following paragraph was added on lines 308-326: “*Lu et al.¹¹ estimated that a carbon price of \$42–52 (t CO₂-eq)⁻¹ can abate 0.88 Gt CO₂-eq yr⁻¹ in China by using 20% of crop residues for 35% biomass co-firing in IGCC with a 90% CO₂ capture efficiency. In comparison, Fig. 3 shows that, when the ratio of biomass in co-firing is drawn downward from 90% to 30% (the scenario B30-2015-PC), the marginal cost to abate 0.88 Gt CO₂-eq yr⁻¹ in China declines slightly from \$60 (the central scenario B90-2015-PC) to \$56 (t CO₂-eq)⁻¹, but the potential for emission reduction is strongly reduced since coal remains the dominant fuel in the fuel mix for the power plants. The marginal cost will increase to \$146 (t CO₂-eq)⁻¹ due to a high cost of plant retrofitting, if as in Lu et al.¹¹, we consider biomass co-firing in the IGCC system (the scenario B90-2015-IGCC); in this scenario, the marginal cost could be even higher if rebuilding new plants is required when shifting the pulverized-coal boilers to IGCC^{54,55}. In the absence of agricultural and forestry residues, the marginal cost increases remarkably to \$202 (t CO₂-eq)⁻¹ (the scenario B30-2015-PC-EneCrop); by contrast, shifting the energy crops from Miscanthus, as the only source of energy crops in the central scenario, to the best-yield crops²⁴ alters the marginal cost moderately (the scenario B90-2015-PC-BestCrop). If national electricity generation increases from 4.24 to 9.5 PWh¹ yr⁻¹ over 2015-2030 (the scenario B90-2030-PC), the marginal cost declines by up to \$97 (t CO₂-eq)⁻¹ when the target is to abate more than 2 Gt CO₂-eq yr⁻¹ in China. This decline comes from a greater capacity of power plants that allows*

to utilize more biomass without requirements for long-distance transport of biomass and the captured CO₂.”.

A new **Table S4** added to the Supporting Information reads as:

Table S4. Parameterization of scenarios in sensitivity tests.

There are eight scenarios: (I) *B90-2015-PC*, where “B90” stands for 90% biomass co-firing, “2015” stands for generating electricity by coal-fired plants in 2015 (4.24 PWh yr⁻¹), and “PC” stands for using pulverized-coal (PC) plants, (II) *B30-2015-PC*, where “B30” stands for 30% biomass co-firing, (III) *B90-2015-IGCC*, where “IGCC” stands for transferring PC to integrated gasification combined cycle (IGCC) plants, (IV) *B30-2015-PC-EneCrop*, where “EneCrop” stands for using dedicated energy crops (*Miscanthus*) only, (V) *B90-2030-PC*, where “2030” stands for generating the projected electricity in 2030 (9.5 PWh yr⁻¹), and (VI) *B90-2015-PC-BestCrop*, where “BestCrop” stands for shifting energy crops from *Miscanthus* to the best-yield crops (Li et al., 2020), (VII) *noBiomass-2015-PC*, where “noBiomass” stands for using coal only in power plants, and (VIII) *B90-2015-PC-noCCS*, where “noCCS” stands for excluding CCS facilities in power plants. All costs are expressed in the US \$ of 2015.

Scenario	Methods
(I) B90-2015-PC	Our central scenario (all parameters are described in Table S2).
(II) B30-2015-PC	Different from B90-2015-PC, we consider 30% biomass co-firing with coal, rather than 90% co-firing. It affects the following parameters:  Unit cost of water consumption in power plants (μ_h^w) changes from \$7.3 to \$7.6 (t biomass)⁻¹, due to reduction in consumption of water in power plant (WC_P) from 4.4 to 4.2 t water (MWh electricity)⁻¹ (Fajardy & Mac Dowell, 2017), and power generation efficiency (η) from 25.1% to 27.3% (Yang et al., 2019); Unit cost of retrofitting power plants (μ_h^d) decreases from \$67.6 to \$57.0 (t biomass)⁻¹, due to reduction in investment cost of facilities in power plants (P) from \$2432 to \$1782 KW⁻¹ (Black & Veatch, 2012; Lu et al., 2019), annualized cost of fixed operation and maintenance for facilities (FOM) from \$58.2 to \$45.1 KW⁻¹ yr⁻¹ (Black & Veatch, 2012), and increase in power generation efficiency (η) from 25.1% to 27.3% (Yang et al., 2019). All other parameters are identical to those in B90-2015-PC.
(III) B90-2015-IGCC	Different from B90-2015-PC, we consider transferring from PC to integrated gasification combined cycle (IGCC) system for biomass and coal co-firing in power plants. It affects the following parameters:  Unit cost of water consumption in power plants (μ_h^w) decreases from \$7.3 to \$5.7 (t biomass)⁻¹, due to reduction in consumption of water in power plant (WC_P) from 4.4 to 2.4 t water (MWh electricity)⁻¹ (Macknick et al., 2012), and increase in power generation efficiency (η) from 25.1% to 35.8% (Lu et al., 2019); Unit cost of retrofitting power plants (μ_h^d) increases from \$67.6 to \$398.1 (t biomass)⁻¹, due to increase in investment costs of facilities in power plants (P) from \$2432 to \$6822 KW⁻¹ (Black & Veatch, 2012), annualized cost of fixed operation and maintenance for facilities (FOM) from \$58.2 to \$67.1 KW⁻¹ yr⁻¹ (Black & Veatch, 2012), annualized cost of variable operation and maintenance for facilities (VOM) from 6.6 to 9.7 US \$ MWh⁻¹ (Black & Veatch, 2012), capital recovery factor (CRF) from 0.09 to 0.174 (Lu et al., 2019), and power generation efficiency (η) from 25.1% to 35.8% (Lu et al., 2019); Unit cost of CO₂ capture and storage (μ_h^{cap}) decreases from \$82.5 to \$63.8 (t biomass)⁻¹ for agricultural residues and energy crops, and from \$82.3 to \$63.6 (t biomass)⁻¹ for wood products, due to decrease in the price of CO₂ capture and storage (PR_{CCS}) from \$53 to \$41 (t CO₂)⁻¹ (Koornneef et al., 2012; Rubin et al., 2015). All other parameters are identical to those in B90-2015-PC.
(IV) B30-2015-PC-EneCrop	Different from B30-2015-PC, we consider using dedicated energy crops (Miscanthus) only. The feedstocks of agricultural residues and wood products are set to be zero, and only energy crops are used for BECCS. All other parameters are identical to those in B30-2015-PC.
(V) B90-2030-PC	Different from B90-2015-PC, we consider that the national electricity generation increases from 4.24 PWh yr⁻¹ in 2015 to 9.5 PWh yr⁻¹ in 2030. All other parameters are identical to those in B90-2015-PC.

(VI)
*B90-2015-PC-
BestCrop*

Different from *B90-2015-PC*, for energy crops, we consider growing the best-yield crops, rather than *Miscanthus*, in China. The potential of energy crops is calculated based on a yield map of best-yield crops (Li et al., 2020).

All other parameters are identical to those in *B90-2015-PC*.

(VII)
*noBiomass-
2015-PC*

Different from *B90-2015-PC*, we consider using coal only in power plants, which are retrofitted for CCS. It affects the following parameters:

1. Biomass feedstocks are set to be zero, and there is no cost related to biomass;
2. Unit cost of CO₂ pipeline cost (μ_{iy}^t) decreases slightly because the power generation efficiency increases from 25.1% (power generation efficiency of biomass co-firing with coal, η) to 27.9% (power generation efficiency of coal-fired plant, v) (Yang et al., 2019) in this scenario;
3. Unit cost of water consumption in power plants (μ_h^w) increases from \$7.3 (t biomass)⁻¹ to 9.9 US \$ (t coal)⁻¹, due to changes in consumption of water (WC_P) from 4.4 to 4.1 t water (MWh electricity)⁻¹ (Fajardy & Mac Dowell, 2017), the heat content from 19 GJ (t biomass)⁻¹ of biomass (λ_h) to 25 GJ (t coal)⁻¹ of coal (λ_c) in this scenario (Cormos, 2012), and power generation efficiency from 25.1% (power generation efficiency of biomass co-firing with coal plant, η) to 27.9% (power generation efficiency of coal plant, v) (Yang et al., 2019);
4. Unit cost of retrofitting power plants (μ_h^d) decreases from \$67.6 (t biomass)⁻¹ to \$65.5 (t coal)⁻¹, due to changes in investment costs of facilities in power plants (P) from \$2432 to \$1457 KW⁻¹ (Lu et al., 2019), annual costs of fixed operation & maintenance (O&M) for facilities (FOM) from \$58.2 to \$38.5 KW⁻¹ yr⁻¹ (Black & Veatch, 2012), the heat content from 19 GJ (t biomass)⁻¹ of biomass (λ_h) to 25 GJ (t coal)⁻¹ of coal (λ_c) in this scenario (Cormos, 2012), and power generation efficiency from 25.1% (power generation efficiency of biomass co-firing with coal plant, η) to 27.9% (power generation efficiency of coal plant, v) (Yang et al., 2019);
5. Unit cost of CO₂ capture and storage (μ_h^{cap}) decreases from \$82.5 (t biomass)⁻¹ for agricultural residues and energy crops and \$82.3 (t biomass)⁻¹ for wood products to \$78.6 (t coal)⁻¹, which are calculated as:

$$\mu_h^{cap} = PR_CCS \cdot EC \cdot \varphi \cdot v \cdot \lambda_c / 3.6$$

where PR_CCS is the price of CO₂ capture and storage; EC is the efficiency of CO₂ capture; φ is the CO₂ emission factor of coal; v is the power generation efficiency of coal-fired plant; λ_c is the heating content of coal; and 3.6 converts 1 MWh to GJ.

For PR_CCS , we adopt a value of \$53±7 ton CO₂⁻¹ (Koornneef et al., 2012; Rubin et al., 2015);

For EC , we adopt a value of 90% (Anderson & Peters, 2016; Lu et al., 2019);

For φ , we adopt a value of 0.85±0.10 t CO₂ (MWh electricity)⁻¹ (Brander et al., 2011);

For v , we adopt a value of 27.9% (Yang et al., 2019);

For λ_c , we adopt a value of 25 GJ (t coal)⁻¹ (Cormos, 2012).

All other parameters are identical to those in *B90-2015-PC*.

Different from *B90-2015-PC*, CCS is not equipped in power plants.

(VIII)
*B90-2015-PC-
noCCS*

1. There is no cost of CCS and CO₂ transport.

2. Unit cost of water consumption in power plants (μ_h^w) changes from \$7.3 to \$8.8 (t biomass)⁻¹, due to changes in consumption of water (WC_P) from 4.4 to 3.7 t water (MWh electricity)⁻¹ (Fajardy & Mac Dowell, 2017), and power generation efficiency (η) from 25.1% to 36.2% (Yang et al., 2019);

3. Unit cost of retrofitting power plants (μ_h^d) decreases from \$67.6 (t biomass)⁻¹ to \$48.5 (t biomass)⁻¹, due to the reduction in investment costs of facilities in power plants (P) from \$2432 to \$975.4 KW⁻¹ (Black & Veatch, 2012; Lu et al., 2019), annualized cost of fixed operation and maintenance (FOM) from \$58.2 to \$44.9 KW⁻¹ yr⁻¹ (Black & Veatch, 2012), annualized cost of variable operation and maintenance (VOM) from \$6.6 to \$4.1 MWh⁻¹ (Black & Veatch, 2012), and the power generation efficiency (η) from 25.1% to 36.2% (Yang et al., 2019).

All other parameters are identical to those in *B90-2015-PC*.

(I) *B90-2015-
PC*

Our central scenario (all parameters are described in **Table S3**).

Unit emission

(II)
B30-2015-PC

Different from *B90-2015-PC*, we consider 30% biomass co-firing with coal, rather than 90% co-firing. Unit emission of retrofitting power plants (τ_h^d) increase from 0.0004 to 0.0005 t CO₂ (t biomass)⁻¹, due to an increase in power generation efficiency (η) from 25.1% to 27.3% (Yang et al., 2019).

All other parameters are identical to those in *B90-2015-PC*.

(III)

Different from *B90-2015-PC*, we consider integrated gasification combined cycle (IGCC) plants, rather than pulverized-coal (PC) plants. Unit emission of retrofitting power plants (τ_h^d) changed from 0.0004 to 0.0017 t CO₂ (t biomass)⁻¹, due to the change in equivalent CO₂ emissions to produce 1

B90-2015-IGCC	MWh electricity in the adjustment of facilities (EF_{adj}) from 0.00033 to 0.00088 t CO _{2eq} (MWh electricity) ⁻¹ (Lu et al., 2019), and electricity efficient (η) from 25.1% to 35.8% (Lu et al., 2019) in this scenario. All other parameters are identical to those in B90-2015-PC .
(IV) B30-2015-PC-EneCrop	Different from B30-2015-PC , we consider using dedicated energy crops (Miscanthus) only. The feedstocks of agricultural residues and wood products are set to be zero, and only energy crops are used for BECCS. All other parameters are identical to those in B30-2015-PC .
(V) B90-2030-PC	Different from B90-2015-PC , we consider that the national electricity generation increases from 4.24 PWh yr ⁻¹ in 2015 to 9.5 PWh yr ⁻¹ in 2030. All other parameters are identical to those in B90-2015-PC .
(VI) B90-2015-PC-BestCrop	Different from B90-2015-PC , for energy crops, we consider growing the best-yield crops, rather than Miscanthus , in China. The potential of energy crops is calculated based on a yield map of best-yield crops (Li et al., 2020). All other parameters are identical to those in B90-2015-PC .
(VII) noBiomass-2015-PC	Different from B90-2015-PC , we consider using coal only in power plants, which are retrofitted for CCS. It affects the following parameters: 1. Biomass feedstocks are set to be zero; 2. Unit emission of retrofitting power plants (r_h^d) increases from 0.0004 t CO ₂ (t biomass) ⁻¹ to 0.0006 t CO ₂ (t coal) ⁻¹ , due to increases in power generation efficiency from 25.1% (power generation efficiency of biomass co-firing with coal plant, η) to 27.9% (power generation efficiency of coal plant, ν) (Yang et al., 2019), and the heat content to 25 GJ (t coal) ⁻¹ of coal (λ_c) in this scenario (Cormos, 2012). 3. Unit emission of captured CO ₂ from coal power plant is 1.5 t CO ₂ (t coal) ⁻¹ (Brander et al., 2011; Cormos, 2012; Yang et al., 2019; Anderson & Peters, 2016; Lu et al., 2019); All other parameters are identical to those in B90-2015-PC .
(VIII) B90-2015-PC-noCCS	Different from B90-2015-PC , CCS is not equipped in power plants. There is no emission related to CCS. All other parameters are identical to those in B90-2015-PC .

Comment B18

Retrofitting is central in your paper, but it is not really explained what is meant by that. I suppose, there are separate decision to retrofit biomass feeding and CCS. What does biomass retrofitting mean: biomass handling, storages, pretreatment etc.? IGCC approach seems to be a bigger task than just retrofitting the pulverized fuel digestion. The CCS has separate aspects. These could be explained shortly.

Response

Thank you for the good suggestion. To explain “*biomass utilization*”, “*retrofitting*”, “*CCS*” and “*IGCC*”, the following sentences were added on lines 92-113 in the **Introduction** paragraph as: “*We aim at facilitating a transition into a carbon-emission-negative power system²³ by BECCS. Our central hypothesis is that BECCS can be harnessed efficiently over China’s expanse by 1) biomass utilization, including collection and pretreatment of agricultural and forestry residues, energy crop production, biomass handling, and transport to power plants, 2) retrofitting of coal-fired power plants to be suitable for biomass co-firing (90% weight) and CCS, and 3) utilization of pipelines to transport CO₂ to geological storage sites (see details in **Methods**). In brief, we assess the feedstocks of 19 ligno-cellulosic biomass in mainland China from agricultural and forestry residues (excluding grains) and dedicated energy crops (*Miscanthus* or other high-yield crops²⁴) on available marginal lands or grasslands. We explicitly optimize the consumption of biomass for BECCS in 2836 counties to achieve a target of national emission reduction, based on: (i) a life-cycle analysis of GHG emissions and costs in the processes involved in BECCS and (ii) spatially explicit constraints on the supply of biomass feedstock, capacity for electricity generation in existing power plants, and capacity for geological carbon storage. Such geographical logistical constraints in a large country like China have not been investigated at a county level by previous top-down studies^{1-3,11,23,25}. It allows us to compare the marginal cost and potential for decarbonizing the power system of BECCS versus biomass utilization or CCS alone. Furthermore, in order to improve the power generation efficiency, we follow a recent study to consider an advanced retrofit of power plants to an integrated gasification combined cycle (IGCC) system¹¹. Alternatively, we test the results for adopting a lower biomass co-firing ratio (30% weight)¹¹. Such an integrated strategy*”

taking into account the specificities of China can be viewed as an early entry point to address global mitigation and national energy security challenges — kick-starting an innovative cycle.”.

We followed the suggestion of Reviewer 2 to consider the effects of biomass utilization and CCS alone. Please refer to our response to **Comment B19**.

We have also clarified the limitation by transferring to an IGCC system. Please refer to our response to **Comment B45**.

Comment B19

The paper examines an all-in approach, where biomass co-firing and CCS (i.e., BECCS) are jointly introduced. It would be interesting to see, what would happen if (i) biomass co-firing without CCS and (ii) not using biomass but implementing CCS on coal-fired power plants were allowed too. I suppose, it could lower the marginal costs, if these decisions could be done separately. Linear program could optimize these options.

Response

Thank you for the good suggestion, and we agree with Reviewer 2 that it is interesting to consider these two scenarios. We have modeled the utilization of biomass and CCS alone in our optimization model, of which the cost curves are compared with that of BECCS in the new **Fig 3** and **Table 1**.

To introduce these scenarios, we added the following sentence on lines 293-307 in the track version as: *“To address the sources of uncertainties, we design seven scenarios by varying parameters (see parameterization of each scenario in **Table S4**) relative to the central scenario, including B30-2015-PC (“B30” for 30% biomass co-firing with coal) and B90-2015-IGCC (“IGCC” for transferring PC to IGCC plants) to achieve a higher power generation efficiency, B30-2015-PC-EneCrop to see the impact of removing agricultural and forestry residues (“EneCrop” for using dedicated energy crops (Miscanthus) only), B90-2030-PC to see the impact of larger power capacities (“2030” for generating the electricity⁴² projected for 2030 (9.5 PWh yr⁻¹)), B90-2015-PC-BestCrop (“BestCrop” for shifting energy crops from Miscanthus to the best-yield crops²⁴), and noBiomass-2015-PC (“noBiomass” for using coal only in power plants with CCS), and B90-2015-PC-noCCS (“noCCS” for co-firing biomass without CCS facilities) to see the effect of CCS and biomass alone, respectively, on marginal cost curves for emission reduction.”,*

and one paragraph was added on lines 327-342 in the track version to discuss results in these two scenarios: *“As BECCS combines biomass utilization and CCS, it is interesting to investigate the individual effects by deploying only one of these two technologies. Relative to the central scenario (B90-2015-PC), the marginal cost to abate 1 Gt CO_{2-eq} yr⁻¹ is increased from ~\$60 to ~\$110 (t CO_{2-eq})⁻¹ if only CCS is implemented on coal-fired power plants in absence of biomass (noBiomass-2015-PC), but it is reduced to ~\$60 (t CO_{2-eq})⁻¹ in a scenario of biomass co-firing without CCS (B90-2015-PC-noCCS). The marginal cost in B90-2015-PC-noCCS to abate 0.3 Gt CO_{2-eq} yr⁻¹ is even close to zero, where the value of substituted coal almost offsets the cost of utilizing biomass without requirements of long-distance transport and retrofitting power plants without requirements for CCS (see a detailed comparison of cost and emission between scenarios in **Table 1**). It is in line with previous studies that biomass utilization could be an option for a shallow decarbonization in China^{11,41}. However, to abate emissions by more than 0.9 Gt CO_{2-eq} yr⁻¹, the marginal cost increases sharply as the target of emission reduction increases due to high costs in the acquisition and logistics of biomass in B90-2015-PC-noCCS, where CCS alone becomes a cheaper option (noBiomass-2015-PC). To abate emissions by more than 2 Gt CO_{2-eq} yr⁻¹, the marginal cost is the lowest when combining these two technologies together (B90-2015-PC).”.*

However, we cannot yet model the optimization among the options of BECCS, biomass utilization, and CCS alone, because the marginal cost of BECCS is dependent on the constraints by biomass feedstocks, power generation capacities and carbon storage capacities. For example, if we adopt biomass utilization alone at a low marginal cost as shown in **Fig 3** to abate 0.1 Gt CO_{2-eq} yr⁻¹, it will consume the cheap biomass and then increase the marginal cost of BECCS. To make the relationship between these options clearer, we added the following sentences on lines 342-346: *“Note that, if we adopt biomass use alone to abate emissions at a low marginal cost as shown in Fig 3, it will consume the cheap biomass and thus increase the marginal cost curve of BECCS. These cost considerations imply that the selection of BECCS, biomass utilization and CCS depends on the overall target of national emission reduction.”.*

A new **Fig 3** reads as:

Fig. 3 | Marginal cost curves of CO₂ emission reduction by BECCS in China. Scenarios include: (I) B90-2015-PC, where “B90” stands for 90% biomass co-firing, “2015” stands for generating electricity by coal-fired plants in 2015 (4.24 PWh yr⁻¹), and “PC” stands for using pulverized-coal (PC) plants, (II) B30-2015-PC, where “B30” stands for 30% biomass co-firing, (III) B90-2015-IGCC, where “IGCC” stands for transferring PC to integrated gasification combined cycle (IGCC) plants, (IV) B30-2015-PC-EneCrop, where “EneCrop” stands for using dedicated energy crops (*Miscanthus*) only, (V) B90-2030-PC, where “2030” stands for generating the projected electricity in 2030 (9.5 PWh yr⁻¹)⁴², and (VI) B90-2015-PC-BestCrop, where “BestCrop” stands for shifting energy crops from *Miscanthus* to the best-yield crops²⁴, (VII) noBiomass-2015-PC, where “noBiomass” stands for using coal only in power plants, and (VIII) B90-2015-PC-noCCS, where “noCCS” stands for excluding CCS facilities in power plants. The national CO₂ emission reduction (calculated from the substituted coal consumption⁴² and the CO₂ emission factor of coal⁵³) by hydropower, wind, solar and nuclear power in China in 2015 as well as the range of their marginal costs (calculated from their prices of electricity generation in different provinces⁵² and the CO₂ emission factor of coal⁵³) are shown as vertical lines. Uncertainty in the marginal cost curve in the central scenario is assessed by considering uncertainty in parameterization in a Monte Carlo simulation (see details in **Methods**), which is shown as the shared area.

A new **Table 1** added to the main text reads as:

Table 1. Costs and CO₂-equivalent emissions in the scenarios of B90-2015-PC, noBiomass-2015-PC, and B90-2015-PC-noCCS.

Scenarios	B90-2015-PC			noBiomass-2015-PC			B90-2015-PC-noCCS		
	0.1	1	1.9	0.1	1	1.9	0.1	1	1.9
Abatement target (Gt CO ₂ -eq yr ⁻¹)									
Biomass consumption (Gt yr ⁻¹)	0.04	0.4	0.8	-	-	-	0.1	0.8	1.8
Total costs (billion \$ yr ⁻¹)	5.2	58.4	118.4	11	110.1	209.8	-0.7	10.2	239.2
Items	Marginal cost (\$ (t CO ₂ -eq) ⁻¹)								
Coal substitution (lower cost)	-31.9	-33.7	-33.7	-	-	-	-82.0	-91.4	-204.0
Retrofitting plants for co-firing and CCS	27.1	28.6	28.7	43.7	43.7	43.7	34.7	38.7	86.3
Biomass acquisition	12.4	13.1	13.1	-	-	-	22.2	47.1	234.2
Biomass pretreatment	5.5	6.5	6.4	-	-	-	9.9	12.0	35.3
Biomass transport	1.1	1.4	12.8	-	-	-	1.9	38.7	248.8
CO ₂ capture and storage (CCS)	33.0	34.9	34.9	52.4	52.4	52.4	-	-	-
CO ₂ transport	2.2	2.6	5.2	6.9	7.8	8.4	-	-	-
Fertilizer usage	0.0	3.7	3.2	-	-	-	0.0	5.6	34.2
Water consumption	2.9	3.6	3.5	6.6	6.6	6.6	6.3	9.0	82.9

Total	52.3	60.7	74.1	109.6	110.5	111.1	-7.0	59.7	517.7
Sources	Marginal emission (t CO ₂ -eq (t CO ₂ -eq) ⁻¹)								
Emission in land-use change	0.00	0.00	0.01	-	-	-	0.00	0.00	0.73
Emission in fertilizer production	0.00	0.03	0.03	-	-	-	0.00	0.05	0.32
Emission in fertilizer application	0.00	0.02	0.01	-	-	-	0.00	0.03	0.14
Emission in biomass treatment	0.08	0.09	0.09	-	-	-	0.14	0.17	0.46
Emission in biomass transport	0.00	0.00	0.01	-	-	-	0.00	0.03	0.19
Emission in retrofitting power plants	0.00	0.00	0.00	0.00	0.00	0.00	-	-	-
Emission in substituted coal	-0.44	-0.46	-0.47	-	-	-	-1.14	-1.28	-2.84
Carbon sequestration in CCS	-0.64	-0.68	-0.68	-1.00	-1.00	-1.00	-	-	-
Total	-1.00	-1.00	-1.00	-1.00	-1.00	-1.00	-1.00	-1.00	-1.00

Comment B20

As a minor thing, in many places, the word “biofuel” could be replaced with “biomass”.

Response

Thank you for the good suggestion, and “*biofuel*” is replaced with “*biomass*” in our paper.

Comment B21

3. Previous assessments

The paper would benefit from proper comparison and discussion of the results in the light of a largish literature on bioenergy potential of China. A quick search provided several related studies (see references below), which try to assess the biomass potential. The reliability of the current results would be improved, if they were contrasted with previous work.

Response

Thank you for the good suggestion. We agree with Reviewer 2 that it is important to compare our estimate of China’s biomass potential with previous studies. We added a paragraph to compare with thirteen studies that we can find on Web of the Science (Li et al., 2001; Li et al., 2005, Yang et al., 2010; Zhou et al., 2011; Zhuang et al., 2011; Qiu et al., 2014; Chen et al., 2016; Xue et al., 2016; Gao et al., 2016; Zhao, 2018; Nie et al., 2019; Nie et al., 2020; Kang et al., 2020). We added a new paragraph on lines 148-157 in the track version as: “*Figure 1 puts the estimates of bioenergy in the context of previous studies²⁶⁻³⁸. Our 2015 estimate for 11 types of agricultural residues is close to the high end of previous estimates (14.7-16.8 EJ yr⁻¹). For example, Li et al. only account for 5 types of agricultural residues²⁶, while Li et al.²⁷ and Yang et al.²⁸ derived their estimates for earlier years than ours. For forestry residues, our estimate based on the statistic of wood products is lower than that by Yang et al.²⁸, who estimated the wood feedstocks based on the forestry area, biomass resource yield and a constant collectable rate, but close to other estimates^{34,37,38}. For energy crops, our estimate, based on the 2015 satellite land-use data³⁹ and the theoretical yield of Miscanthus³³, is close to the median of previous estimates^{30,33,35-37}.”.*

A new Fig. 1 added to the main text reads as:

Fig. 1 | Comparison between bioenergy potential in this study and the literature. Solid circles give the results as a maximum production potential in this study, while previous estimates are shown as open circles²⁶⁻³⁸. Error bars show the uncertainty range in this study. The bioenergy is converted from the ton of standard coal equivalent (tce)^{26,28,32} using a constant of 29.3 GJ tce⁻¹ (from ref²⁶), or from the weight of biomass^{30-33,37} using a constant of 19 GJ (t biomass⁻¹) (from ref⁴⁰).

References:

Li, J. J., Xing, Z., DeLaquil, P. & Larson, E. D. Biomass energy in China and its potential. *Energy for Sustainable Development* **5**, 66-80 (2001).

Li, J. et al. Assessment of sustainable energy potential of non-plantation biomass resources in China. *Biomass Bioenergy* **29**, 167-177 (2005).

Yang, et al. Quantitative appraisal and potential analysis for primary biomass resources for energy utilization in China. *Renewable Sustainable Energy Rev.* **14**, 3050-3058 (2010).

Zhou, X., et al. Assessment of sustainable biomass resource for energy use in China. *Biomass Bioenergy* **35**,

1-11 (2011).

Zhuang, D., Jiang, D., Liu, L. & Huang, Y. Assessment of bioenergy potential on marginal land in China. *Renewable Sustainable Energy Rev.* **15**, 1050-1056 (2011).

Qiu, H., Sun, L., Xu, X., Cai, Y. & Bai, J. Potentials of crop residues for commercial energy production in China: A geographic and economic analysis. *Biomass Bioenergy* **64**, 110-123 (2014).

Chen, W., Wu, F. & Zhang, J. Potential production of non-food biofuels in China. *Renewable Energy* **85**, 939-944 (2016).

Xue, S., Lewandowski, I., Wang, X. & Yi, Z. Assessment of the production potentials of Miscanthus on marginal land in China. *Renewable Sustainable Energy Rev.* **54**, 932-943 (2016).

Gao, J. et al. An integrated assessment of the potential of agricultural and forestry residues for energy production in China. *Gcb Bioenergy* **8**, 880-893 (2016).

Zhao, G. Assessment of potential biomass energy production in China towards 2030 and 2050. *Int. J. Sustainable Energy* **37**, 47-66 (2018).

Nie, Y. et al. Assessment of the potential and distribution of an energy crop at 1-km resolution from 2010 to 2100 in China—The case of sweet sorghum. *Appl. Energy* **239**, 395-407 (2019).

Nie, Y. et al. Spatial distribution of usable biomass feedstock and technical bioenergy potential in China. *GCB Bioenergy* **12**, 54-70 (2020).

Resource and Environment Data Cloud Platform (REDCP). *Remote Sensing Monitoring Data of Land-use in China in 2015*. <http://www.resdc.cn/data.aspx?DATAID=184> (2019).

Kumar, A., Cameron, J. B. & Flynn, P. C. Biomass power cost and optimum plant size in western Canada. *Biomass Bioenergy* **24**, 445-464 (2003).

Kang, Y. et al. Bioenergy in China: Evaluation of domestic biomass resources and the associated greenhouse gas mitigation potentials. *Renewable Sustainable Energy Rev.*, 109842 (2020).

Comment B22

In addition, with an omitted discussion, I cannot be certain, whether the spatial approach is such novelty after all. (There are spatial analyses in the papers listed in the references.)

Response

To our knowledge, there are studies applying a spatially explicit method for several countries (e.g., Kraxner, 2014a,b, see a literature review in **Table S1**), but there is only one study which has applied a spatially explicit approach to estimate marginal cost curves of BECCS by accounting for the constraints by biomass and carbon storage over western North America (Sanchez et al. 2015). To clarify it, the sentence on lines 89-92 in the track version was revised as: “*Studies that have investigated the potential and marginal cost of BECCS at a regional scale are listed in Table S1. Spatially explicit method has been applied for several countries^{19,20}, but there is, to our knowledge, only one study that estimates marginal cost curves of BECCS over western North America⁴.*”, and the following sentence was added on lines 244-248 as: “*To our knowledge, there is only one study that has estimated the marginal cost curve for carbon emission reduction using BECCS over western North America, where the marginal cost is $\sim \$100 (t CO_{2-eq})^{-1}$ to abate $0.1 Gt CO_{2-eq} yr^{-1}$ by BECCS⁴; in comparison, a marginal cost of $\$100 (t CO_{2-eq})^{-1}$ abates $2.0_{-0.6}^{+0.9} Gt CO_{2-eq} yr^{-1}$ in China as shown in Fig. 3.*”

Reference:

Kraxner, F. et al. BECCS in South Korea—Analyzing the negative emissions potential of bioenergy as a mitigation tool. *Renewable Energy* **61**, 102-108 (2014a).

Kraxner, F. et al. Energy resilient solutions for Japan—a BECCS case study. *Energy Procedia* **61**, 2791-2796 (2014b).

Sanchez, D. L., Nelson, J. H., Johnston, J., Mileva, A. & Kammen, D. M. Biomass enables the transition to a carbon-negative power system across western North America. *Nat. Clim. Chang.* **5**, 230-234 (2015).

Comment B23

Similarly, a discussion on marginal costs of emission abatement is lacking. Are these costs high or low compared to other studies or other emission abatement solutions?

Response

Thank you for the good suggestion. To compare the marginal cost of BECCS to other studies or other emission abatement options, we added a new paragraph on lines 244-263 in the track version as: “*To our knowledge, there is only one study that has estimated the marginal cost curve for carbon emission reduction using BECCS over western North America, where the marginal cost is $\sim \$100$ (t CO_{2-eq})⁻¹ to abate 0.1 Gt CO_{2-eq} yr⁻¹ by BECCS⁴; in comparison, a marginal cost of $\$100$ (t CO_{2-eq})⁻¹ abates 2.0^{+0.9}_{-0.6} Gt CO_{2-eq} yr⁻¹ in China as shown in Fig. 3.*

Recently, in order to improve the power generation efficiency, Lu et al.¹¹ suggested that BECCS is cost-competitive with coal in integrated gasification combined cycle (IGCC) plants at a carbon price of $\$42\text{--}52$ (t CO_{2-eq})⁻¹, comparing to $\$60$ (t CO_{2-eq})⁻¹ in our central scenario. This difference is likely due to the fact that Lu et al.¹¹ did not consider the dependence of marginal cost on the target of emission reduction, due to lack of logistics system for biomass and CO₂ transport. For CCS as an emerging decarbonization option, the marginal cost increases from $\$70$ (t CO_{2-eq})⁻¹ (ref⁴⁸) to $\$150$ (t CO_{2-eq})⁻¹ (ref⁴⁹) is due to a high cost of carbon separation and compression. Furthermore, over China, the potential of nuclear electricity is limited by safety risk⁵⁰, while the expansion of hydropower is limited by uncertainty in the water resource⁵¹. When we compare the marginal cost to abate the same amount of emissions, the marginal cost of BECCS is close to that of wind and solar power (from $\$38$ to $\$109$ (t CO_{2-eq})⁻¹, estimated from the prices of wind and solar power⁵²), but higher than that of hydropower and nuclear power (from $\$-18$ to $\$11$ (t CO_{2-eq})⁻¹, estimated from the prices of hydropower and nuclear power⁵²) as shown in Fig. 3.”.

Comment B24

4. BECCS-Paper would benefit from a clear exposition of BECCS’s effects. How much efficiency of power generation is lowered and what are the resulting investment costs (biomass use increases marginal costs, but the fixed costs need to be considered too.) How much does the electricity price increase? (I do not know how electricity price is determined in China.)

Response

Thank you for the good suggestion.

To clarify the effect of BECCS on electricity price due to the investment costs, we added the following sentences on lines 383-388 in the track version as: “*Consequently, the price of electricity generation will increase to achieve a higher target of emission reduction. For example, to abate 1, 3 and 5 Gt CO_{2-eq} yr⁻¹ by BECCS in China, the power generation efficiency is reduced from 39.3% (coal-fired plants) to 25.1% (90% biomass co-firing) to generate 0.55, 1.74 and 3.33 PWh yr⁻¹ of electricity in BECCS, which will increase the electricity price from $\$0.060$ kWh⁻¹ to $\$0.074$, 0.125 and 0.248 kWh⁻¹, respectively.”.*

The electricity price ($\$0.060$ kWh⁻¹) in 2015 is taken from NEA (2018), and the price increase is calculated as the ratio of costs to the total electricity generation by coal-fired plants in China (4.24 PWh yr⁻¹ in 2015), e.g. $\$0.074$ kWh⁻¹ = $0.060 + 58 / 4.24 / 1000$, since the price of electricity is controlled by the central government in China.

Reference:

National Energy Administration (NEA) of the People’s Republic of China. Regulatory Bulletin on the State Power Price in 2017. http://www.nea.gov.cn/137519800_15391333051221n.pdf (2018).

Comment B25

Comparison without BECCS would be very useful! Scenarios with separate biomass retrofitting and CCS implementation and after that the full BECCS scenario. Or if the decisions to use biomass and CCS could be made separately, the linear program would find an optimal mix of solutions.

Response

Thank you for the good suggestion. We added two scenarios: (VII) *noBiomass-2015-PC*, where “noBiomass” stands for using coal in power plants only, and (VIII) *B90-2015-PC-noCCS*, where “noCCS” stands for no CCS facilities in power plants (see detailed parameterization of these two scenarios in a new **Table S4** added

to the Supporting Information). These results are shown in the new Fig. 3 and Table 1, and discussed in our paper (please refer to the discussion added to the main text and our response to **Comment B19**).

Comment B26

The co-firing shares of biomass are very high (up to 90%). Is this feasible? What kind of preprocessing is needed? Pelletization, torrefication, etc.? I suppose, biomass has notably lower energy density than coal. Therefore, an ordinary pulverized fuel burner cannot be fed fuel at its optimal/maximal rate with biomass in energy terms. Also, the amount of volatiles and moisture in biomass differs from coal, which distorts the optimized combustion process of a pulverized coal boiler. Hence, a heavy preprocessing of biomass can be needed. I suppose, this is taken into account at some level by low efficiency coefficient (26.6%) and preprocessing costs? They could be discussed more explicitly.

Response

A 90% co-firing ratio of biomass is reported in China's power plants (e.g., Li et al., 2012; Huang et al., 2019; Yang et al., 2019). For example, it is suggested that “while the co-firing ratio (CFR) in PC plants is usually less than 10% due to technical constraints [16], [17], this ratio will gradually increase with the development of biomass pretreatment technology [18] and boiler improvement.” (Yang et al., 2019). Therefore, we consider the costs of BECCS due to biomass pretreatment, retrofitting power plants for biomass co-firing, and the effect of reducing power generation efficiency on increasing the fuel demand in our model.

To clarify the assumption in our model and the associated limitation, the following sentences were added on lines 782-794 in the track version as: “Transformation of the power sector to BECCS operation is a massive project for China⁴¹. To reach a 90% biomass co-firing ratio for a deep decarbonization target, it requires boiler adjustments and biomass pretreatment technologies, such as drying, grinding, milling, torrefaction, pelletization⁷³⁻⁷⁵. We accounted for the costs in biomass pretreatment and power plants retrofitting for biomass co-firing (see parameterization of two cost items (μ_h^p and μ_v^d) in **Table S2**), and a reduction of the power generation efficiency⁴¹. As a result of biomass co-firing, the power generation efficiency in China's PC power plants decreases from 39.3% for coal combustion to 36.2% for 90% biomass co-firing without CCS, 27.3% for co-firing 30% of biomass with CCS, and 25.1% for co-firing 90% of biomass with CCS^{41,76}. For IGCC plants, we adopted a power generation efficiency of 35.8% from a recent study for China¹¹. More studies are needed to examine the feasibility of a 90% biomass co-firing ratio and explore other technologies that can improve the power generation efficiency.”.

References:

- Li, J., Brzdekiewicz, A., Yang, W., & Blasiak, W. Co-firing based on biomass torrefaction in a pulverized coal boiler with aim of 100% fuel switching. *Appl. Energy* **99**, 344-354. (2012).
- Huang, C. W., Li, Y. H., Xiao, K. L., & Lasek, J. Cofiring characteristics of coal blended with torrefied Miscanthus biochar optimized with three Taguchi indexes. *Energy* **172**, 566-579 (2019).
- Yang, B., Wei, Y. M., Hou, Y., Li, H., & Wang, P. Life cycle environmental impact assessment of fuel mix-based biomass co-firing plants with CO₂ capture and storage. *Appl. Energy* **252**, 113483 (2019).

Comment B27

Why is the biomass share so low (30%) in IGCC plants? Should not it be easier to get high biomass shares with IGCC than with pulverized combustion?

Response

Thank you for the good suggestion. In our previous manuscript, we adopted a biomass co-firing ratio of 30% in IGCC plants to be comparable with the scenario of B30-2015-PC. We agree with Reviewer 2 that “it (should) be easier to get high biomass shares with IGCC than with pulverized combustion”, and now we adopt a biomass co-firing ratio of 90% in IGCC plants in the scenario of B90-2015-IGCC to be comparable with the scenario of B90-2015-PC. Please refer to configurations of all scenarios in our response to **Comment B17**.

Comment B28

5. Data set-The biomass data set is impressive and to my understanding the allocation of provincial figures between counties seems reasonable. However, I have some reservations about technological data.

Response

Thank you for the positive comments. We hope to provide basic data to help decision makers determine the potential contribution of BECCS to decarbonize the power system in a near term, and present a conceptual study for how the data can be used to develop an optimal model by retrofitting existing power plants for biomass utilization and CCS together at a county level. We hope that our paper can motivate more studies to estimate the cost curves of mitigation options in China.

We have clarified the parameterization of technological data (see our revision of **Tables S2-S4** in the Supporting Information and our response to **Comments B17, B29**), updated the power generation efficiency of power plants measured in China (see our response to **Comment B30**), and justified the sources of technological data (see our response to **Comments B31, B32, B45, B55, B56, B61**). Please see our point-by-point response to comments below.

Comment B29

The calculation of unit emission factors is not explicit. Notes of Table S4 tell that these are functions of specific parameters. The functions should be explicitly stated.

Response

Sorry for the confusion and thank you for the good suggestion. We now present all functions explicitly in the new **Table S2** (for calculation of unit costs) and new **Table S3** (for calculation of unit emissions).

A new **Table S2** reads as:

Table S2. Unit costs of BECCS in the central scenario.

B90-2015-PC is our central scenario, where “B90” stands for 90% biomass co-firing with coal, “2015” stands for generating electricity in existing power plants in 2015 (total electricity generation is 4.24 PWh yr⁻¹), and “PC” stands for electricity generation in pulverized-coal (PC) plants. Uncertainty is given as a percentage relative to the central value. All costs are expressed in the US \$ of 2015.

Cost items	Unit cost (US \$ (t biomass) ⁻¹) and its uncertainty	Methods
Biomass acquisition (μ_h^a)	31.0 (±10%, agricultural residues, and fuelwood)	Unit cost of biomass acquisition of agricultural residues and energy crops is calculated as a sum of costs in seeding, pesticide, machine cultivation, machine sowing, machine harvests, land occupation, and labor consumption from Table S8 (Khanna et al., 2008; Komarek, 2013; HPBS, 2019).
	240.3 (±10%, commercial wood)	Because fuelwood is not commercial with a definite price in China, we assume that its unit cost is the same as agricultural residues.
	131.6 (±10%, energy crops)	Price of commercial wood in China (\$240.3 (t biomass) ⁻¹) is compiled from the State Forestry Administration (SFA) of People’s Republic of China (SFA, 2015).

Unit cost of biomass due to fertilizer usage is calculated as:

$$\mu_h^f = \sum_{m=1}^3 C_{fer_{mh}} \cdot PR_{fer_m}$$

Fertilizer usage (μ_h^f)	9.1 (±10%, agricultural residues)	where m is the type of fertilizer (1, 2, 3 for N fertilizer (N), phosphorus pentoxide (P ₂ O ₅) and potassium oxide (K ₂ O) fertilizers, respectively); h is the type of biomass; C_{fer} is the demand of fertilizer to sustain the element balance in soils when biomass is taken away from the land for combustion in power plants; and PR_{fer} is the price of fertilizer.
	0 (±0%, wood products)	For C_{fer} , we adopt values of 0.0069±0.0004 t N (t crop) ⁻¹ , 0.0026±0.0005 t P ₂ O ₅ (t crop) ⁻¹ , and 0.0128±0.0022 t K ₂ O (t crop) ⁻¹ for agricultural residues (Ren et al., 2019); 0.0131±0.0090 t N (t energy crop) ⁻¹ , 0.0112±0.0072 t P ₂ O ₅ (t energy crop) ⁻¹ , and 0.0247±0.0092 t K ₂ O (t energy crop) ⁻¹ for energy crops (Lewandowski et al., 1995).
	19.2 (±50%, energy crops)	For PR_{fer} , we adopt values of \$584 (t N) ⁻¹ for N, \$284 (t P ₂ O ₅) ⁻¹ for P ₂ O ₅ , and \$340 (t K ₂ O) ⁻¹ for K ₂ O (Khanna et al., 2008).

Biomass transport (μ_{ix}^b)	0.2-25.1 ($\pm 20\%$, within a county)	Unit cost of biomass transport is calculated as: $\mu_{ix}^b = BTD_{ix} \cdot BTP$
	4.7-72.0 ($\pm 20\%$, between counties in one province)	where i denotes a county; x is the case of biomass transport (within one county, between counties in one province, and between different provinces); BTD is the distance of biomass transport (km), which is described in the section “Constraints on biomass supply, electricity generation and carbon storage” in Methods of the main text; BTP is the unit cost of biomass transport per distance and per ton biomass.
	173.6 ($\pm 20\%$, between provinces)	For BTP , we adopt a value of $\$0.10 \pm 0.05$ (t biomass) $^{-1}$ km $^{-1}$ for China (Yu & Fan, 2009; Lu et al., 2019).
CO ₂ pipeline transport (μ_{iy}^t)	0.1-59.4 ($\pm 20\%$, within a county)	Unit cost is calculated for pipelines constructed to transport CO ₂ captured from power plants, using a function developed by McCollum and Ogden. (McCollum & Ogden, 2006). The detailed method is described in the section “Costs of CO ₂ transport by pipeline” in Methods of the main text.
	0.6-66.3 ($\pm 20\%$, between counties in one province)	
	27.7-85.0 ($\pm 20\%$, between provinces)	
Biomass pretreatment (μ_h^p)	15.3 ($\pm 20\%$, agricultural residues)	Unit cost of biomass pretreatment by consuming diesel and electricity is calculated as: $\mu_h^p = DEC_{pre_h} \cdot PR_{DE} / LHV_{DE} + ELC_{pre_h} \cdot PR_{EL} / 1000 / 3.6$
	13.8 ($\pm 10\%$, wood products)	where h is the type of biomass; DEC_{pre} is the consumption of fuel energy (using diesel) in biomass pretreatment; LHV_{DE} is the lower heating value of diesel; ELC_{pre} is the consumption of electricity in biomass pretreatment; PR_{DE} is the price of diesel; and PR_{EL} is the price of electricity; and 3.6 converts 1 MWh to GJ.
	19.8 ($\pm 30\%$, energy crops)	For DEC_{pre} , we adopt values of 7.4 MJ (t biomass) $^{-1}$ for agricultural residues, 0 MJ (t biomass) $^{-1}$ for wood products, and 108 MJ (t biomass) $^{-1}$ for energy crops (Fajardy & Mac Dowell, 2017).
		For LHV_{DE} , we adopt a value of 37.4 \pm 1.7 MJ (Litter diesel) $^{-1}$ (Fajardy & Mac Dowell, 2017). For PR_{DE} , we adopt a value of $\$0.82 \pm 0.013$ (Litter diesel) $^{-1}$ (Chang et al, 2015). For ELC_{pre} , we adopt values of 911 \pm 166 MJ (t biomass) $^{-1}$ for agricultural residues, 828 MJ (t biomass) $^{-1}$ for wood products, and 1048 \pm 403 MJ (t biomass) $^{-1}$ for energy crops (Fajardy & Mac Dowell, 2017). For PR_{EL} , we adopt a value of $\$60 \pm 10$ (MWh electricity) $^{-1}$ (NEA, 2018).
Water consumption in agricultural irrigation (μ_h^g)	1.2 ($\pm 20\%$, agricultural residues)	Unit cost of water consumption in agricultural irrigation is calculated as: $\mu_h^g = WC_{A_h} \cdot PR_{AW}$
	0 (± 0 , wood products)	where h is the type of biomass; WC_A is the consumption of water in irrigation; PR_{AW} is the price of agricultural water.
	37.8 ($\pm 20\%$, energy crops)	For WC_A , we adopt values of 4.44 t water (t biomass) $^{-1}$ for agricultural residues (HPBS, 2019), and 140 t water (t biomass) $^{-1}$ for energy crops (Fajardy & Mac Dowell, 2017). For PR_{AW} , we adopt a value of $\$0.27 \pm 0.05$ (t water) $^{-1}$ (Liu & Zou, 2014; Molinos-Senante & Donoso, 2016).
Water consumption due to co-firing and CCS in power plants (μ_h^w)	7.3 ($\pm 20\%$)	Unit cost of water consumption due to co-firing and CCS in power plants is calculated as: $\mu_h^w = WC_P \cdot PR_{IW} \cdot \eta \cdot \lambda_h / 3.6$
		where h is the type of biomass; WC_P is the consumption of water in power plant; PR_{IW} is the price of industrial water (lower than the household price in China); η is the power generation efficiency in power plant; λ_h is the heat content of biomass; and 3.6 converts 1 MWh to GJ. For WC_P , we adopt a value of 4.4 t water (MWh electricity) $^{-1}$ (Fajardy & Mac Dowell, 2017). For PR_{IW} , we adopt a value of $\$1.25 \pm 0.25$ (t water) $^{-1}$ (Liu & Zou, 2014; Molinos-Senante & Donoso, 2016). For η , we adopt a value of 25.1% (Yang et al., 2019). For λ_h , we adopt a value of 19 \pm 1 GJ (t biomass) $^{-1}$ (Kumar et al., 2003).
Retrofitting power plants to be suitable for biomass	67.6 ($\pm 10\%$)	Unit cost of retrofitting power plants to be suitable for biomass co-firing and CCS is calculated as:

co-firing and
CCS in
power plants
(μ_h^d)

$$\mu_h^d = \left(\frac{P \cdot CRF + FOM}{FRT \cdot CF} \cdot 1000 + VOM \right) \cdot \eta \cdot \lambda_h / 3.6$$

where h is the type of biomass; P is the investment cost of facilities in power plants (including the co-firing and CCS system); CRF is the capital recovery factor; CF is the capacity factor; FOM is annualized cost of fixed operation & maintenance; VOM is annualized cost of variable operation & maintenance; FRT is the full-load running time of power plants in one year; η is the power generation efficiency in power plants; λ_h is the heat content of biomass; and 3.6 converts 1 MWh to GJ.

For P , we adopt a value of \$2432 KW⁻¹ (Black & Veatch, 2012; Lu et al., 2019). Unit cost of power plant retrofitting are calculated as a sum of the retrofitting cost for CCS (1457 (2015 US \$) KW⁻¹) from Lu et al. (2019) and the retrofitting cost for biomass co-firing (975.6 (2015 US \$) KW⁻¹) from Black & Veatch, (2012). Following an equation in Black & Veatch, (2012), the latter number (975.6) is calculated as 990 × 1.095 × 90%, where 990 is the cost given by Black & Veatch, (2012), 1.095 is the deflation rate to convert the 2009 US dollar to the 2015 US dollar (https://wenku.baidu.com/view/f27e3eeb18e8b8f67c1cfad6195f312b3169eba8.html), and 90% is the biomass co-firing ratio.

For CRF , we adopt a value of 9% (calculated by equation 9 in the main text).

For CF , we adopt a value of 80% (Lu et al., 2019).

For FOM , we adopt a value of \$58.2 KW⁻¹ yr⁻¹ (Black & Veatch, 2012).

For VOM , we adopt a value of \$6.6 MWh⁻¹ (Black & Veatch, 2012).

For FRT , we adopt a value of 7800 h (Koornneef et al., 2012).

For η , we adopt a value of 25.1% (Yang et al., 2019).

For λ_h , we adopt a value of 19±1 GJ (t biomass)⁻¹ (Kumar et al., 2003).

Unit cost of CCS in power plants is calculated as:

$$\mu_h^{cap} = PR_{CCS} \cdot CC_h \cdot EC \cdot 3.67$$

where h is the type of biomass; PR_{CCS} is the price of CO₂ capture and storage; CC_h is carbon content in biomass; EC is the efficiency of CO₂ capture; 3.67 is the parameter convert C to CO₂.

For PR_{CCS} , we adopt a value of \$53±7 (t CO₂)⁻¹ (Koornneef et al., 2012; Rubin et al., 2015).

For CC_h , we adopt values of 47.1% for agricultural residues and energy crops (Zhang et al., 2015; Fajardy & Mac Dowell, 2017), and 47% for wood products (IPCC, 2006).

For EC , we adopt a constant value of 90% (Anderson & Peters, 2016; Lu et al., 2019).

CO₂ capture
and storage
(μ_h^{cap})

82.5 (±10%,
agricultural
residues and
energy crops)

82.3 (±10%,
wood products)

A new **Table S3** reads as:

Table S3. Unit emissions of BECCS in the central scenario.

B90-2015-PC is our central scenario, where “B90” stands for 90% biomass co-firing with coal, “2015” stands for generating electricity in existing power plants in 2015 (total electricity generation is 4.24 PWh yr⁻¹), and “PC” stands for electricity generation in pulverized-coal (PC) plants. Uncertainty is given as a percentage relative to the central value.

Emission items	Unit emission (t CO ₂ (t biomass) ⁻¹) and its uncertainty	Methods
----------------	--	---------

		Unit emission of biomass pretreatment is calculated as:
		$r_h^p = DEC_pre_h \cdot EF_DE / LHV_DE + ELC_pre_h \cdot \varphi / 1000 / 3.6$
		where h is the type of biomass; DEC_pre is the consumption of fuel energy (using diesel) in biomass pretreatment; LHV_DE is the lower heating value of diesel; ELC_pre is the consumption of electricity in biomass pretreatment; EF_DE is the equivalent CO ₂ emission factor of diesel; φ is the equivalent CO ₂ emission factor of coal in producing electricity.
Biomass pretreatment (r_h^p)	0.22 (±20%, agricultural residues)	For DEC_pre , we adopt values of 7.4 MJ (t biomass) ⁻¹ for agricultural residues, 0 MJ (t biomass) ⁻¹ for wood products, and 108 MJ (t biomass) ⁻¹ for energy crops (Fajardy & Mac Dowell, 2017).
	0.20 (±10%, wood products)	For LHV_DE , we adopt a value of 37.4±1.7 MJ (Litter diesel) ⁻¹ (Fajardy & Mac Dowell, 2017).
	0.26 (±40%, energy crops)	For EF_DE , we adopt a value of 0.0034±0.0001 t CO ₂ -eq (Litter diesel) ⁻¹ (Fajardy & Mac Dowell, 2017).
		For ELC_pre , we adopt values of 911±166 MJ (t biomass) ⁻¹ for agricultural residues, 828 MJ (t biomass) ⁻¹ for wood products, and 1048±403 MJ (t biomass) ⁻¹ for energy crops (Fajardy & Mac Dowell, 2017).
		For φ , we adopt a value of 0.85±0.10 t CO ₂ -eq (MWh electricity) ⁻¹ (Brander et al., 2011).

		Unit emission of fertilizer production is calculated as:
		$r_h^{zp} = \sum_{m=1}^3 C_fer_{mh} \cdot EF_fp_m$
		where m is the type of fertilizer (1, 2, 3 for N, P ₂ O ₅ and K ₂ O fertilizers, respectively); h is the type of biomass; C_fer is the consumption of fertilizer to sustain the element balance in soils when biomass is taken away from the land for combustion in power plants; EF_fp is the equivalent CO ₂ emission factor to produce one ton of fertilizer.
Production of fertilizers (r_h^{zp})	0.08 (±60%, agricultural residue)	For C_fer , we adopt values of 0.0069±0.0004 t N (t crop) ⁻¹ , 0.0026±0.0005 t P ₂ O ₅ (t crop) ⁻¹ , and 0.0128±0.0022 t K ₂ O (t crop) ⁻¹ for agricultural residues (Ren et al., 2019);
	0 (±0%, wood products)	0.0131±0.0090 t N (t energy crop) ⁻¹ , 0.0112±0.0072 t P ₂ O ₅ (t energy crop) ⁻¹ and 0.0247±0.0092 t K ₂ O (t energy crop) ⁻¹ for energy crops (Lewandowski et al., 1995).
	0.18 (±60%, energy crops)	For EF_fp , we adopt values of 8.67±3.08 t CO ₂ -eq (t N) ⁻¹ for N, 3.21±2.26 t CO ₂ -eq (t P ₂ O ₅) ⁻¹ for P ₂ O ₅ and 1.16±0.43 t CO ₂ -eq (t K ₂ O) ⁻¹ for K ₂ O (Chen et al., 2015).

		Unit emission of fertilizer application in agriculture is calculated as:
		$r_h^{za} = \sum_{m=1}^3 C_fer_{mh} \cdot EF_fa_m$
		where m is the type of fertilizer (1, 2, 3 for N, P ₂ O ₅ and K ₂ O fertilizers, respectively); h is the type of biomass; C_fer is the consumption of fertilizer to sustain the element balance in soils when biomass is taken away from the land for combustion in power plants; EF_fa is the equivalent CO ₂ emission factor when one ton of fertilizer is added in soils.
Application of fertilizers in agricultural lands (r_h^{za})	0.04 (±30%, agricultural residues)	For C_fer , we adopt values of 0.0069±0.0004 t N (t crop) ⁻¹ , 0.0026±0.0005 t P ₂ O ₅ (t crop) ⁻¹ and 0.0128±0.0022 t K ₂ O (t crop) ⁻¹ for agricultural residues (Ren et al., 2019);
	0 (±0%, wood products)	0.0131±0.0090 t N (t energy crop) ⁻¹ , 0.0112±0.0072 t P ₂ O ₅ (t energy crop) ⁻¹ and 0.0247±0.0092 t K ₂ O (t energy crop) ⁻¹ for energy crops (Lewandowski et al., 1995).
	0.08 (±50%, energy crops)	For EF_fa , we adopt values of 3.60±0.80 t CO ₂ -eq (t N) ⁻¹ for N, 1.10±0.50 t CO ₂ -eq (t P ₂ O ₅) ⁻¹ for P ₂ O ₅ and 0.64±0.22 t CO ₂ -eq (t K ₂ O) ⁻¹ for K ₂ O (Fajardy & Mac Dowell, 2017).

		Unit emission of biomass transport from the harvest sites to the power plants by vehicles (consuming diesel) is calculated as:
		$r_{ix}^b = BTD_{ix} \cdot BEF$
		where i denotes a county; x is the case of biomass transport (within a county, between counties in one province, and between different provinces); BTD is the biomass transport distance (km), which is described in the section “Constraints on biomass supply, electricity generation and carbon storage” in Methods of the main text; BEF is the equivalent CO ₂ emission factor to transport one ton of biomass over 1 km by vehicles (using diesel).
Transport of biomass from collection sites to power plants by diesel vehicles (r_{ix}^b)	0.0001-0.02 (±5%, within a county)	For BEF , we adopt a value of 0.000077±0.000003 t CO ₂ -eq (t biomass) ⁻¹ km ⁻¹ (Fajardy & Mac Dowell, 2017).
	0.004-0.06 (±5%, between counties in one province)	
	0.13 (±5%, between different provinces)	

Retrofitting power plants in power plants to be suitable for biomass co-firing and CCS (r_h^d)	0.0004 ($\pm 10\%$)	Unit emission of retrofitting power plants to be suitable for biomass co-firing and CCS is calculated based on the costs to produce 1 MWh electricity and the electricity generation from bioenergy as: $r_h^d = EF_adj \cdot \eta \cdot \lambda_h / 3.6$ where h is the type of biomass; EF_adj is the equivalent CO₂ emissions to produce 1 MWh electricity in the adjustment of facilities; η is the power generation efficiency in power plant; λ_h is the heat content of biomass; and 3.6 converts 1 MWh to GJ. For EF_adj, we adopt a value of 0.00033 t CO_{2-eq} (MWh electricity)⁻¹ (Lu et al., 2019). For η, we adopt a value of 25.1% (Yang et al., 2019). For λ_h, we adopt a value of 19±1 GJ (t biomass)⁻¹ (Kumar et al., 2003).
Land use and land cover change due to growing energy crops in the marginal lands and grasslands (r_h^{uc})	0 ($\pm 0\%$, agricultural residues and wood products) 0 ($\pm 10\%$, energy crops in marginal lands) 0.52 ($\pm 10\%$, energy crops in grasslands)	Unit emission of land use and land cover change due to growing energy crops is calculated as: $r_h^{uc} = LUC_h \cdot CC_h \cdot 3.67$ where h is the type of biomass; LUC_h is the percentage of equivalent CO₂ emissions due to land use and land cover change in the transition of marginal land or grassland to land growing dedicated energy crops relative to the CO₂ sequestration in biomass; CC_h is the carbon content in biomass; 3.67 (=44/12) converts carbon to CO₂. For LUC, we adopt values of 0 for agricultural residues and wood products; 0 for energy crops growing in marginal lands; and 30% for energy crops growing in grasslands (Fajardy & Mac Dowell, 2017). For CC_h, we adopt values of 47.1% for agricultural residues and energy crops (Zhang et al., 2015; Fajardy & Mac Dowell, 2017), and 47% for wood products (IPCC, 2006).
Sequestration of CO ₂ from biomass (r_h^s)	1.6 ($\pm 6\%$)	Unit emission of CO₂ sequestration from biomass is calculated as: $r_h^s = CC_h \cdot EC \cdot 3.67$ where h is the type of biomass; CC_h is the carbon content in biomass; EC is the efficiency of CO₂ capture; 3.67 (=44/12) converts carbon to CO₂. For CC_h, we adopt values of 47.1% for agricultural residues and energy crops (Zhang et al., 2015; Fajardy & Mac Dowell, 2017), and 47% for wood products (IPCC, 2006). For EC, we adopt a value of 90% (Anderson & Peters, 2016; Lu et al., 2019).

Please also see a new **Table S4** for configuration of other scenarios in our response to **Comment B17**.

Comment B30

Why do not you use China data for power plants? It seems that most of the values are case studies (e.g. Fajardy & MacDowell 2017). I suppose, China has a fleet of coal power plants with variety of efficiencies. This is something that affects the results notably and should be discussed, at least.

Response

Thank you for the good suggestion. We have revised our model by updating the power generation efficiency measured in China's pulverized-coal (PC) power plants, which decreases from 39.3±1.0% for coal combustion without CCS to 36.2% for co-firing 90% of biomass without CCS, 27.3% for co-firing 30% of biomass with CCS, and 25.1% for co-firing 90% of biomass with CCS (Xu et al., 2013; Yang et al., 2019). For integrated gasification combined cycle (IGCC) plants, we adopted an power generation efficiency of 35.8% from the study of Lu et al. (2019). The following sentences were added on lines 788-794 in the track version as: "*As a result of biomass co-firing, the power generation efficiency in China's PC power plants decreases from 39.3% for coal combustion to 36.2% for 90% biomass co-firing without CCS, 27.3% for co-firing 30% of biomass with CCS, and 25.1% for co-firing 90% of biomass with CCS^{41,76}. For IGCC plants, we adopted a power generation efficiency of 35.8% from a recent study for China¹¹. More studies are needed to examine the feasibility of a 90% biomass co-firing ratio and explore other technologies that can improve the power generation efficiency.*".

References:

- Xu, G., et al. Analysis and optimization of CO₂ capture in an existing coal-fired power plant in China. *Energy* **58**, 117-127 (2013).
- Yang, B., Wei, Y. M., Hou, Y., Li, H., & Wang, P. Life cycle environmental impact assessment of fuel mix-based biomass co-firing plants with CO₂ capture and storage. *Appl. Energy* **252**, 113483 (2019).

Lu, X. et al. Gasification of coal and biomass as a net carbon-negative power source for environment-friendly electricity generation in China. *Proc. Natl. Acad. Sci. U.S.A.* **116**, 8206-8213 (2019).

Comment B31

Investment costs? In Black & Veatch the numbers seem to be higher than yours. Is this just because of inflation or didn't I find correct numbers? There is no investment decision in your optimization problem. How is the changing utilization rate taken into account in the unit costs?

Response

Sorry for this confusion. The investment cost (2432 (2015 US \$) KW^{-1}) is calculated as a sum of “975.6 (2015 US \$) KW^{-1} ” spent on retrofitting power plants for biomass and coal co-firing from Black & Veatch, (2012) and “1457 (2015 US \$) KW^{-1} ” spent on retrofitting power plants for CCS from Lu et al. (2019). Here, “975.6 (2015 US \$) KW^{-1} ” is calculated as $975.6 = 990 \times 1.095 \times 90\%$, where “990 (2009 US \$) KW^{-1} ” is the number given in Black & Veatch, (2012), 1.095 is a deflation rate to convert the 2009 US dollar to the 2015 US dollar (<https://wenku.baidu.com/view/f27e3eeb18e8b8f67c1cfad6195f312b3169eba8.html>), and 90% is biomass co-firing ratio. For this function, we have followed an approach recommended by Black & Veatch, (2012):

[Redacted]

To clarify the sources of these data, the following sentences were added in the new **Table S2** in Supporting Information: “*Unit cost of power plant retrofitting are calculated as a sum of the retrofitting cost for CCS (1457 (2015 US \$) KW^{-1}) from Lu et al. (2019) and the retrofitting cost for biomass co-firing (975.6 (2015 US \$) KW^{-1}) from Black & Veatch, (2012). Following an equation in Black & Veatch, (2012), the latter number (975.6) is calculated as $990 \times 1.095 \times 90\%$, where 990 is the cost given by Black & Veatch, (2012), 1.095 is the deflation rate to convert the 2009 US dollar to the 2015 US dollar (<https://wenku.baidu.com/view/f27e3eeb18e8b8f67c1cfad6195f312b3169eba8.html>), and 90% is the biomass co-firing ratio.*”.

We have revised our decision variable (the biomass consumption by county, T_{ixyh}) and the optimization formulation, following the good suggestion of Reviewer 2 (please refer to our response to **Comment B6**). In our new optimization formulation (see a new **equation 1A**), the utilization rate is taken into account as a product of T_{ixyh} and unit cost of retrofitting power plants for co-firing and CCS (see μ_h^d in the new **equation 3**). Then, unit cost (μ_h^d , that is $\$67.6 \text{ (t biomass)}^{-1}$) is calculated as a product of the cost of facilities to generated 1 kWh of electricity and the electricity generated from one ton of biomass, which is now described in the new **Table S2** as:

Retrofitting power plants to be suitable for biomass co-firing and CCS in power plants (μ_h^d) [⊖]	67.6 (±10%) [⊖]	Unit cost of retrofitting power plants to be suitable for biomass co-firing and CCS is calculated as: [⊖] $\mu_h^d = \left(\frac{P \cdot CRF + FOM}{FRT \cdot CF} \cdot 1000 + VOM \right) \cdot \eta \cdot \lambda_h / 3.6$ where h is the type of biomass; P is the investment cost of facilities in power plants (including the co-firing and CCS system); CRF is the capital recovery factor; CF is the capacity factor; FOM is annualized cost of fixed operation & maintenance; VOM is annualized cost of variable operation & maintenance; FRT is the full-load running time of power plants in one year; η is the power generation efficiency in power plants; λ_h is the heat content of biomass; and 3.6 converts 1 MWh to GJ. [⊖] For P , we adopt a value of \$2432 KW ⁻¹ (Black & Veatch, 2012; Lu et al., 2019). Unit cost of power plant retrofitting are calculated as a sum of the retrofitting cost for CCS (1457 (2015 US \$) KW ⁻¹) from Lu et al. (2019) and the retrofitting cost for biomass co-firing (975.6 (2015 US \$) KW ⁻¹) from Black & Veatch, (2012). Following an equation in Black & Veatch, (2012), the latter number (975.6) is calculated as 990 × 1.095 × 90%, where 990 is the cost given by Black & Veatch, (2012), 1.095 is the deflation rate to convert the 2009 US dollar to the 2015 US dollar (https://wenku.baidu.com/view/f27e3eeb18e8b8f67c1cfad6195f312b3169eba8.html), and 90% is the biomass co-firing ratio. [⊖] For CRF , we adopt a value of 9% (calculated by equation 9 in the main text). [⊖] For CF , we adopt a value of 80% (Lu et al., 2019). [⊖] For FOM , we adopt a value of \$58.2 KW ⁻¹ yr ⁻¹ (Black & Veatch, 2012). [⊖] For VOM , we adopt a value of \$6.6 MWh ⁻¹ (Black & Veatch, 2012). [⊖] For FRT , we adopt a value of 7800 h (Koornneef et al., 2012). [⊖] For η , we adopt a value of 25.1% (Yang et al., 2019). [⊖] For λ_h , we adopt a value of 19±1 GJ (t biomass) ⁻¹ (Kumar et al., 2003). [⊖]
---	-----------------------------	---

References:

Lu, X. et al. Gasification of coal and biomass as a net carbon-negative power source for environment-friendly electricity generation in China. *Proc. Natl. Acad. Sci. U.S.A.* **116**, 8206-8213 (2019).

Black & Veatch. Cost and Performance Data for Power Generation Technologies. <https://www.bv.com/docs/reports-studies/nrel-cost-report.pdf> (2012).

Comment B32

On row 104 you apply 90% efficiency for CCS but on row 520 you use 40%. Which is a correct number?

Response

Sorry for this confusion. “90% efficiency of CCS” indicates that 90% of CO₂ emitted from biomass combustion is captured in power plants, while “40% of CO₂ emitted from power plants” indicates that pipelines are constructed to transport 40% of CO₂ emitted from the power plants in all counties. We have now clarified it and, to explain why we adopt the fraction of 40%, the following sentences were revised on lines 888-892 in the track version as: “In equation 8, the unit cost is extremely high when the CO₂ flow rate is low (unit cost is infinite if χ is close to zero). To estimate the potential of BECCS, we considered that pipelines are constructed to transport a given amount of CO₂, that is 40% of total emissions from all power plants in 2030 according to a recent study for China⁸⁴, which does not depend on the deployment of BECCS.”.

To highlight the limitation of adopting “40% of CO₂ emitted from power plants”, the following sentences were added on lines 928-931 in the track version as: “There are three limitations. First, the marginal cost curve of BECCS depends on the national target of CCS. As shown in Fig. S10, if the pipelines are designed to transport 20%, rather than 40%, of CO₂ emitted from all power plants in our central scenario, the marginal cost of BECCS will increase by 3.4% to abate 0-5 Gt CO_{2-eq} yr⁻¹ of national net emissions.”.

We have examined the sensitivity of our estimated marginal cost of BECCS to the assumed “40% of CO₂ emitted from power plants” in a new Fig. S10, which is now added to the Supporting Information.

A new Fig. S10 reads as:

Figure S10. Sensitive of BECCS marginal cost to percentage of CO₂ emitted from all power plants that are transported by pipelines.

Relative to the central scenario (B90-2015-PC), we consider that the pipelines of CO₂ are alternatively used to transport 20% (red line), rather than 40% (blue line), of CO₂ emitted from all power plants. As a result, it reduces CO₂ flow rate in the pipelines, increases unit cost of CO₂ transport (see equation 8 in the main text for unit cost of CO₂ transport as a function of CO₂ flow rate), and produces a higher marginal cost of emission reduction by BECCS in our study. It means that the marginal cost of BECCS can be reduced (or increased) under a more (or less) ambitious target of CCS over the whole country.

Comment B33

6. The scale of the operation

To me the BECCS operation seems to be a massive national project. I was wondering what the province level economic effects would be. How much labor would be needed? For example, the labor costs of energy crops are five times those of agricultural crops (Table S5). Does this mean that five times more labor is needed?

Response

Thank you for the good suggestion. Sorry for that we do not have a method yet to directly estimate “*How much labor would be needed*”. When “*the labor costs of energy crops are five times those of agricultural crops*”, the required labor forces also depend on the type of labor (human forces or machines). Instead, we estimated the costs of labor forces used to produce the biomass, which may provide new sources of income for farmers.

We added a new **Table S5** to present the bioenergy, costs and abated emissions of BECCS by province, and a new **Table S6** to show the fraction of these costs due to labor forces. Accordingly, the following sentences were added on lines 420-432 in the main text of the track version as: “*At the provincial level, Shandong province is the largest contributor to abate net emissions of 1 Gt CO₂-eq yr⁻¹, providing bioenergy of 1.15 EJ yr⁻¹ at a cost of \$8.4 billion yr⁻¹, followed by Henan province (1.07 EJ yr⁻¹), Jiangsu province (0.76 EJ yr⁻¹) and Hebei province (0.69 EJ yr⁻¹) (see the potential and costs of BECCS by province in **Table S5**). In contrast, Sichuan province is the largest contributor of bioenergy (5.41 EJ yr⁻¹) when the goal is to abate net emissions of 5 Gt CO₂-eq yr⁻¹, followed by Yunnan province (3.49 EJ yr⁻¹), Qinghai province (3.18 EJ yr⁻¹) and Tibet province (2.66 EJ yr⁻¹). In China, total costs to abate 1, 3 and 5 Gt CO₂-eq yr⁻¹ amounts to \$58, 275 and 796 billion yr⁻¹, of which 22%, 34% and 32% are spent on biomass acquisition and 9%, 10% and 12% cover labor costs, respectively; these spendings provide new sources of income to farmers over China’s rural areas (see **Table S6** for the fraction of acquisition and labor costs in total costs).*”.

A new **Table S5** reads as:

Table S5. Provincial bioenergy, costs and abated emissions by BECCS.

Provinces	Bioenergy (EJ yr ⁻¹)	Costs (billion US \$ yr ⁻¹)	Abated emissions (Gt CO ₂ -eq yr ⁻¹)
-----------	----------------------------------	---	---

	1Gt	3Gt	5Gt	1Gt	3Gt	5Gt	1Gt	3Gt	5Gt
Beijing	0.024	0.060	0.060	0.175	0.717	0.719	0.003	0.006	0.006
Chongqing	0.199	0.400	0.854	1.456	3.923	14.879	0.025	0.047	0.084
Tianjin	0.060	0.071	0.071	0.426	0.603	0.604	0.007	0.009	0.009
Anhui	0.534	1.193	1.575	3.929	11.449	17.707	0.067	0.144	0.177
Fujian	0.114	0.798	1.549	0.816	11.610	25.405	0.015	0.093	0.157
Guizhou	0.296	0.678	1.599	2.174	7.031	25.805	0.038	0.084	0.161
Hebei	0.694	1.243	1.783	5.084	11.309	20.227	0.087	0.149	0.195
Heilongjiang	0.166	1.709	2.324	1.234	19.900	34.959	0.021	0.207	0.264
Henan	1.071	1.936	2.088	7.907	16.488	19.025	0.134	0.235	0.249
Hubei	0.414	0.880	1.093	3.078	7.835	12.961	0.052	0.109	0.126
Hunan	0.227	0.765	1.191	1.701	7.106	17.591	0.029	0.095	0.133
Jiangsu	0.762	1.437	1.437	5.598	17.307	17.307	0.095	0.180	0.180
Jiangxi	0.123	0.695	0.967	0.906	7.154	13.057	0.016	0.084	0.106
Liaoning	0.375	0.769	0.932	2.738	6.735	9.421	0.047	0.094	0.108
Shaanxi	0.228	1.137	2.095	1.674	14.620	33.260	0.029	0.123	0.204
Shandong	1.145	2.096	2.227	8.418	23.382	25.571	0.143	0.255	0.266
Shanxi	0.280	1.095	1.362	2.047	14.219	18.612	0.035	0.122	0.145
Sichuan	0.179	0.952	5.407	1.327	11.806	120.279	0.023	0.116	0.495
Zhejiang	0.133	0.433	0.433	0.973	5.736	5.742	0.017	0.050	0.050
Hainan	0.034	0.182	0.182	0.243	2.405	2.406	0.004	0.019	0.019
Guangdong	0.293	1.187	1.788	2.125	17.468	28.009	0.038	0.152	0.204
Gansu	0.057	0.655	1.133	0.425	8.378	18.645	0.007	0.075	0.115
Guangxi	0.009	0.579	1.810	0.065	6.239	36.957	0.001	0.073	0.189
Jilin	0.202	1.031	1.429	1.505	9.295	19.032	0.025	0.127	0.166
Ningxia	0.064	0.201	0.201	0.467	2.453	2.461	0.008	0.023	0.023
Qinghai	0.000	0.159	3.180	0.002	2.650	76.091	0.000	0.018	0.306
Yunnan	0.014	0.574	3.491	0.101	7.389	78.484	0.002	0.070	0.314
Xinjiang	0.116	0.660	0.660	0.853	5.826	5.846	0.015	0.082	0.082
Tibet	0.000	0.013	2.662	0.000	0.209	64.715	0.000	0.002	0.233
Inner Mongolia	0.111	1.268	2.170	0.829	13.816	30.393	0.014	0.155	0.232
Shanghai	0.020	0.031	0.031	0.148	0.310	0.311	0.003	0.004	0.004
Total	7.945	24.886	47.784	58.425	275.369	796.482	1.000	3.000	5.000

, and a new **Table S6** reads as:

Table S6. Acquisition and labor costs of agricultural residues, wood products and energy crops.

Abated emissions (Gt CO ₂ -eq)	Total costs (billion US \$ yr ⁻¹)	Acquisition costs (billion US \$ yr ⁻¹)				Fraction of biomass acquisition	Labor costs (billion US \$ yr ⁻¹)			Fraction of labor costs
		Agricultural	Fuelwood	Commercial	Energy		Agricultural	Fuelwood	Energy	

yr ⁻¹)	US \$ yr ⁻¹) residues			wood	crops	costs	residues		crops	
1	58	10	3	0	0	22%	4	1	0	9%
3	275	25	4	31	33	34%	10	2	15	10%
5	796	25	4	40	186	32%	10	2	86	12%

Comment B34

Smaller comments

R75: List could include costs (fuel costs and reduced efficiency of power plants due to CCS and co-firing).

Response

According to this comment, the sentence on lines 81-88 in the main text of the track version was revised as: *“Developing the value chain for BECCS implies addressing the following challenges: land and biomass availability¹², costs of biomass acquisition and pretreatment¹¹, requirements for water and fertilizer¹³, associated GHG emissions¹⁴, investment to make power plants suitable for biomass co-firing with coal in the case of China¹⁵, carbon capture and storage (CCS)¹³, CO₂ transport from power plants to repositories for storage¹⁶, reduced power generation efficiency in power plants due to CCS and co-firing¹⁷, inertia of energy system⁹, and public perception of carbon removal¹⁸.”*

Comment B35

R78: I suppose, the link between Table S1 and the preceding text could be made clearer.

Response

To make the link between Table S1 and the main text clearer, the following sentences was added on lines 89-92 in the main text of the track version as: *“Studies that have investigated the potential and marginal cost of BECCS at a regional scale are listed in **Table S1**. Spatially explicit method has been applied for several countries^{19,20}, but there is, to our knowledge, only one study that estimates marginal cost curves of BECCS over western North America⁴.”*

Comment B36

R124: This 0.9 Gt/yr is little surprising after the fact that only 0.36 Gt/yr can be combusted and storage in a same county. Reader could be helped by mentioning that now you do not consider the CO₂ transportation limit.

Response

Thank you for the good suggestion. To clarify that we do not consider the CO₂ transportation limit in this figure, the sentence on lines 210-214 in the track version was revised as: *“In total, there are 222 GW of existing power capacities, which can be generated by co-firing 0.9 Gt yr⁻¹ of biomass within the same county. This permits to reduce 1.0±0.1 Gt CO₂ yr⁻¹ from coal emissions and remove 1.4±0.1 Gt CO₂ yr⁻¹ from the atmosphere by transporting over different distances the captured CO₂.”*

Comment B37

Figure 1, panel D: Storage for 30 years? Is it then released? Why not longer timescales? Based on Figure S7, this should not make such a difference, but 30 years looks quite bad from the climate point of view.

Response

We adopt a timescale of 30 years for storage of carbon in a near term to avoid greater uncertainty in the longer future. When the full capacity of storage is used up over 30 years, we assume that these repositories are no longer used for further storage and the carbon will be kept in the repositories. To make our assumption clearer, the sentence on lines 719-724 in the track version was revised as: *“We assumed a lifetime of 30 years for reservoirs to match medium-term mitigation target and avoid uncertainties in the longer future — the choice of this parameter slightly affects the constraints on carbon storage in counties where bioenergy is produced (see a sensitivity test in **Fig. S6**). We considered that these repositories are no longer used for further storage if it reaches the full capacity of storage after 30 years, and the stored carbon cannot be released.”*

Comment B38

R143: I do not understand the reference to carbon price.

Response

Sorry for the confusion. To clarify the reference to this carbon price, the following sentence on lines 244-248 in the track version was revised as: “*To our knowledge, there is only one study that has estimated the marginal cost curve for carbon emission reduction using BECCS over western North America, where the marginal cost is $\sim \$100$ ($t\ CO_{2-eq}$)⁻¹ to abate $0.1\ Gt\ CO_{2-eq}\ yr^{-1}$ by BECCS⁴; in comparison, a marginal cost of $\$100$ ($t\ CO_{2-eq}$)⁻¹ abates $2.0_{-0.6}^{+0.9}$ $Gt\ CO_{2-eq}\ yr^{-1}$ in China as shown in Fig. 3.*”.

Comment B39

R 146: “... abatement could be offset by ...” I do not understand. Do they offset? What is the idea here?

Response

Sorry for the confusion. We supposed to say that the emission reduction by replacing coal and sequestering carbon in biomass are **partly** offset by life-cycle emissions. To make it clearer, the sentence on lines 359-361 in the track version was revised as: “... *the above two emission reductions of $2.28_{-0.85}^{+0.80}$ $Gt\ CO_{2-eq}\ yr^{-1}$ will be partly offset by $0.28_{-0.05}^{+0}$ $Gt\ CO_{2-eq}\ yr^{-1}$ due to life-cycle emissions from retrofitting power plants for...*”.

Comment B40

R151: Perhaps, the CCS does not “dominate”, when the marginal biofuel acquisition and pretreatment costs are almost the same...

Response

Sorry for the confusion. To clarify it, the sentence on lines 364-371 in the track version was revised as: “*the marginal cost ($\$92$ ($t\ CO_{2-eq}$)⁻¹) decomposes into $\$36$ ($t\ CO_{2-eq}$)⁻¹ by CCS, $\$30$ ($t\ CO_{2-eq}$)⁻¹ by biomass acquisition and pretreatment, $\$29$ ($t\ CO_{2-eq}$)⁻¹ by retrofitting power plants to be suitable for biomass co-firing and CCS, $\$16$ ($t\ CO_{2-eq}$)⁻¹ by logistics of biomass transport, $\$7$ ($t\ CO_{2-eq}$)⁻¹ by water consumption for irrigation in agriculture and for CCS in power plants, $\$5$ ($t\ CO_{2-eq}$)⁻¹ by fertilizers usage, and $\$4$ ($t\ CO_{2-eq}$)⁻¹ by CO_2 transport, which is partly offset by the saving of $\$35$ ($t\ CO_{2-eq}$)⁻¹ made from the substitution of coal.*”.

Comment B41

R151 onwards: These marginal cost components sum up over one hundred that was mentioned earlier in the paragraph. You lost me here.

Response

The “*marginal cost of $\$100$ ($t\ CO_{2-eq}$)⁻¹*” is derived as a sum of all the costs to deploy BECCS (see all items in a new equation 3), which should be subtracted by the saving made from the substitution of coal. To clarify this calculation, the sentence on lines 364-371 in the track version was revised as: “*the marginal cost ($\$92$ ($t\ CO_{2-eq}$)⁻¹) decomposes into $\$36$ ($t\ CO_{2-eq}$)⁻¹ by CCS, $\$30$ ($t\ CO_{2-eq}$)⁻¹ by biomass acquisition and pretreatment,..., which is partly offset by the saving of $\$35$ ($t\ CO_{2-eq}$)⁻¹ made from the substitution of coal.*”.

Comment B42

R155: “To reduce the marginal costs while making the compromise to abate less CO_2 emissions, it is possible to focus on biofuel with lower production costs, and on shorter distances for the transport of biofuel and captured CO_2 .” To me the causality goes to other direction: By focusing on biofuel with lower production costs and on shorter distances for the transport of biofuel and captured CO_2 , the marginal costs but also the CO_2 emission abatement are reduced.

Response

We agree with Reviewer 2 that the causality should go in the suggested direction, and the sentence on lines 376-379 in the track version was revised as: “*Through optimizing biomass use at lower acquisition costs and on shorter distances for the transport of biomass and captured CO_2 , the minimal marginal cost is obtained to achieve a given target of national emission reduction.*”.

Comment B43

R158: I am sorry to say that the example does not provide a “clear illustration”. Economic efficiency is unclear for me here. Yes, there are increasing marginal costs related to BECCS. Maybe that is what you try to convey here? (Additional note: I would like to see Fig. 3S in the main text.)

Response

Thank you for the good suggestion. We agree with Reviewer 2 that this graph shows increasing marginal cost related to BECCS. Now, we move **Fig. S3** to the main text as **Fig. 4**. We avoided saying “*Economic efficiency*”, and the following sentences on lines 379-383 in the track version were revised as: “*There is an increasing marginal cost to abate more CO₂ emissions by BECCS. As shown in Fig. 4, the total cost grows in a concave shape as the target of emission reduction rises, where dedicated energy crops take up a larger fraction in bioenergy supply with higher costs in biomass acquisition and increasing CO₂ emissions from land-use change.*”.

Comment B44

R167: This 30% possibility comes from out of the blue. It should be mentioned in the introductory paragraphs. Also, you could explain the reader why the efficiency depends on the co-fire share of biomass.

Response

Thank you for the good suggestion. In the Introductory paragraph, to introduce this “30%”, we added one sentence on lines 120-123 as: “*Furthermore, in order to improve the power generation efficiency, we follow a recent study to consider an advanced retrofit of power plants to an integrated gasification combined cycle (IGCC) system¹¹. Alternatively, we test the results for adopting a lower biomass co-firing ratio (30% weight)¹¹.*”. To explain how we consider the “30%”, we added the following sentence on lines 293-295 in the **Sensitivity Analyses** section: “*To address the sources of uncertainties, we design seven scenarios by varying parameters (see parameterization of each scenario in **Table S4**) relative to the central scenario, including B30-2015-PC (“B30” for 30% biomass co-firing with coal)...*”.

To explain why the efficiency depends on the co-fire share of biomass, the following sentence was added on lines 175-179 in the main text as: “*However, the power generation efficiency declines as a result of biomass co-firing, since a high moisture and a fibrous structure of biomass reduce the stability of combustion flame⁴⁴, while it consumes more energy to break the oxygenated chemical bonds in the gasification of biomass¹¹.*”. To explain the data source for the reduced power generation efficiency, the following sentence was added on lines 788-791 in the **Methods** section as: “*As a result of biomass co-firing, the power generation efficiency in China’s PC power plants decreases from 39.3% for coal combustion to 36.2% for 90% biomass co-firing without CCS, 27.3% for co-firing 30% of biomass with CCS, and 25.1% for co-firing 90% of biomass with CCS^{41,76}.*”

Comment B45

R168: Can you really retrofit a coal power plant into IGCC or do you have to build a completely new power plant?

Response

In a recent study by Lu et al. (2019), co-firing biomass with coal was considered in an integrated gasification combined cycle (IGCC) system combined with carbon capture and storage (CBECCS); however, a spatially explicit analysis as we do was not performed in that study. To be comparable with Lu et al. (2019) and show the effect of using IGCC, we have followed the approach and used the data adopted by Lu et al. (2019) to consider the cost of transferring the pulverized-coal power plants to IGCC system. However, we notice that Lu et al. (2019) did not claim whether these power plants were rebuilt, and they did not include the cost of removing the existing pulverized-coal plants:

[Redacted]

, which are suggested to be high in another study (Bui et al., 2018).

To make our assumption and the limitation clearer, we added the following sentence on lines 314-318 in the track version of the main text as: “*The marginal cost will increase to \$146 (t CO_{2-eq})⁻¹ due to a high cost of plant retrofitting, if as in Lu et al.¹¹, we consider biomass co-firing in the IGCC system (the scenario B90-2015-IGCC); in this scenario, the marginal cost could be even higher if rebuilding new plants is required when shifting the pulverized-coal boilers to IGCC^{54,55}.*”.

References:

Lu, X. et al. Gasification of coal and biomass as a net carbon-negative power source for environment-friendly electricity generation in China. *Proc. Natl. Acad. Sci. U.S.A.* **116**, 8206-8213 (2019).

Bui, M. et al. Carbon capture and storage (CCS): the way forward. *Energy Environ. Sci.* **11**, 1062-1176 (2018).

Comment B46

R170: “The marginal costs are less sensitive to the capacity of electricity generation and the yield of energy crops when the target is to abate less than 3 Gt CO_{2-eq} yr⁻¹.” I cannot understand the reference here. The previous sentences are disconnected (0.88 Gt abatement).

Response

Sorry for the confusion. We supposed to compare the costs of BECCS in the central scenario with those in the two scenarios using different types of energy crops and generating different electricity in power plants. To make it clearer, the sentence on lines 318-321 in the track version was revised as: “*In the absence of agricultural and forestry residues, the marginal cost increases remarkably to \$202 (t CO_{2-eq})⁻¹ (the scenario B30-2015-PC-EneCrop); by contrast, shifting the energy crops from Miscanthus, as the only source of energy crops in the central scenario, to the best-yield crops²⁴ alters the marginal cost moderately (the scenario B90-2015-PC-BestCrop).*”.

Comment B47

Figure 2: What determines the limit for nuclear energy capacity?

Response

Sorry for this confusion. We did not model the limit for nuclear energy capacity; instead, we supposed to show the CO₂ emission reduction when nuclear energy substitutes coal for electricity generation in 2015. To make it clearer, the following sentence in the caption of Fig. 3 was revised as: “*The national CO₂ emission reduction (calculated from the substituted coal consumption⁴² and the CO₂ emission factor of coal⁵³) by hydropower, wind, solar and nuclear power in China in 2015 as well as the range of their marginal costs (calculated from their prices of electricity generation in different provinces⁵² and the CO₂ emission factor of coal⁵³) are shown as vertical lines.*”.

Comment B48

R203: Consider changing “water” to “hydro”.

Response

According to this comment, “*water*” was revised to “*hydro*” in our paper.

Comment B49

R207: 2.3 is 19% of 12.2 not 30%. Please, correct something.

Response

Sorry for the confusion. “*2.3*” and “*12.2*” are for two different years, and “*2.3*” is 30% of “*7.7*”. 7.7 is revised as 7.5, and 12.2 is revised as 9.5 according to the National Bureau of Statistics of China (NBSC, 2016), and China Energy Outlook 2030 (CERS, 2016) in our revised manuscript. To explain these percentages clearer, the sentence on lines 440-443 in the track version was revised as: “*the power capacities of WWSN are projected to grow from 2.3 PWh yr⁻¹ (contributing 30% of total electricity generation of 7.5 PWh yr⁻¹ in 2020) to 3.8 PWh yr⁻¹ (contributing 40% of total electricity generation of 9.5 PWh yr⁻¹ in 2030) over 2020–2030 (Fig. 6a).*”.

Reference:

China Energy Research Society (CERS). *China Energy Outlook 2030* (Energy & Management Press, Beijing, 2016).

National Bureau of Statistics of the People’s Republic of China (NBSC). *National data* <http://data.stats.gov.cn/> (2016).

Comment B50

R212: Are these payments made once or are they annual payments until 2030? How much will the WWSN-project cost in terms of GDP? Combined capacity of hydro and nuclear is quite high (Fig. 2). What would be the cost to abate 2 Gt CO₂eq/yr with those two technologies? That is, why you consider only BECCS and not the hydro and nuclear too.

Response

Sorry for the confusion. These investments are paid annually until 2030. To clarify it, we now give the total investment (\$2.3 and 6.8 trillion over 2021-2030) and revised the sentence on lines 446-450 in the track version as: “*Investing \$2.3 and 6.8 trillion over 2021-2030 (that is 1% and 3% of China’s GDP annually) into BECCS causes a decline to 20 and 1 Gt CO₂-eq, respectively in power sector emissions. Furthermore, investing \$7.2 trillion over 2021-2030 (that is 3.2% of China’s GDP annually) permits to reach a carbon-emission neutral power system for China (Fig. 6b).*”.

We cannot yet estimate the total costs of WWSN to abate 2 Gt CO₂eq yr⁻¹ using hydropower and nuclear power, because their marginal cost curves are not available to us. In our model, we adopted the projected capacity from the plan of the Chinese government, rather than modelling the competition of water (i.e., hydropower), wind, solar and nuclear (WWSN) to BECCS. This allows us to focus on discussing on the costs and potential of deploying BECCS, and qualitatively show the trade-off between expanding WWSN and investing in BECCS. Please refer to our response to **Comment B14**.

Comment B51

R221 “we want to keep” Do you mean that you keep in the calculation or the World is expecting that China’s power sector keeps, or that you as researcher want...

Response

We supposed to say that the World is expecting that China’s power sector will keep this contribution in the future. To make it clearer, the sentence on lines 461-464 in the track version was revised as: “*The power sector contributes 50% to China’s total CO₂ emissions¹¹, so the World may expect that cumulative emissions from the China’s power sector stay within 5% of the remaining global carbon budget⁵ to limit warming to 2 °C.*”.

Comment B52

R247: “retrofitting coal plants to co-fire with biofuel” and CCS? Or is this separate biomass case without CCS. I suppose not, as on row 250 CO₂ is stored.

Response

We proposed to say “*retrofitting plants to co-fire biomass and equipping them with CCS*”. To make it clear, the sentence on lines 496-498 in the track version was revised as: “*First, retrofitting plants to co-fire biomass and equipping them with CCS would form an early entry point to address national and global challenges.*”.

Comment B53

R266: “the latter is missing in most global integrated assessment models” I fear that this topic is missing also in this study, except for couple of citations to previous literature.

Response

Thank you for the good suggestion. Since this sentence is not relevant to our study, it is now deleted in the new version.

Comment B54

Eqs (3A-C): Consider removing unnecessary parentheses.

Response

Thank you for the good suggestion, and all unnecessary parentheses are removed.

Comment B55

R514: How are the CO₂ transport capacities (89 and 26698 kt yr⁻¹) calculated?

Response

“*89 and 26698 kt yr⁻¹*” are taken from the limits of flow rate in a previous study (Sanchez et al., 2018),

[Redacted]

and the following sentences were added on lines 903-909 in the track version to explain how these two limits are used: “*In addition, Sanchez et al. suggested that the flow rate of CO₂ transported by pipelines is limited*

by the nominal pipeline size¹⁶, and we adopted the minimal ($\chi^{MIN}=89 \text{ kt yr}^{-1}$) and maximal ($\chi^{MAX}= 26698 \text{ kt yr}^{-1}$) flow rate from that study due to lack of standards in China. Therefore, the capacity of CCS is set to be zero for a county if χ is below 89 kt yr^{-1} , so that there is no pipeline to transport CO_2 captured in power plants in this county; χ is set to 26698 kt yr^{-1} if it is above 26698 kt yr^{-1} , so that unit cost does not decrease when χ exceeds this threshold.”.

Reference:

Sanchez, D. L., Johnson, N., McCoy, S. T., Turner, P. A. & Mach, K. J. Near-term deployment of carbon capture and sequestration from biorefineries in the United States. *Proc. Natl. Acad. Sci. U.S.A.* **115**, 4875-4880 (2018).

Comment B56

Eqs 9A-C and row 529: Where do these equations come from? What is 1660? Capital Fs should be χ s? Why the exponents are 0.65 and 1.13? What kind of values does the distances L get?

Response

This equation is taken from a previous study (McCollum et al. 2006) as:

[Redacted]

“1660” is converted from “9970” as: $9970 \cdot (1/365)^{0.35} \cdot 1.313 = 1660$, where 1.313 is the deflation rate to convert the 2005 US dollar to the 2015 US dollar (<https://wenku.baidu.com/view/f27e3eeb18e8b8f67c1cfad6195f312b3169eba8.html>), and 365 is days in one year (the CO₂ flow rate is in unit of t CO₂ day⁻¹ in McCollum et al. 2006, and t CO₂ day⁻¹ in our calculation).

The coefficient “0.65” is obtained as we divided the costs by the amount of CO₂ transported in the pipelines: $1 - 0.35 = 0.65$, and “1.13” is obtained as we multiplied the cost by the distance of CO₂ transport: $0.13 + 1 = 1.13$.

To clarify the sources of data, the paragraph on lines 874-885 in the track version was revised as:

“We determined unit cost (μ_{iy}^t , \$ (t biomass)⁻¹) of pipeline constructed to transport CO₂ captured from biomass burnt in power plants using a function developed by McCollum et al.⁸¹ as:

$$\mu_{iy}^t = 1.31 \cdot 9770 \cdot L_{iy}^{0.13} \cdot (\chi_{iy}/365)^{0.35} \cdot F^L \cdot F^T \cdot (CRF + OM) \cdot \frac{L_{iy}}{\chi_{iy}} \cdot CC \cdot 3.67 \cdot EC \quad (8)$$

where 1.31 is a deflation rate to convert the 2005 US dollar to the 2015 US dollar; 365 is days in one year; 9770 is a calibrated constant⁸¹, χ_{iy} is CO₂ mass flow rate (t yr⁻¹), L_{iy} is the length of pipeline (km), F^L is a location factor, F^T is a terrain factor, CRF is capital recovery factor (yr⁻¹, see equation 9), OM is an annualized constant ratio of operational and maintenance costs relative to capital investments (0.025 yr⁻¹)⁸¹, CC is carbon content in biomass (47%)^{17,82,83}, 3.67 converts C to CO₂, and EC is efficiency of CO₂ capture (90%). For simplicity, we adopt $F^L = F^T = 1$, which can be improved when we can obtain more information on the routes of pipelines. Using a constant discount rate (r) of 7% per year¹¹, the capital recovery factor (CRF) is derived as...”

“Capital F should be χ ”. Sorry for the typo. “F” is corrected as “ χ ”.

We calculated the distance of CO₂ transport as the radii (D) of circles with areas equal to the areas of the county, province and country, $D = (A/\pi)^{1/2}$, where A is the area. To clarify our assumption in this method, the following sentences on lines 894-897 in the track version were revised as:

“We approximated the distance of CO₂ transported in county i (D_i), province j (D_j) or the country (D_0) as the radii of circle with area equal to the area of the county, province or the country, $D = (A/\pi)^{1/2}$, where A is the area...”

, and to highlight the limitation, the following paragraph was added on lines 928-935 as: *“There are three limitations. First, the marginal cost curve of BECCS depends on the national target of CCS. As shown in **Fig. S10**, if the pipelines are designed to transport 20%, rather than 40%, of CO₂ emitted from all power plants in our central scenario, the marginal cost of BECCS will increase by 3.4% to abate 0-5 Gt CO₂-eq yr⁻¹ of national net emissions. Second, our calculation should be improved by using more precisely calculated distances of CO₂ transport when accurate locations of power plants and carbon storage sites are available. Third, the impact of terrain on unit cost of CO₂ transport in pipeline is not considered in our present model, which should be improved when precise information on the routes of pipelines are obtained.”*

Reference:

McCullum, D. L., & Ogden, J. M. Techno-Economic Models for Carbon Dioxide Compression, Transport, and Storage & Correlations for Estimating Carbon Dioxide Density and Viscosity. Institute of Transportation Studies University of California-Davis. UCD-ITS-RR-06-14 (2006).

Comment B57

Figure S1: The Figure is rather difficult to read. There are two T pillars and the middle pillar are continuing beyond the cut-off, the level of which is not clear to me.

Response

Sorry for the confusion. **Figure S1** was revised with a consistent y-axis. The new **Fig. S1** reads as:

Figure S1. Carbon budget of BECCS in China.

Maximal potential of CO₂ captured from combustion of biomass, CO₂ emissions from existing coal-fired power plants, and allowable annual CO₂ flux in the proven land repositories and off-shore basins are shown for year 2015.

Comment B58

Figure S2: How can the “BECCS electricity generation” be greater than the “current fossil electricity generation”? Is there new capacity or is the capacity utilization rate very low currently? Or do you mean “biofuel-based generation potential” vs. electricity generation capacity? Then I suggest rephrasing the terms.(Note: It would be easier to compare the “amount of bioenergy” to the “maximum fuel capacity of power plants”.)

Response

Thank you for the good suggestion. We supposed to show the ratio of the “*biofuel-based generation potential*” to the “*current electricity generation capacity*”. According to the suggestion, we have revised the **Fig. S2** to show the ratio of the amount of bioenergy potential to the maximum fuel capacity of existing power plants.

The new **Fig. S2** reads as:

Figure S2. Spatial constraints on the capacity of power plants and carbon storage for BECCS.

(a) shows the ratio of the amount of bioenergy potential (Z) to the maximum fuel capacity that are supported by existing power plants (P). A ratio of $Z/P > 1$ indicates that the amount of bioenergy potential will exceed the maximum fuel capacity of existing power plants in that county. Cumulative Z (bars) and P (dots) for counties with different Z/P ratios are shown in the insert. (b) shows the ratio of the maximal potential of CO_2 captured from combustion of biomass produced in BECCS (T) to the allowable CO_2 storage at a lifetime of 30 years (S). A ratio of $T/S > 1$ indicates that the CO_2 captured from biomass will exceed the capacity of geological carbon storage in that county. Cumulative T (bars) and S (dots) for counties with different T/S ratios are shown in the insert. It shows that there is only a fraction of counties with an excellent matching of biomass feedstock, electricity generation, and geological storage capacity in space.

Comment B59

Figure S7: Difficult to read. What is going on? Time development is unnoticeable. Title of the vertical axis is confusing: How can carbon sequestration happen across counties?

Response

Sorry for the confusion. To show the impact of time for carbon storage in the reservoirs, the maximal potentials of CO_2 capture are given on each bar.

A new **Fig. S6** (**Fig. S7** in the last version) reads as:

Figure S6. Impact of carbon storage length on the spatial constraints of carbon storage.

The green bars show the maximal potential of CO_2 capture from biomass combustion with requirements of CO_2 transport from the county of biomass production to other counties for storage. The red bars show the maximal potential of CO_2 capture from biomass combustion without requirements of CO_2 transport from the

county of biomass production to other counties for storage.

Comment B60

Figure S8: Panel A shows the “monthly minimum temperature”. Should not there be 12 values for each county? Is this the minimum of monthly minimum temperatures or a minimum monthly temperature in a specific month? Unit must have some error too, or is “monthly minimum temperature” an annual mean of these minimum temperatures? In panels E and F, the measurement of biofuel/biomass in tons of CO₂ is little confusing. At least it needs clarification that you want to operate with “carbon sink” units.

Response

Sorry for the confusion. “*monthly minimum temperature*” was revised to “*minimum of monthly average temperature in 2015*”, and the unit is confirmed to be “°C”.

To clarify that we are operating with units of carbon sink, “*biofuel crops in marginal lands*” was revised to “*potential of CO₂ capture from combustion of energy crops grown on marginal lands*” in panel (e), and “*biofuel crops in grasslands*” was revised to “*potential of CO₂ capture from combustion of energy crops grown on grasslands*” in panel (f).

A new Fig. S7 (Fig. S8 in the last version) reads as:

Figure S7. Spatial distribution of dedicated energy crops in China.

To estimate the yield of *Miscanthus*, we used spatially explicit parameters including minimum of monthly average temperature (a), annual average precipitation rate (b) and annual average active sunshine hours (c) in 2015. Panel (d) shows the predicted yield of *Miscanthus* by county. The potential of CO₂ capture from combustion of energy crops grown on marginal lands (e), and grasslands (f). Gray color in (d) shows the areas not suitable for growing energy crops, where the minimum of monthly average temperature is below -23 °C or the annual average precipitation rate is lower than 400 mm yr⁻¹ (Xue et al., 2016).

Reference:

Xue, S., Lewandowski, I., Wang, X., & Yi, Z. Assessment of the production potentials of *Miscanthus* on marginal land in China. *Renewable Sustainable Energy Rev.* **54**, 932-943 (2016).

Comment B61

Figure S9: The text is slightly confusing. Do you mean that national pipeline has diameter of 76.2 cm whereas county pipeline has diameter of 7.62 cm? Where does the “a function of CO₂ flow rate” refer to? Title of Sanchez et al. (2018) implies that their data is from US. Is it applicable to China too?

Response

Sorry for the confusion. Our model does not calculate the diameter of a pipeline. In the new **equation 8** in the main text, the unit cost of CO₂ transport is calculated using a function from a previous study (McCollum et al. 2006), based on the CO₂ flow rate and the length of pipeline. Sanchez et al. (2018) suggested that the CO₂ flow rate is also limited by the nominal pipeline size. We have adopted the minimum ($\chi^{MIN}=89 \text{ kt yr}^{-1}$) and maximum ($\chi^{MAX}=26698 \text{ kt yr}^{-1}$) of the CO₂ flow rate from that study due to lack of standards in China. Please refer to our response to **Comment B56** for clarification of this method and the limitation.

To clarify the method and the limitation of applying data from US to China, the following sentences were added on lines 903-909 in the track version: “*In addition, Sanchez et al. suggested that the flow rate of CO₂ transported by pipelines is limited by the nominal pipeline size¹⁶, and we adopted the minimal ($\chi^{MIN}=89 \text{ kt yr}^{-1}$) and maximal ($\chi^{MAX}=26698 \text{ kt yr}^{-1}$) flow rate from that study due to lack of standards in China. Therefore, the capacity of CCS is set to be zero for a county if χ is below 89 kt yr^{-1} , so that there is no pipeline to transport CO₂ captured in power plants in this county; χ is set to 26698 kt yr^{-1} if it is above 26698 kt yr^{-1} , so that unit cost does not decrease when χ exceeds this threshold.*”

and the caption of **Fig. S8** (**Fig. S9** in the last version) was revised as: “*This graph shows that, to transport CO₂ from the site of capture to the site of storage, the pipelines in all counties in a province are converged to one provincial pipeline, and then all provincial pipelines are converged to one pipeline in the country. In equation 8 of the main text, the unit cost of CO₂ transport is calculated as a function of the CO₂ flow rate and the length of pipeline (McCollum and Ogden, 2006). Due to increasing flow rate of CO₂, the unit cost of CO₂ transport decreases from a county pipeline to a provincial or national pipeline. In addition, Sanchez et al. (2018) suggested that the CO₂ flow rate is limited by the nominal pipeline size, and we adopted the minimal ($\chi^{MIN}=89 \text{ kt yr}^{-1}$) and maximal ($\chi^{MAX}=26698 \text{ kt yr}^{-1}$) from that study due to lack of standards in China.*”

References:

Sanchez, D. L., Johnson, N., McCoy, S. T., Turner, P. A. & Mach, K. J. Near-term deployment of carbon capture and sequestration from biorefineries in the United States. *Proc. Natl. Acad. Sci. U.S.A.* **115**, 4875-4880 (2018).

McCollum, D. L. & Ogden, J. M. Techno-Economic Models for Carbon Dioxide Compression, Transport, and Storage & Correlations for Estimating Carbon Dioxide Density and Viscosity. Institute of Transportation Studies University of California-Davis. UCD-ITS-RR-06-14 (2006).

Comment B62

Figure S10: Panels C & F: Is this cost to neighboring province or is there one cost for transports between all the provinces? Typically, transport costs between “n” different areas implies “(n-1) x (n-1)” transport unit costs...

Response

Thank you for the comment.

In panel (c), the unit cost of biomass transport is a function of transport distance (L) (see μ_{ix}^b in **Table S2**), and L is a constant in different provinces. We tried to account for the “(n-1) x (n-1)” values of L between “n” in different provinces, but it made the optimizing model too complicated to run. To make our model computable, our optimization model has considered three cases of biomass and CO₂ transport: from county i to county i ($x/y=1$), from another county in the same province to county i ($x/y=2$), and from a county in other provinces to county i ($x/y=3$). Therefore, L and the unit cost of biomass transport is a constant in panel (c).

In panel (f), the unit cost of CO₂ transport is a function of transport distance (L) and flow rate of CO₂ captured in a province (χ_p) (see **equation 8** in the main text). Although we adopt a constant L in different provinces, the unit cost of CO₂ transport is different in different provinces due to a difference in χ_p (e.g., the unit cost is lower in a province with a larger flow rate of CO₂ transported in the pipelines).

To clarify the limitation of this method, we added the following sentences on line 858-867 in the main text as: “*We approximated the distance of transport as the radii (D) of circles with areas equal to the areas of the county, province and country, $D=(A/\pi)^{1/2}$, where A is the area. There are (n-1)x(n-1) transport distances between n different counties/provinces. Our model has calculated the distance of transport according to the three cases of transport: within county i , from county i to other counties in the same province, and from county i to a county in other provinces. It helps us to qualitatively account for different distances of transport*”

when x or y increases from 1 to 3. This approximation only allows us to account for the priority of transport over shorter distances, but precise calculations are needed when exactly determined locations of biomass supply sites, power plants and carbon storage sites are available.”.

To clarify the calculated distance of biomass and CO₂ transport, the following sentences were added in the caption of **Fig. S9** (**Fig. S10** in the last version) as: “In panel (c) and (f), the distance of biomass and CO₂ transport is a constant in different provinces. Our model calculates the distance based on three cases of transport: within county i , between county i and other counties in the same province, and between county i and a county in other provinces. This simple method helps us qualitatively account for that the transport distance increases when x or y increases from 1 to 3. However, precisely calculated distances of biomass and CO₂ transport are required in the design of a real-world transport system using more spatially explicit data.”.

The new **Fig. S9** reads as:

Figure S9. Unit cost of biomass and CO₂ transport in China.

Panels (a-c) show the unit cost of transport of biomass from the sites of collection in county i to a power plant in county i (a), transport of biomass from other counties in province j to a power plant in county i in province j (b), and transport of biomass from counties in other provinces to a power plant in county i in province j (c). Panels (d-f) show the unit cost of transport of CO₂ captured in a power plant in county i to sites suitable for storage in county i (d), transport of CO₂ captured in a power plant in county i in province j to other counties in province j for storage (e), and transport of CO₂ captured in a power plant in county i to other provinces for storage (f). Panel (g) shows the unit cost of CO₂ transport within the county (green squares), from one county to another county in a province (yellow triangles) and from one province to another province (blue circles) as a function of the CO₂ flow rate.

In panel (c) and (f), the distance of biomass and CO₂ transport is a constant in different provinces. Our model calculates the distance based on three cases of transport: within county i , between county i and other counties in the same province, and between county i and a county in other provinces. This simple method helps us qualitatively account for that the transport distance increases when x or y increases from 1 to 3. However, precisely calculated distances of biomass and CO₂ transport are required in the design of a real-world transport system using more spatially explicit data.

REVIEWER COMMENTS

Reviewer #3 (Remarks to the Author):

Overall impressions

This study is important to understand whether there is a low or negative carbon use for the relatively new assets in coal power production in China. The study improves upon the existing understanding by getting into the details that matter in understanding the cost and net emissions of BECCS, those details being spatial mismatches and the differences between different sources of biomass. The findings are of interest to the global energy research and policy community. The finding that very large emissions reduction potential is available but at a cost above some other low carbon electricity options is useful. Creating a marginal cost curve is a good approach to show that the cost of emissions reductions are not constant and discussions of the potential for BECCS should account for that. I have some concerns about the analysis and the writing that are described below.

The abstract still needs editing to drive home the biggest takeaways from the study. The first 3 sentences are good. The fourth gets into too much detail of the analysis and is oddly phrased. The 2 Gt of emissions reduction should be put into context. The final sentence does not capture the importance of the concept it is relaying. How does this spatial analysis change the understanding of BECCS in China? It looks like it leads to a large increase in the estimated abatement costs, right? That is important! Another potentially important finding to add in here is that there is a path dependence in this system. The lowest cost near term abatement does not naturally evolve into the largest potential abatement.

The section on "Contribution of BECCS to China's emission reduction" is a good addition to put the analysis in context. It does require some editing to keep the main points out of the weeds of the number values and fix some of the prose for clarity.

The policy implications section needs revision. If you had one minute in front of China's Ministry of Energy, what would you want them to know about this study? That is your policy implications section. The current section reads more like a summary of benefits of BECCS. A point within this section, you cannot claim to enhance energy efficiency when the system described reduces efficiency.

Given the high bar for publication in Nature Communications, I am hesitant to accept the manuscript. There are issues that need to be corrected, which if addressed would lead me to recommend the manuscript for publication. I think the subject matter and the study approach are of high value.

Treatment of agricultural residues

Ag. residues are the major contributor to low cost BECCS in the study. The big take home message is that there is a large carbon mitigation potential in BECCS in China utilizing existing coal facilities with agricultural residues, which makes getting the agricultural residue right extremely important. There are a number of issues with the treatment of residues related to the quantity available, cost of acquisition and associated emissions. The first problem has to do with the quantity. The analysis assumes 100% harvest of all residues. This is not feasible for several reasons. Some fraction of residues will remain in the field due to the impracticality of harvesting. The quality of biomass degrades when dirt is added. Harvesting clean residues means leaving a significant fraction in the field. Additionally, studies show that a fraction of the residues need to remain on the field to protect the soil from erosion and to prevent loss of soil carbon, which can negatively impact the net carbon budget of the biomass system. This leads to the concept of sustainably removable yield of residues. For further understanding, check into the work of Muth et al (2013). These factors can severely limit the available biomass from residues. Some of the categories of residues like cotton have been excluded completely in United States analysis because of the sustainably removable yield is too low to make harvest economic, which bring me to the second concern with agricultural residues.

The cost of acquisition is the same for all residues according to Table S2. Residue acquisition requires fixed field operations that lead to a strong yield dependence on the harvest cost especially for low yielding residues. The given cost matches costs for corn stover acquisition but is likely much too low for lower yielding residues such as soy, potato and sugar beet. The model is designed to capture these differences between biomass types but the parameterization does not capture them.

Lastly, it appears from Table S3 that there are no emissions associated with the harvest of agricultural residues. Only emissions accounted for at the field are due to the additional fertilizer required. This is missing some emissions from the harvesting equipment. More complicated are the net land use emissions associated with a program of residue removal that depends on the current practice of residue management. If they are not currently being completely removed, there is likely a significant change in soil carbon that can greatly influence the carbon balance of the biomass (see Wilhelm et al 2007). At the very least, this adds a large degree of uncertainty in the emissions accounting of the BECCS system. Proper accounting would improve the analysis and make the results more robust.

Treatment of spatial aspects of the system

The paper promises to capture the spatial mismatch of biomass resource, coal plants, and sequestration sites. It succeeds only partially on this promise. The transportation distance estimates for within county and to a lesser extent within province capture some of the heterogeneity in the spatial layout of resources. By not explicitly modeling the transport of biomass from resource counties to coal plant counties, the study loses a lot and produces biased results. In the extreme case ($x=3$), all biomass is assumed to travel halfway across the country to a coal plant. This greatly overestimates the costs. Within province ($x=2$), it is less obvious what the bias is but the result is that the low end of the marginal cost curve (low emissions reduction) the estimated cost is high while at the high end, the estimated cost is likely low. Constraining the analysis by county boundaries is artificial and leads to higher than actual cost. These issues can be overcome with included the biomass logistics in the model. I understand this increases the computational requirements for the model but it has been done at similar scale (see Johnson et al 2014). The $x=3$ scenarios are not accurate of any system that might occur and should not be reported. At the very least, the uncertainty in the transportation cost increases as the scenarios go from $x=1$ to $x=3$. This will apply to the emissions associated with transportation as well.

Linearization of non-linear phenomena

Several non-linear phenomena are linearized in the model. The treatment of co-firing of biomass is sound but a little confusing to the reader the scenarios have a 90% or 30% biomass co-firing limit but they are described as all 90% or 30% co-firing. Do the results mostly fall at the limits? If so, then the estimates are sound, if not there may be issues with the costs estimates which linearize the cost of 90% co-firing when the outcome is only 50% co-firing. Second, the pipeline sizing uses an assumption of 40% of the power plant CO₂ production capacity in a county. It is unclear whether this matches the output of the model. It seems that the results of converting all the coal plants to BECCS would lead to more 40% of the CO₂ capacity of the county.

References:

- Muth, D., Jr., K. M. Bryden, and R. G. Nelson. 2013. "Sustainable agricultural residue removal for bioenergy: A spatially comprehensive US national assessment " *Applied Energy* 102: 403–17. doi: 10.1016/j.apener-gy.2012.07.028.
- Wilhelm, Wally W., et al. "Corn stover to sustain soil organic carbon further constrains biomass supply." *Agronomy journal* 99.6 (2007): 1665-1667.
- Johnson, Nils, Nathan Parker, and Joan Ogden. "How negative can biofuels with CCS take us and at what cost? Refining the economic potential of biofuel production with CCS using spatially-explicit modeling." *Energy Procedia* 63 (2014): 6770-6791.

Manuscript ID: NCOMMS-20-07414-T

Title: Spatially explicit analysis identifies significant potential for bioenergy with carbon capture and storage in China

Authors: Xiaofan Xing, Rong Wang, Nicolas Bauer, Philippe Ciais, Junji Cao, Olivier Boucher, Daniel Goll, Josep Peñuelas, Ivan A. Janssens, Yves Balkanski, James Clark, Jianmin Ma, Bo Pan, Shicheng Zhang, Xingnan Ye, Yutao Wang, Qing Li, Gang Luo, Guofeng Shen, Wei Li, Yechen Yang, Siqing Xu, Lin Wang, Xin Yang, Jianmin Chen

Referee #3

Comment C1

Overall impressions

This study is important to understand whether there is a low or negative carbon use for the relatively new assets in coal power production in China. The study improves upon the existing understanding by getting into the details that matter in understanding the cost and net emissions of BECCS, those details being spatial mismatches and the differences between different sources of biomass. The findings are of interest to the global energy research and policy community. The finding that very large emissions reduction potential is available but at a cost above some other low carbon electricity options is useful. Creating a marginal cost curve is a good approach to show that the cost of emissions reductions are not constant and discussions of the potential for BECCS should account for that. I have some concerns about the analysis and the writing that are described below.

Response

Thank you very much for the insightful comments. We are happy to know that our study is important to global energy research and policy community. We have followed all suggestions to revise our analysis or improve the writing. Please see a point-by-point response as below. The line numbers in this Response correspond to those in the track-changed version.

Comment C2

The abstract still needs editing to drive home the biggest takeaways from the study. The first 3 sentences are good. The fourth gets into too much detail of the analysis and is oddly phrased. The 2 Gt of emissions reduction should be put into context. The final sentence does not capture the importance of the concept it is relaying. How does this spatial analysis change the understanding of BECCS in China? It looks like it leads to a large increase in the estimated abatement costs, right? That is important! Another potentially important finding to add in here is that there is a path dependence in this system. The lowest cost near term abatement does not naturally evolve into the largest potential abatement.

Response

We wholeheartedly agree that the lowest cost of near-term abatement does not evolve into the largest potential abatement naturally, and this is the motivation of our study. Our spatial analysis provides the marginal cost curve for BECCS, which helps us to understand the contribution of BECCS to achieving carbon neutrality before 2060 as a national target: “*We aim to have CO₂ emissions peak before 2030 and achieve carbon neutrality before 2060*”, Chinese President Xi Jinping told the United Nations General Assembly via a video link on 22 September.” (Normile, 2020).

Following this suggestion, the fourth sentence was revised to: “*The marginal cost of BECCS to abate 2 Gt CO_{2-eq} yr⁻¹ is lower than current price of solar, wind, nuclear and hydro power, but this cost increases remarkably to abate more emissions.*”, and the last sentence was revised to: “*Our spatially explicit method helps to reduce uncertainty in the abatement costs of BECCS, identifies the best opportunities and illustrates the limitations imposed by logistical challenges and spatial mismatches in the path of deploying large-scale BECCS for carbon neutrality in China*”.

Reference:

Normile, D. China’s bold climate pledge earns praise—but is it feasible? *Science* **370**, 17-18 (2020).

Comment C3

The section on “Contribution of BECCS to China’s emission reduction” is a good addition to put the analysis in context. It does require some editing to keep the main points out of the weeds of the number values and fix some of the prose for clarity.

Response

Following this suggestion, we had revised the section “**Contribution of BECCS to China’s emission reduction**” to focus on how BECCS facilitates decarbonization of the China’s power system, reading as: “*We assess the contribution of BECCS to China’s CO₂ emission reduction within a low-carbon energy portfolio of water (i.e., hydropower), wind, solar and nuclear (WWSN) power^{42,59} (Methods). Emissions over 2021–2030 in China’s power sector will reach 53 Gt CO_{2-eq} in a baseline case without expansion of WWSN and development of BECCS, and decline to 47 Gt CO_{2-eq} under the current legislation^{42,59} on WWSN. According to our estimated marginal cost curve for BECCS, investing \$2.3 trillion over 2021–2030 (that is 1% of China’s GDP) into BECCS can reduce emissions of 21 Gt CO_{2-eq}, while increasing this investment to \$7.4 trillion permits to reach a carbon-emission neutral power system by 2030 (Fig. 6a,b). To be consistent with the nationally determined contribution (NDC)⁶, that is reducing the carbon intensity by 70% over 2005–2030, a scenario has to reach total emissions of 47 Gt CO_{2-eq} from China’s power sector over 2021–2030. This cap can be achieved without BECCS (Fig. 6b). However, the choice of using BECCS allows for China to reach lower emissions than those announced in the NDC, at a marginal cost lower than current price for solar and wind power⁵² (Fig. 3). In a scenario limiting emissions from the China’s power sector within 5% of the remaining global carbon budget⁵ for 2 °C (Methods), an investment of \$1.1 trillion over 2021–2030 (that is 0.50% of GDP) is required into generating electricity of 1.1 PWh yr⁻¹ with BECCS at a marginal cost of \$82⁺⁹⁶₋₃₅ (t CO_{2-eq})⁻¹ (Fig. 6c). Therefore, investment in BECCS is inevitable to reach carbon neutrality in China’s power sector, in particular if the penetration of WWSN is delayed⁶⁰”.*

Comment C4

The policy implications section needs revision. If you had one minute in front of China’s Ministry of Energy, what would you want them to know about this study? That is your policy implications section. The current section reads more like a summary of benefits of BECCS. A point within this section, you cannot claim to enhance energy efficiency when the system described reduces efficiency.

Response

Carbon neutrality is documented as a national target for the China’s government: “*We aim to have CO₂ emissions peak before 2030 and achieve carbon neutrality before 2060*”, Chinese President Xi Jinping told the United Nations General Assembly via a video link on 22 September (Normile, 2020). The key message from our study to the policy makers is that we confirm the value of BECCS in meeting near- and long-term emission reduction targets, which can inform the China’s government of the direction for investment in BECCS and clarify the environmental, social and technological challenges of developing large-scale BECCS in China.

Following this suggestion, we had revised the “**Policy implications**” section to focus on the implication of our study to the carbon-neutrality target of the China’s government, reading as: “*The pressing climate change issue creates incentives for near-term negative-emission technologies⁶¹. China’s government has targeted at reaching the carbon neutrality by 2060, but the path to this target remains unclear⁶². A major challenge in emission reduction for the growing China’s economy is that existing coal power plants become stranded assets combined with the need to ramp-up new generation capacities⁹. Our spatially explicit method shows that retrofitting existing coal-fired plants allows China to face these challenges in a win-win solution by 2030. First, retrofitting plants to co-fire biomass and equipping them with CCS will form an early entry point to abate 2 Gt CO_{2-eq} yr⁻¹ of emissions in the short term. Second, taking a longer-term perspective, the stock of biomass, the growing electricity demand and the potential for geological storage needs to be matched more flexibly and efficiently in space. If the option of retrofitting power plants for biomass co-firing and CCS is implemented, China can achieve carbon emission abatement beyond the announced NDC. This study confirms the value of BECCS in meeting near- and long-term emission reduction targets⁶³, informs the China’s government of the direction for investment in BECCS, and clarifies the environmental, social and technological challenges⁶⁴ of developing large-scale BECCS in China.*”.

Reference:

Normile, D. China’s bold climate pledge earns praise—but is it feasible? *Science* **370**, 17-18 (2020).

Comment C5

Given the high bar for publication in Nature Communications, I am hesitant to accept the manuscript. There are issues that need to be corrected, which if addressed would lead me to recommend the manuscript for publication. I think the subject matter and the study approach are of high value.

Response

Thank you for the positive comment. We are very happy to improve our model and fix the problems pointed out here. We had defended our model in responses to Comment C6, C11 and C12, while we modified our model by considering 1) additional costs for using technologies to protect soil from erosion and prevent loss of carbon (Comment C7), 2) different acquisition costs for low-yielding crops (Comment C8), 3) equipment emissions in harvesting (Comment C9) and 4) larger uncertainty in the distance for transportation of biomass and CO₂ from county to county (scenario of $x,y=2$) or from province to province (scenario of $x,y=3$) than for transportation in a county (scenario of $x,y=1$) (Comment C10).

After modifying our model, the marginal cost curves for BECCS in our Fig. 3 had changed from:

to:

Resulting from these updates, the marginal cost of BECCS in our central scenario (the blue line in Fig. 3) had increased by 10% ($\$6 \text{ (t CO}_2\text{-eq)}^{-1}$) as an average for abatement of 0–2 Gt $\text{CO}_2\text{-eq yr}^{-1}$ and by 1% ($\$3 \text{ (t CO}_2\text{-eq)}^{-1}$) for abatement of 2–5 Gt $\text{CO}_2\text{-eq yr}^{-1}$. The uncertainty range had increased from $\$40.3\text{--}88.2 \text{ (t CO}_2\text{-eq)}^{-1}$ to $\$46.1\text{--}100.5 \text{ (t CO}_2\text{-eq)}^{-1}$ for abatement of 1 Gt $\text{CO}_2\text{-eq yr}^{-1}$, from $\$44.6\text{--}185.7 \text{ (t CO}_2\text{-eq)}^{-1}$ to $\$48.0\text{--}196.5 \text{ (t CO}_2\text{-eq)}^{-1}$ for abatement of 2 Gt $\text{CO}_2\text{-eq yr}^{-1}$, and from $\$105.1\text{--}383.6 \text{ (t CO}_2\text{-eq)}^{-1}$ to $\$104.5\text{--}400.0 \text{ (t CO}_2\text{-eq)}^{-1}$ for abatement of 3 Gt $\text{CO}_2\text{-eq yr}^{-1}$, respectively.

Therefore, these updates do not alter our conclusion that there is a remarkable emission reduction potential available by using BECCS at a lower cost than other renewable energy in China.

All numbers in our paper are updated basing upon the updated model.

Comment C6

Treatment of agricultural residues

Ag. residues are the major contributor to low cost BECCS in the study. The big take home message is that there is a large carbon mitigation potential in BECCS in China utilizing existing coal facilities with agricultural residues, which makes getting the agricultural residue right extremely important. There are a number of issues with the treatment of residues related to the quantity available, cost of acquisition and associated emissions. The first problem has to do with the quantity. The analysis assumes 100% harvest of all residues. This is not feasible for several reasons. Some fraction of residues will remain in the field due to the impracticality of harvesting. The quality of biomass degrades when dirt is added. Harvesting clean residues means leaving a significant fraction in the field.

Response

We considered that 100% of agricultural residue associated with the harvested grain, rather than 100% of agricultural residue growing in the field, can be used for bioenergy. Our method is different from previous studies basing on the net primary productivity of crop (Yang *et al.*, 2010; Zhao *et al.*, 2018), which relied on a fraction of straw and grain lost in harvesting. We estimated the quantity of agricultural residue basing on the quantity of harvested grain and the crop-specified straw-to-grain ratio for above-ground biomass (excluding difficult-to-obtain biomass like roots). We assumed that the fraction of straw lost in harvesting is the same as that of grain. We supposed that this assumption is not very bad when we do not have better information for the fraction of straw and grain lost in harvesting, so we kept this assumption in our model.

To clarify this assumption, we added the following sentences in lines 562-569 as: “*Our method estimating the quantity of agricultural residue is different from previous studies basing on net primary productivity of crops^{28,35}, which adopted the fraction of straw and grain that can be collected in the field. We assumed that the fraction of straw growing in the field is equal to that for grain. We estimated the quantity of agricultural residue basing on the quantity of harvested grain and a crop-specified straw-to-grain ratio for above-ground biomass (excluding difficult-to-obtain biomass like roots) (Table S10). To estimate the quantity of agricultural residues, 100% of agricultural residue associated with the harvested grain, rather than 100%*

of agricultural residue growing in the field, are used for BECCS in this study.”.

References:

Yang, Y. et al. Quantitative appraisal and potential analysis for primary biomass resources for energy utilization in China. *Renewable Sustainable Energy Rev.* **14**, 3050-3058 (2010).

Zhao, G. Assessment of potential biomass energy production in China towards 2030 and 2050. *Int. J. Sustainable Energy* **37**, 47-66 (2018).

Comment C7

Treatment of agricultural residues

Additionally, studies show that a fraction of the residues need to remain on the field to protect the soil from erosion and to prevent loss of soil carbon, which can negatively impact the net carbon budget of the biomass system. This leads to the concept of sustainably removable yield of residues. For further understanding, check into the work of Muth et al (2013). These factors can severely limit the available biomass from residues. Some of the categories of residues like cotton have been excluded completely in United States analysis because of the sustainably removable yield is too low to make harvest economic, which brings me to the second concern with agricultural residues. ... More complicated are the net land use emissions associated with a program of residue removal that depends on the current practice of residue management. If they are not currently being completely removed, there is likely a significant change in soil carbon that can greatly influence the carbon balance of the biomass (see Wilhelm et al 2007). At the very least, this adds a large degree of uncertainty in the emissions accounting of the BECCS system. Proper accounting would improve the analysis and make the results more robust.

References:

Muth, D., Jr., K. M. Bryden, and R. G. Nelson. 2013. Sustainable agricultural residue removal for bioenergy: A spatially comprehensive US national assessment. *Applied Energy* 102: 403–17. doi: 10.1016/j.apenergy.2012.07.028.

Wilhelm, Wally W., et al. “Corn stover to sustain soil organic carbon further constrains biomass supply.” *Agronomy journal* 99.6 (2007): 1665-1667.

Response

Thank you very much for bringing the literature of Wilhelm et al. (2007) and Muth et al. (2013) to our notice. We wholeheartedly agree that protecting soil from erosion and preventing loss of carbon are important issues that should be considered when agricultural residues are used for BECCS. We had reviewed the literature estimating the fraction of sustainable agricultural residues for bioenergy (Kadam and McMillan, 2003; Nikolaou et al., 2003; Katterer et al., 2004; Nelson et al., 2004; Graham et al., 2007; Ericsson and Nilsson, 2006; Fischer et al., 2010; Scarlat et al., 2010; Karkee et al., 2012; Muth et al., 2013; Sahoo et al., 2016) in **Table S6**. We had also carefully reviewed the literature using technologies to prevent soil erosion and soil carbon loss (Wen et al., 2020; PIRD, 2020; Arunrat et al., 2020; DAFM, 2020; Tanveer et al., 2019; Li et al., 2019; Rahma et al., 2019; Wang et al., 2019; Guo et al., 2019; Mo et al., 2019; Li et al., 2018; Dai et al., 2018; Wang et al., 2018; Pan et al., 2018; Xu et al., 2018; Pan et al., 2017; Wang et al., 2017; Chan et al., 2010) in **Table S7**. Following this suggestion, we designed two experiments, where an average fraction of sustainable agricultural residues (50%) reduces the quantity of agricultural residues or additional technologies are taken for soil remediation after taking the agricultural residues away at an average cost of \$ 76 ha⁻¹ yr⁻¹. The cost of five soil remediation technologies taken from the literature (Kuhlman et al., 2010; Chukalla et al., 2017) is listed in **Table S8**. In a new **Fig S3a**, the marginal cost curve moves leftward if 50% of agricultural residues are taken away for bioenergy (the orange line), but the marginal cost increases slightly when soil remediation technologies (e.g., plastic film mulch) are taken to protect soil from erosion and prevent soil carbon loss (Chukalla et al., 2017) (the black line).

A new **Fig S3a** was added into the Supporting Information as:

Figure S3. Sensitive tests for the marginal cost of BECCS. ... (a) Relative to the central scenario adopting additional technologies at an average cost of \$ 76 ha⁻¹ yr⁻¹ (Tables S6, S7) for soil remediation (the black line), we considered one scenario using an average fraction of sustainable agricultural residues (50%) (Table S5) (the orange line), and one scenario using 100% agricultural residues without soil remediation technologies (the purple line). Comparing to the scenario without considering the issues of soil degradation (the purple line), the marginal cost curve moves leftward if 50% of agricultural residues are taken away for bioenergy (the orange line), while the marginal cost increases slightly when soil remediation technologies (e.g., plastic film mulch) can be used to protect soil from erosion and prevent soil carbon loss (the black line) (Chukalla et al., 2017).

This item is now included in Tables S2, S11.

A new paragraph was added into the section of “**Sensitivity analyses**” in lines 291-302 as: “*While agricultural residues are considered as cheap sources of bioenergy, attentions should be paid to the impact on soil erosion⁵⁶ and carbon content⁵⁷ if agricultural residues were taken away for bioenergy. To investigate the influence, we designed two experiments using an average fraction of sustainable agricultural residues (50%) (Table S6) to reduce the quantity of agricultural residues for BECCS or taking additional technologies for soil remediation at an average cost of \$ 76 ha⁻¹ yr⁻¹ (Tables S7, S8). Fig. S3a shows that, relative to a scenario without considering these issues, the marginal cost curve would move leftward if 50% of agricultural residues are taken away for bioenergy (the orange line), while the marginal cost increases slightly when soil remediation technologies (e.g., plastic film mulch) can be taken to protect soil from erosion and prevent soil carbon loss⁵⁸. More studies are needed to understand the environmental impacts of BECCS on soil degradation^{56,57} and the effect of soil remediation technologies⁵⁸.*”.

The new Tables S6-S8 were added into the Supporting Information as:

Table S6. Sustainable agricultural residues for bioenergy.

Methodology	Fraction of sustainable agricultural residue removal (%)	Reference
Estimated basing on US Department of Agriculture (USDA) guidelines for residue management.	40-50 (wheat, barley, rye and oats) and 40 (maize)	Kadam and McMillan, 2003
Estimated basing on the current fraction of straw usage.	30 (wheat, barley, rye, oats, maize and rapeseed)	Nikolaou et al., 2003

Estimated basing on the carbon input in the top layer of soil.	60 (wheat, barley, rye, oats and maize)	Kätterer et al., 2004
Estimated using the rain and wind erosion equations, where the input data include soil type, acres of that particular soil type, field topology characteristics (percentage low and high slopes), erodibility, and tolerable soil-loss limit.	30-70 (maize)	Graham et al., 2007
Assumed basing on empirical data.	25 (wheat, barley, rye, oats and maize)	Ericsson and Nilsson, 2006
Assumed basing on empirical data.	50 of crop residues	Fischer et al., 2010
Estimated basing on a summary of literature.	40 (wheat, rye, barely) and 50 (maize, rice and sunflower)	Scarlat et al., 2010
Estimated using the revised Universal Soil Loss Equation 2 (RUSLE2) erosion model, where the input data include management practices, soil structure, slope, slope length, temperature, precipitation, crop grain yield and supporting practices.	56-98 (corn and soybean)	Karkee et al., 2012
Estimated using the RUSLE2 model, where input data include the SSURGO soil survey database, the NRCS managed RUSLE2 climate database, the CLIGEN daily climate generator, the WINDGEN daily wind speed and direction generator, land management and crop yield.	83 (corn), 1.9 (rice) and 14 (wheat)	Muth et al., 2013
Estimated using the RUSLE2 model, where the input data include detailed site information and crop management practice.	68-76 (cotton)	Sahoo et al., 2016

Table S7. Application of soil remediation technologies in the literature.

TP: Tillage practice; IRF: Increase rotational frequency; GM: Grass mulch; PM: Plastic film mulch; AC: Animal manure and compost fertilizer; IR: Irrigation; CT: Contour tillage; FA: Fallow; PA: Precision agriculture; RR: Root recycling; BC: Biochar; BI: Bioremediation.

Technologies											Reference	
TP	IRF	GM	PM	AC	IR	CT	FA	PA	RR	BC	BI	
√		√	√			√						Wen et al., 2020
	√	√		√	√		√		√	√		PIRD, 2020
√												Arunrat et al., 2020
√	√	√		√					√			DAFM, 2020
√	√	√	√	√				√		√	√	Tanveer et al., 2019
√												Li et al., 2020
			√									Rahma et al., 2019
√			√									Wang et al., 2019
√												Guo et al., 2019

√			√							Li et al., 2018
√	√	√			√					Dai et al., 2018
			√							Pan et al., 2018
	√									Pan et al., 2017
			√	√						Wang et al., 2017
√	√		√	√		√	√	√	√	Chan et al., 2010

Table S8. Cost of soil remediation technologies.

Technologies	Cost (\$ ha ⁻¹ yr ⁻¹)	Reference
Root mulch	58	Kuhlman et al., 2010
Grass mulch	76	Kuhlman et al., 2010
Tillage practice	78	Kuhlman et al., 2010
Contour tillage	27	Kuhlman et al., 2010
Plastic film mulch	140	Chukalla et al., 2017

References:

- Arunrat, N., Pumijumng, N., Sreenonchai, S. & Chareonwong, U. Factors Controlling Soil Organic Carbon Sequestration of Highland Agricultural Areas in the Mae Chaem Basin, Northern Thailand. *Agronomy*, **10**, 305 (2020).
- Chan, K. Y., Oates, A., Liu, D. L., Li, G. D. & Conyers, M. K. *A Farmer's Guide To Increasing Soil Organic Carbon Under Pastures*. https://www.dpi.nsw.gov.au/_data/assets/pdf_file/0014/321422/A-farmers-guide-to-increasing-Soil-Organic-Carbon-under-pastures.pdf (2010).
- Chukalla, A. D., Krol, M. S. & Hoekstra, A. Y. Marginal cost curves for water footprint reduction in irrigated agriculture: guiding a cost-effective reduction of crop water consumption to a permit or benchmark level. *Hydrol. Earth Syst. Sci.* **21**, 3507 (2017).
- Dai, C. et al. Exploring optimal measures to reduce soil erosion and nutrient losses in southern China. *Agric. Water Manage.* **210**, 41-48 (2018).
- Department of Agriculture, Food, and the Marine (DAFM). Maintenance of Soil Organic Matter. <https://www.agriculture.gov.ie/farmerschemespayments/crosscompliance/soilorganicmatter/> (2020).
- Ericsson, K. & Nilsson, L. J. Assessment of the potential biomass supply in Europe using a resource-focused approach. *Biomass Bioenergy* **30**, 1-15 (2006).
- Fischer, G. et al. Biofuel production potentials in Europe: Sustainable use of cultivated land and pastures, Part II: Land use scenarios. *Biomass Bioenergy* **34**, 173-187 (2010).
- Graham, R. L., Nelson, R., Sheehan, J., Perlack, R. D. & Wright, L. L. Current and potential US corn stover supplies. *Agron. J.* **99**, 1-11 (2007).
- Guo, S. et al. Cross-ridge tillage decreases nitrogen and phosphorus losses from sloping farmlands in southern hilly regions of China. *Soil Tillage Res.* **191**, 48-56 (2019).
- Kadam, K. L. & McMillan, J. D. Availability of corn stover as a sustainable feedstock for bioethanol production. *Bioresour. Technol.* **88**, 17-25 (2003).
- Karkee, M., McNaull, R. P., Birrell, S. J. & Steward, B. L. Estimation of optimal biomass removal rate based on tolerable soil erosion for single-pass crop grain and biomass harvesting system. *Trans. ASABE* **55**, 107-115 (2012).

- Kätterer, T., Andrén, O. & Persson, J. The impact of altered management on long-term agricultural soil carbon stocks—a Swedish case study. *Nutr. Cycling Agroecosyst.* **70**, 179-188 (2004).
- Kuhlman, T., Reinhard, S. & Gaaff, A. Estimating the costs and benefits of soil conservation in Europe. *Land Use Policy* **27**, 22-32 (2010).
- Li, M. et al. Cropland physical disturbance intensity: plot-scale measurement and its application for soil erosion reduction in mountainous areas. *Journal of Mountain Science* **15**, 198-210 (2018).
- Li, T. et al. Exploring the interaction of surface roughness and slope gradient in controlling rates of soil loss from sloping farmland on the Loess Plateau of China. *Hydrol. Processes* **34**, 339-354 (2020).
- Muth Jr, D. J., Bryden, K. M. & Nelson, R. G. Sustainable agricultural residue removal for bioenergy: A spatially comprehensive US national assessment. *Appl. Energy* **102**, 403-417 (2013).
- Nelson, R. G., Walsh, M., Sheehan, J. J. & Graham, R. Methodology for estimating removable quantities of agricultural residues for bioenergy and bioproduct use. *Appl. Biochem. Biotechnol.* **113**, 13-26 (2004).
- Nikolaou, A., Remrova, M. & Jeliakov, I. Lot 5: Bioenergy's Role in the EU Energy Market. *Biomass Availability in Europe* (2003).
- Pan, D. et al. Application rate influences the soil and water conservation effectiveness of mulching with chipped branches. *Soil Sci. Soc. Am. J.* **82**, 447-454 (2018).
- Pan, D. et al. Dynamics of runoff and sediment trapping performance of vegetative filter strips: Run-on experiments and modeling. *Sci. Total Environ.* **593**, 54-64 (2017).
- Pan, D. et al. Effect of plant cover type on soil water budget and tree photosynthesis in jujube orchards. *Agric. Water Manage.* **184**, 135-144 (2017).
- Primary Industries and Regional Development (PIRD). *Managing Soil Organic Carbon On Western Australian Farms*. <https://www.agric.wa.gov.au/soil-carbon/managing-soil-organic-carbon-western-australian-farms> (2020)
- Rahma, A. E., Warrington, D. N. & Lei, T. Efficiency of wheat straw mulching in reducing soil and water losses from three typical soils of the Loess Plateau, China. *International Soil and Water Conservation Research* **7**, 335-345 (2019).
- Sahoo, K., Hawkins, G. L., Yao, X. A., Samples, K. & Mani, S. GIS-based biomass assessment and supply logistics system for a sustainable biorefinery: A case study with cotton stalks in the Southeastern US. *Appl. Energy* **182**, 260-273 (2016).
- Scarlat, N., Martinov, M. & Dallemand, J. F. Assessment of the availability of agricultural crop residues in the European Union: potential and limitations for bioenergy use. *Waste Manage.* **30**, 1889-1897 (2010).
- Tanveer, S. K., Lu, X., Hussain, I. & Sohail, M. Soil Carbon Sequestration through Agronomic Management Practices. In *CO₂ Sequestration*. IntechOpen (2019)
- Wang, L. et al. Soil C and N dynamics and hydrological processes in a maize-wheat rotation field subjected to different tillage and straw management practices. *Agric. Ecosyst. Environ.* **285**, 106616 (2019).
- Wang, Y. et al. Modelling soil detachment of different management practices in the red soil region of China. *Land Degrad. Dev.* **28**, 1496-1505 (2017).
- Wen, X. & Zhen, L. Soil erosion control practices in the Chinese Loess Plateau: A systematic review. *Environmental Development* **34**, 100493 (2020).
- Wilhelm, W. W. et al. Corn stover to sustain soil organic carbon further constrains biomass supply. *Agr. J.* **99**, 1665-1667 (2007).

Comment C8

Treatment of agricultural residues

The cost of acquisition is the same for all residues according to Table S2. Residue acquisition requires fixed field operations that lead to a strong yield dependence on the harvest cost especially for low yielding residues.

The given cost matches costs for corn stover acquisition but is likely much too low for lower yielding residues such as soy, potato and sugar beet. The model is designed to capture these differences between biomass types but the parameterization does not capture them.

Response

Following this suggestion, we tried to account for higher acquisition costs of lower-yielding residues. The harvest cost used in our study was taken from a study in China (Komarek *et al.* 2013), which did not distinguish the harvest costs for different agricultural residues. We found that the same harvest cost was applied for different agricultural residues due to a lack of data in China in other studies (Zhang *et al.*, 2013; Sun *et al.*, 2017; Lu *et al.*, 2019). For this reason, we assumed that the harvest cost for low-yielding agricultural residues including soy, potato and sugar beet is two times of that for high-yielding agricultural residues. A figure below (this figure is not shown in our paper) indicates that changing this parameter has a minor impact on the marginal cost curve of BECCS (from the black line to the blue line):

The different cost of acquisition is now included in **Table S2**.

To clarify the impact of this parameter, we added the following sentence in lines 677-680 in **Methods** as: *“Acquisition requires fixed field operations that lead to dependence of the harvest cost on the yield. Due to a lack of data, we assumed that the harvest cost for low-yielding residues (soy, potato and sugar beet) is two times of that for other residues⁸³, which had a minor impact on our result.”*

References:

Komarek, A. M. Costs and benefits of crop residue retention in a Chinese subsistence farming system. No. 424-2016-27114, (2013).

Zhang, Q., Zhou, D., Zhou, P. & Ding, H. Cost analysis of straw-based power generation in Jiangsu Province, China. *Appl. Energy*, **102**, 785-793 (2013).

Sun, Y., Cai, W., Chen, B., Guo, X., Hu, J. & Jiao, Y. Economic analysis of fuel collection, storage, and transportation in straw power generation in China. *Energy*, **132**, 194-203 (2017).

Lu, X. *et al.* Gasification of coal and biomass as a net carbon-negative power source for environment-friendly electricity generation in China. *Proc. Natl. Acad. Sci. U.S.A.* **116**, 8206-8213 (2019).

Comment C9

Treatment of agricultural residues

Lastly, it appears from Table S3 that there are no emissions associated with the harvest of agricultural residues. Only emissions accounted for at the field are due to the additional fertilizer required. This is missing some emissions from the harvesting equipment.

Response

Following this suggestion, we considered that 15.6 liters of diesel are consumed in harvesting biomass in one

hectare (Masek et al., 2015), which is converted to 0.012 t CO₂-eq (t biomass)⁻¹ using an average yield of residues in China for agricultural crop (4.4 t biomass ha⁻¹) (NBSC, 2016, REDCP, 2019) and energy crop (24.9 t biomass ha⁻¹) (Xue et al., 2016, REDCP, 2019) and the emission factor of diesel (0.0034 t CO₂-eq L⁻¹) (Fajardy and Mac Dowell, 2017).

This item is now included in **Table S3**.

References:

Masek, J., Novak, P., & Pavlicek, T. Evaluation of combine harvester fuel consumption and operation costs. *In 14th International Scientific Conference Engineering for Rural Development Proceedings* **14**, 78-83 (2015).

Resource and Environment Data Cloud Platform (REDCP). *Remote Sensing Monitoring Data of Land-use in China in 2015*. <http://www.resdc.cn/data.aspx?DATAID=184> (2019).

Xue, S., Lewandowski, I., Wang, X. & Yi, Z. Assessment of the production potentials of *Miscanthus* on marginal land in China. *Renewable Sustainable Energy Rev.* **54**, 932-943 (2016).

National Bureau of Statistics of the People's Republic of China (NBSC). *China Statistics Yearbook 2001-2016* (China Statistics Press, Beijing, 2016).

Comment C10

Treatment of spatial aspects of the system

The paper promises to capture the spatial mismatch of biomass resource, coal plants, and sequestration sites. It succeeds only partially on this promise. The transportation distance estimates for within county and to a lesser extent within province capture some of the heterogeneity in the spatial layout of resources. By not explicitly modeling the transport of biomass from resource counties to coal plant counties, the study loses a lot and produces biased results. In the extreme case ($x=3$), all biomass is assumed to travel halfway across the country to a coal plant. This greatly overestimates the costs. Within province ($x=2$), it is less obvious what the bias is but the result is that the low end of the marginal cost curve (low emissions reduction) the estimated cost is high while at the high end, the estimated cost is likely low. Constraining the analysis by county boundaries is artificial and leads to higher than actual cost. These issues can be overcome with included the biomass logistics in the model. I understand this increases the computational requirements for the model but it has been done at similar scale (see Johnson et al 2014). The $x=3$ scenarios are not accurate of any system that might occur and should not be reported. At the very least, the uncertainty in the transportation cost increases as the scenarios go from $x=1$ to $x=3$. This will apply to the emissions associated with transportation as well.

Response

Thank you for pointing us to the study by Johnson et al. (2014). We wholeheartedly agree that the best spatially-explicit analysis is to optimize the quantity of biomass and CO₂ transported from site to site, but this is limited by overloaded computational costs (as noted by the reviewer). Johnson et al. (2014) used data for routes of CO₂ pipelines in the USA defined by the National Pipeline Mapping System, but similar data are not available for China. Johnson et al. (2014) considered transportation of CO₂ only and optimized a given number of variables for CO₂ transported by ~1000 pipelines as:

Reprinted from Energy Procedia Volume 63, Johnson et al, How negative can biofuels with CCS take us and at what cost? Refining the economic potential of biofuel production with CCS using spatially-explicit modeling, p 6770-6791, Copyright 2014 with permission from Elsevier

Fig. 10. Optimal biorefinery and CCS infrastructure in 2050 (MaxCO₂); 650 Mt CO₂/year captured

There is no doubt that the method used by Johnson et al. (2014) is nice, but there are some differences between our model in China (abating ~5 Gt CO₂ yr⁻¹) and the model in the USA (abating 0.65 Gt CO₂ yr⁻¹). Our model is designed to optimize transportation for 21 types of biomass and CO₂ across 2836 counties. There are 2836×2836×22=176943712 variables if we follow the method of Johnson et al. (2014) to optimize the transportation from county to county. By considering transportation in three scenarios (x,y=1,2,3) in our model, the number of variables is reduced to 2836×21×3×3=536004. Running our model for one time takes around 50 hours using 8 CPU. Increasing computational time by 330 times would make our analysis unaffordable and prevent us from assessing the sensitivity to many factors and analyzing the uncertainty in the Monte Carlo simulation in Fig 3.

Therefore, we followed this suggestion to consider that the uncertainty in transportation distance increases as the scenarios go from (x,y=1) and (x,y=2,3). We designed four sensitivity experiments, where the distance of transportation for biomass and CO₂ is altered by ±50% in the scenarios of x,y=2,3. As shown in Fig. S3b, the impact on the BECCS marginal cost is low when the abatement target is <2 Gt CO₂ yr⁻¹ (likely because long-range transportation is not used), although the impact increases to abate more emissions. We considered an additional uncertainty of ±50% in the transportation distance for x,y=2,3 in our Monte Carlo simulation, which increased the uncertainty in the marginal cost (see update of Fig 3 in our response to Comment C5).

The study of Johnson et al. (2014) is included in Table S1.

To clarify the impact of this parameter on our results, we added the sentences in lines 735-742 as: “To consider that the uncertainty in the transportation cost increases as the scenarios go from x,y=1 to x,y=3, we designed four additional sensitivity tests, where the distance of transportation is changed by ±50% in the scenarios with x,y=3 or x,y=2,3. After changing the distance of transportation, the low marginal costs of BECCS remain robust to abate <2 Gt CO₂ yr⁻¹, although the impact becomes more significant as the target of abated emissions increases (Fig. S3b). We considered an additional uncertainty of ±50% in the distance for biomass and CO₂ transportation in the scenarios of x,y=2,3 in our Monte Carlo simulations.”.

A new Fig. S3b was added into the Supporting Information as:

Figure S3. Sensitive tests for the marginal cost of BECCS. ... (b) Relative to the central scenario with the distance of transportation of biomass and CO₂ calculated by our simple method (the black line), we designed four sensitivity experiments, where the distances of transportation for biomass and CO₂ are altered by $\pm 50\%$ in the scenarios of $x,y=2$ (the cyan and pink lines) or $x,y=2,3$ (the blue and orange lines). Comparing to the central scenario, the influence on the marginal cost of BECCS is low when the abatement target is < 2 Gt CO₂ yr⁻¹ when long-range transportation is not used, although the influence is more significant to abate more emissions.

Reference:

Johnson, N., Nathan, P. & Joan, O. How negative can biofuels with CCS take us and at what cost? Refining the economic potential of biofuel production with CCS using spatially-explicit modeling. *Energy Procedia* **63**, 6770-6791 (2014).

Comment C11

Linearization of non-linear phenomena

Several non-linear phenomena are linearized in the model. The treatment of co-firing of biomass is sound but a little confusing to the reader the scenarios have a 90% or 30% biomass co-firing limit but they are described as all 90% or 30% co-firing. Do the results mostly fall at the limits? If so, then the estimates are sound, if not there may be issues with the costs estimates which linearize the cost of 90% co-firing when the outcome is only 50% co-firing.

Response

We do not linearize the cost of 90% co-firing when the targeted ratio of biomass in fuel is 50%. We consider that $50/90=55.6\%$ of power plants in a county are retrofitted for 90% co-firing, while the remaining power plants are still burning coal. The fraction of biomass co-firing is not optimized in our model, which depends on the national target of emission reduction. This parameter is related to rated evaporation, boiler thermal load and fuel flow rate when retrofitting a coal-fired plant for biomass co-firing. After a plant is retrofitted, it is hard to adjust this parameter to meet a different target of emission reduction. Following the practice in the literature (Lu *et al.*, 2019), we designed different scenarios by varying the fraction of biomass co-firing from 30% to 90%. Fig. 3 shows that the marginal cost to abate emissions is lower in the scenario of 30% co-firing (the black line) than 90% co-firing (the blue line) due to a higher power generation efficiency, but the potential of emission abatement is lower due to a lower capacity of burning biomass.

To make this point clearer, the sentences in lines 643-650 were revised as: “*The fraction of biomass co-firing is a variable that is not optimized in our model. When a plant is retrofitted for a given biomass co-firing ratio, it is hard to adjust this parameter to meet a different target of emission reduction. Following the practice in*

the literature¹¹, we designed different scenarios by varying the fraction of biomass co-firing from 30% to 90% (see Fig. 3). We did not linearize the cost under a given co-firing ratio when the targeted ratio of biomass in fuel is <90% in a county. For example, we considered that 50/90=55.6% of power plants are retrofitted in the scenario of 90% co-firing, while the remaining power plants are still burning coal.”.

Our new Fig. 3 in the main text reads as:

Fig. 3 | Marginal cost curves of CO₂ emission reduction by BECCS in China. Scenarios include: (I) B90-2015-PC, where “B90” stands for 90% biomass co-firing, “2015” stands for generating electricity by coal-fired plants in 2015 (4.24 PWh yr⁻¹), and “PC” stands for using pulverized-coal (PC) plants, (II) B30-2015-PC, where “B30” stands for 30% biomass co-firing, (III) B90-2015-IGCC, where “IGCC” stands for transferring PC to integrated gasification combined cycle (IGCC) plants, (IV) B30-2015-PC-EneCrop, where “EneCrop” stands for using dedicated energy crops (Miscanthus) only, (V) B90-2030-PC, where “2030” stands for generating the projected electricity in 2030 (9.5 PWh yr⁻¹)⁴², and (VI) B90-2015-PC-BestCrop, where “BestCrop” stands for shifting energy crops from Miscanthus to the best-yield crops²⁴, (VII) noBiomass-2015-PC, where “noBiomass” stands for using coal only in power plants, and (VIII) B90-2015-PC-noCCS, where “noCCS” stands for excluding CCS facilities in power plants. The national CO₂ emission reduction (calculated from the substituted coal consumption⁴² and the CO₂ emission factor of coal⁵³) by hydropower, wind, solar and nuclear power in China in 2015 as well as the range of their marginal costs (calculated from their prices of electricity generation in different provinces⁵² and the CO₂ emission factor of coal⁵³) are shown as vertical lines. Uncertainty in the marginal cost curve in the central scenario is assessed by considering uncertainty in parameterization in a Monte Carlo simulation (see details in **Methods**), which is shown as the shaded area.

Comment C12

Second, the pipeline sizing uses an assumption of 40% of the power plant CO₂ production capacity in a county. It is unclear whether this matches the output of the model. It seems that the results of converting all the coal plants to BECCS would lead to more 40% of the CO₂ capacity of the county.

Response

The unit cost of transporting CO₂ depends on the pipeline size, which is a function of the total quantity of CO₂ transported in the pipeline. When the total quantity is lower, the pipeline is thinner and the unit cost is higher. We consider that all pipelines are designed to transport 40% of the total CO₂ emissions in a county, which helps us to estimate the unit cost of transporting CO₂. If the CO₂ transported in the pipeline is larger than the designed quantity, another pipeline of the same size at the same unit cost is designed, because the capacity of pipeline is limited to <26698 kt yr⁻¹ (Sanchez et al., 2018).

To clarify this assumption, the following sentence was revised in lines 783-786 as: “We assumed that, if the CO₂ emissions exceed 40% of the CO₂ capacity of the county, another pipeline of the same size at the same unit cost is constructed, because the flux of CO₂ transported in pipeline size¹⁶ is limited to <26698 kt yr⁻¹.”.

To consider the impact of this parameter, we tested the sensitivity of BECCS marginal cost to the assumed “40% of CO₂ emitted from power plants” in **Fig. S3c**, reading as:

Figure S3. Sensitive tests for the marginal cost of BECCS. ... (c) Relative to the central scenario, we consider that the pipelines of CO₂ are alternatively used to transport 20% (red line), rather than 40% (blue line), of CO₂ emitted from all power plants. As a result, it reduces CO₂ flow rate in the pipelines, increases unit cost of CO₂ transport (see equation 8 in the main text for unit cost of CO₂ transport as a function of CO₂ flow rate), and produces a higher marginal cost of emission reduction by BECCS in our study. It means that the marginal cost of BECCS can be reduced (or increased) under a more (or less) ambitious target of CCS over the whole country.

Reference:

Sanchez, D. L., Johnson, N., McCoy, S. T., Turner, P. A. & Mach, K. J. Near-term deployment of carbon capture and sequestration from biorefineries in the United States. *Proc. Natl. Acad. Sci. U.S.A.* **115**, 4875-4880 (2018).

REVIEWER COMMENTS

Reviewer #3 (Remarks to the Author):

The resubmission takes into account many of the issues that were highlighted in the previous review. A few of these were rigorously addressed. The review of and adjustment for sustainably removable agricultural residues is good as well as fixing the missing emissions from agricultural residue harvest. Revisions to the text to improve the communication of results and the contribution to China's CO₂ emissions reduction were good to see. However, the attempts to address some of the more significant issues were not adequate. These are discussed in more detail below.

The contribution of this paper is accounting for the spatial aspects of the system. By not revising the analysis to appropriately capture the spatial aspect of the system the contribution is missing. The authors did not adequately respond to the issue of the extreme case in the model ($x=3$) where all biomass is assumed to be transported halfway across the country to reach a coal power plant. This case would be one first estimate without any spatial information about the location of the coal plants, biomass and sequestration sites. It is a worse estimate than using an average transport distance.

There are better ways to reduce the size of the model when the goal is to capture the spatial distribution of the resources, coal plants and sequestration sites. One is to split the biomass and sequestration logistics into separate models with the bioenergy potential feeding the sequestration model. Biomass is expensive to transport making the optimal coal plant to accept the biomass insensitive to the downstream sequestration costs. This allows decomposition of the model. Another is to eliminate variables that are illogical in an optimal solution such as moving biomass beyond the nearest ten counties with a coal plant. This greatly reduces the size of the problem and makes it possible to solve. The Johnson et. al. (2014) paper may have resulted in lower abatement but the generalized model size is similar to that presented here. The difference is through using the methods above and combining similar feedstock types (in terms of co-firing properties and emissions) the computational effort is reduced while maintaining the spatial aspects of the system. For these spatial simulations of potential energy systems it is more important to get a logical system than the absolute optimal. The uncertainty in the parameters mean that the optimal solution is not all that meaningfully different from a near-optimal result.

The authors argue that capturing the uncertainty through Monte Carlo analysis is more important than capturing these spatial aspects. That is a reasonable response. However, the use of a spatial optimization model that does not capture the spatial aspects is more misleading than a simple Monte Carlo analysis that is informed by spatial information (distribution distances between coal plants and biomass resources or sequestration sites).

Finally, the author's response to the issue of linearizing non-linear things is not adequate. For example, "We do not linearize the cost of 90% co-firing when the targeted ratio of biomass in fuel is 50%. We consider that $50/90=55.6\%$ of power plants in a county are retrofitted for 90% co-firing, while the remaining power plants are still burning coal." This is still linearizing the cost. Are there 1000 coal plants in a single county so that only 556 can be converted to 90% co-firing? If not, then at least one plant is not operating at 90% co-firing. This is not likely to cause a major change in the outcome but post-processing of the results would give a more accurate cost estimate. The issue is that this false argument calls into question the author's ability to make the tradeoffs in balancing the need to simplify the system to create a computationally feasible model while still capturing the important aspects of the system.

Comment C1

Overall impressions The resubmission takes into account many of the issues that were highlighted in the previous review. A few of these were rigorously addressed. The review of and adjustment for sustainably removable agricultural residues is good as well as fixing the missing emissions from agricultural residue harvest. Revisions to the text to improve the communication of results and the contribution to China's CO₂ emissions reduction were good to see. However, the attempts to address some of the more significant issues were not adequate. These are discussed in more detail below.

Response

Thank you very much for the positive comments on our revision. We have carefully considered the remaining comments. We hope the revised model with the new results presented below can answer the remaining two technical issues.

Comment C2

The contribution of this paper is accounting for the spatial aspects of the system. By not revising the analysis to appropriately capture the spatial aspect of the system the contribution is missing. The authors did not adequately respond to the issue of the extreme case in the model ($x=3$) where all biomass is assumed to be transported halfway across the country to reach a coal power plant. This case would be one first estimate without any spatial information about the location of the coal plants, biomass and sequestration sites. It is a worse estimate than using an average transport distance.

There are better ways to reduce the size of the model when the goal is to capture the spatial distribution of the resources, coal plants and sequestration sites. One is to split the biomass and sequestration logistics into separate models with the bioenergy potential feeding the sequestration model. Biomass is expensive to transport making the optimal coal plant to accept the biomass insensitive to the downstream sequestration costs. This allows decomposition of the model. Another is to eliminate variables that are illogical in an optimal solution such as moving biomass beyond the nearest ten counties with a coal plant. This greatly reduces the size of the problem and makes it possible to solve. The Johnson et. al. (2014) paper may have resulted in lower abatement but the generalized model size is similar to that presented here. The difference is through using the methods above and combining similar feedstock types (in terms of co-firing properties and emissions) the computational effort is reduced while maintaining the spatial aspects of the system. For these spatial simulations of potential energy systems it is more important to get a logical system than the absolute optimal. The uncertainty in the parameters

mean that the optimal solution is not all that meaningfully different from a near-optimal result. The authors argue that capturing the uncertainty through Monte Carlo analysis is more important than capturing these spatial aspects. That is a reasonable response. However, the use of a spatial optimization model that does not capture the spatial aspects is more misleading than a simple Monte Carlo analysis that is informed by spatial information (distribution distances between coal plants and biomass resources or sequestration sites).

Response

Thank you very much for this insightful comment. We realized that uncertainties from our Monte Carlo analysis cover the ranges, but do not well represent the spatial aspects of biomass transportation. We now split the biomass and sequestration logistics into separate models and explicitly considered a network for biomass transportation between each county and the nearest ten counties over China by following the reviewer's suggestion. Please see a new paragraph added to the **Methods** sections in line 651-671 as: "*Johnson et al. considered the routes of biomass transportation in a model by optimizing the amount of biomass transported between two counties to achieve the lowest cost for bioenergy in power plants in the USA⁸⁴. The computational loads are too heavy to consider the transportation between any two of 2836 counties in China. Because the unit cost of the transportation of biomass is higher than CO₂, we used a similar method as Johnson et al.⁸⁴ to consider the transportation of biomass from the nearest ten counties to power plants in each county. For each route of the biomass transportation, we take the center of non-urban pixels in a county as the starting point of the route, and take the center of power plants in a county as the destination of the route. The distance of transportation and the unit costs of biomass transportation are calculated for each county correspondingly (Table S10). We take the amount of biomass transported from each county to the nearest ten counties as variables for optimization in our model to minimize the total economic cost under a target of CO₂ emission reduction. For the retrofitted power plants in a county, the burnt biomass can be harvested from the same county without long-distance transportation or transported from the nearest ten counties. By varying the source of biomass (x) from 1 to 11 in this scenario, the optimization function (similar to equation 1) is expressed as:*

$$\min_{T_{ixyh}} C = \sum_{i=1}^{2836} \sum_{h=1}^{21} \sum_{x=1}^{11} \sum_{y=1}^3 \mu_{ixyh} T_{ixyh} - \frac{\kappa E}{v\lambda_c}, \quad \forall \frac{\phi E}{v\lambda_c} - N = F \quad (8)$$

where T_{ixyh} is the biomass burnt in county i in route x (1 for city i , and 2 to 11 for the nearest ten cities). For each county, we identify that the biomass is transported from county i to county i_j in a route x , and then the constraint on biomass availability in county i is expressed as:

$$\sum_{y=1}^3 T_{ilyh} + \sum_{y=1}^3 \sum_{j=1}^n T_{jxyh} \leq B_{ih}, \quad i=1 \text{ to } 2836 \text{ and } h=1 \text{ to } 21 \quad (9)$$

where n is the number of counties, to which the biomass harvested in county i has been transported.”

The marginal cost curve of BECCS in this scenario (“*B90-2015-PC-routes*”) is now shown in a new Fig. 3a. Because this scenario explicitly considers the transportation of biomass between each county and only the nearest ten counties, the potential of CO₂ emission reduction by BECCS (up to 3.2 Gt CO₂-eq yr⁻¹) is lower than a scenario of “*B90-2015-PC*” (up to 5.3 Gt CO₂-eq yr⁻¹) that considers the transportation of all biomass in China using a simple method (“*B90-2015-PC*”). We also show the routes of biomass transportation between any two counties to abate 2 Gt CO₂-eq yr⁻¹ in China (Fig. 3b). The new Fig. 3 reads as:

Figure 3 | Marginal cost curves of BECCS in China. (a) Marginal cost curves of BECCS in scenarios include: “*B90-2015-PC*” for retrofitting pulverized-coal (PC) plants under 90% biomass co-firing (B90) to generate electricity in 2015, *B30-2015-PC* for 30% biomass co-firing (B30), “*B90-2015-IGCC*” for transferring PC to integrated gasification combined cycle (IGCC) plants, “*B30-2015-PC-EneCrop*” for using dedicated energy crops (EneCrop) only,

“B90-2030-PC” for generating the projected electricity in 2030, “B90-2015-PC-BestCrop” for using the best-yield crops (BestCrop), “noBiomass-2015-PC” for using coal in power plants equipped with CCS, “B90-2015-PC-noCCS” for biomass co-firing without CCS and “B90-2015-PC-routes” for considering the routes of biomass transportation between each county and the nearest ten counties. CO₂ emission reduction by hydropower, wind, solar and nuclear power in China in 2015 and the marginal costs are shown as red lines. Uncertainties in the marginal costs in the scenario of B90-2015-PC estimated from Monte Carlo simulations are shown as the shaded area. The marginal costs of retrofitting 300, 600, ..., 15000 of 15244 power plants in the scenario of B90-2015-PC or retrofitting 300, 600, ..., 8700 of 8789 power plants in B90-2015-PC-routes are shown by circles. (b) Routes of the biomass transportation between counties to abate 2 Gt CO_{2-eq} yr⁻¹ in B90-2015-PC-routes. (c) Spatial distribution of the retrofitted power plants for BECCS to abate 2 Gt CO_{2-eq} yr⁻¹ in B90-2015-PC.

In addition, we show the amount of the burnt biomass from the same county without transportation or transported from the nearest ten counties in a new **Fig. S3** as:

Figure S3. Sources of biomass burnt in the power plants retrofitted for BECCS in each county. (a, d) Amounts of biomass taken from the same county without transportation. (b, e) Amounts of biomass transported from the nearest ten counties. (c, f) Fraction of the burnt biomass transported from the nearest ten counties to abate 2 or 3 Gt CO_{2-eq} yr⁻¹ by BECCS in China.

Comment C3

Finally, the author’s response to the issue of linearizing non-linear things is not adequate. For example, “We do not linearize the cost of 90% co-firing when the targeted ratio of biomass in

fuel is 50%. We consider that $50/90=55.6\%$ of power plants in a county are retrofitted for 90% co-firing, while the remaining power plants are still burning coal.” This is still linearizing the cost. Are there 1000 coal plants in a single county so that only 556 can be converted to 90% co-firing? If not, then at least one plant is not operating at 90% co-firing. This is not likely to cause a major change in the outcome but post-processing of the results would give a more accurate cost estimate. The issue is that this false argument calls into question the author’s ability to make the tradeoffs in balancing the need to simplify the system to create a computationally feasible model while still capturing the important aspects of the system.

Response

Thank you very much for this insightful comment. We realized that it is very important to improve our spatial-explicit model by considering the capacity of each power plant. Following the reviewer’s suggestion, we now considered that only an integral number of power plants in each county can be retrofitted for biomass co-firing under a constant ratio to coal (e.g., 90% in the scenario of “B90-2015-PC”). Please see a new paragraph added to the **Methods** sections in line 565-572 as: “*We considered that an integral number of power plants can be retrofitted for biomass co-firing to achieve a target of CO₂ emission reduction, based on the capacity of 15462 power plants in 2836 counties over China. First, we calculated the electricity generated by the power plants retrofitted for biomass co-firing from the optimized consumption of biomass in each county (T_{ixyh}). Second, we ranked the power plants in this county in order of the capacity and found the last power plant that is retrofitted to ensure that the electricity generated by retrofitted plants is closest to the target in this county. The marginal cost of BECCS and CO₂ emission reduction are re-calculated when an integral number of power plants are retrofitted in all counties.*”.

Using this improved model, for example, we now show the marginal costs of BECCS by retrofitting 300, 600, ..., 15000 of 15244 power plants in the scenario of *B90-2015-PC* or retrofitting 300, 600, ..., 8700 of 8789 power plants in the scenario of *B90-2015-PC-routes* in Fig. 3a (shown by the circles). We also show the spatial distribution of the retrofitted power plants to abate 2 Gt CO_{2-eq} yr⁻¹ over China (Figs. 3c). The new Fig. 3 reads as:

Figure 3 | Marginal cost curves of BECCS in China. (a) Marginal cost curves of BECCS in scenarios include: “B90-2015-PC” for retrofitting pulverized-coal (PC) plants under 90% biomass co-firing (B90) to generate electricity in 2015, B30-2015-PC for 30% biomass co-firing (B30), “B90-2015-IGCC” for transferring PC to integrated gasification combined cycle (IGCC) plants, “B30-2015-PC-EneCrop” for using dedicated energy crops (EneCrop) only, “B90-2030-PC” for generating the projected electricity in 2030, “B90-2015-PC-BestCrop” for using the best-yield crops (BestCrop), “noBiomass-2015-PC” for using coal in power plants equipped with CCS, “B90-2015-PC-noCCS” for biomass co-firing without CCS and “B90-2015-PC-routes” for considering the routes of biomass transportation between each county and the nearest ten counties. CO₂ emission reduction by hydropower, wind, solar and nuclear power in China in 2015 and the marginal costs are shown as red lines. Uncertainties in the marginal costs in the scenario of B90-2015-PC estimated from Monte Carlo simulations are shown as the shaded area. The marginal costs of retrofitting 300, 600, ..., 15000 of 15244 power plants in the scenario of B90-2015-PC or retrofitting 300, 600, ..., 8700 of 8789 power plants in B90-2015-PC-routes are shown by circles. (b) Routes of the biomass transportation between counties to abate 2 Gt CO_{2-eq} yr⁻¹ in B90-2015-PC-routes. (c) Spatial distribution of the retrofitted power plants for BECCS to abate 2 Gt CO_{2-eq} yr⁻¹ in B90-2015-PC.

Using the improved model, we update all results and figures in our paper.

REVIEWER COMMENTS

Reviewer #3 (Remarks to the Author):

The authors have responded to my criticism but created a new problem in doing so. Reducing model size needs to be done with great care to ensure that potential solutions are not removed that could appear in an optimal solution. That has not happened in this case. The total potential abatement should not drop due to changes made to reduce the model size. It appears that the authors did not make sure that all biomass supplies had available coal plants to receive it or there is some other issue that I am not able to troubleshoot. If the authors can fix this problem without creating new ones, I recommend publication.

Comment C1

The authors have responded to my criticism but created a new problem in doing so. Reducing model size needs to be done with great care to ensure that potential solutions are not removed that could appear in an optimal solution. That has not happened in this case. The total potential abatement should not drop due to changes made to reduce the model size. It appears that the authors did not make sure that all biomass supplies had available coal plants to receive it or there is some other issue that I am not able to troubleshoot. If the authors can fix this problem without creating new ones, I recommend publication.

Response

Thank you very much for the positive comments on our revision. We revised our model to consider the last comment from Reviewer 3. In the previous version, the total capacity of BECCS was lower after considering the biomass transported from the nearest ten counties to a county, because the supply of biomass from other counties was not considered. According to this suggestion, for power plants in each county, we take the amount of the burnt biomass transported from **the nearest ten counties, other counties in this province, or counties in other provinces as the variables** for optimization in our model. As a result, the potential of emission reduction by BECCS ($5.24 \text{ Gt CO}_2\text{-eq yr}^{-1}$) is now close to the result in the central scenario of “B90-2015-PC” ($5.25 \text{ Gt CO}_2\text{-eq yr}^{-1}$). Based on this revision, all results are updated. The new Fig. 3 reads as:

Figure 3 | Marginal cost curves of BECCS in China. (a) Marginal cost curves of BECCS in scenarios include: “B90-2015-PC” for retrofitting pulverized-coal (PC) plants under 90% biomass co-firing (B90) to generate electricity in 2015, B30-2015-PC for 30% biomass co-firing (B30), “B90-2015-IGCC” for transferring PC to integrated gasification combined cycle

(IGCC) plants, “B30-2015-PC-EneCrop” for using dedicated energy crops (EneCrop) only, “B90-2030-PC” for generating the projected electricity in 2030, “B90-2015-PC-BestCrop” for using the best-yield crops (BestCrop), “noBiomass-2015-PC” for using coal in power plants equipped with CCS, “B90-2015-PC-noCCS” for biomass co-firing without CCS and “B90-2015-PC-routes” for considering the routes of biomass transportation between each county and the nearest ten counties. CO₂ emission reduction by hydropower, wind, solar and nuclear power in China in 2015 and the marginal costs are shown as red lines. Uncertainties in the marginal costs in the scenario of B90-2015-PC estimated from Monte Carlo simulations are shown as the shaded area. The marginal costs of retrofitting 300, 600, ..., 15000 of 15244 power plants in the scenario of B90-2015-PC or retrofitting 300, 600, ..., 15000 of 15159 power plants in B90-2015-PC-routes are shown by circles. (b) Routes of the biomass transportation between counties to abate 2 Gt CO_{2-eq} yr⁻¹ in B90-2015-PC-routes. (c) Spatial distribution of the retrofitted power plants for BECCS to abate 2 Gt CO_{2-eq} yr⁻¹ in B90-2015-PC.

The paragraph in **Methods** sections in line 656-669 was revised as:

“For power plants in each county, we take the amount of the burnt biomass transported from the nearest ten counties, other counties in this province, or counties in other provinces as the variables for optimization in our model to minimize the total economic cost under a target of CO₂ emission reduction. For the retrofitted power plants in a county, the burnt biomass can be harvested from the same county without long-distance transportation or transported from the nearest counties or other counties at a longer distance. By varying the source of biomass (x) from 1 to 12 in this scenario, the optimization function (similar to equation 1) is expressed as:

$$\min_{T_{ixyh}} C = \sum_{i=1}^{2836} \sum_{h=1}^{21} \sum_{x=1}^{12} \sum_{y=1}^3 \mu_{ixyh} T_{ixyh} - \frac{\kappa E}{\nu \lambda_c}, \quad \nabla \frac{\phi E}{\nu \lambda_c} - N = F \quad (8)$$

where T_{ixyh} is the biomass burnt in county i from a specific source x (1 to 10 for the nearest ten cities including county i , 11 for other counties in this province and 12 for counties in other provinces). While we determine the distance of biomass transportation between any two counties for $x=1-10$, we use the average distance between county i and other counties for $x=11, 12$ to estimate the unit cost and emission of biomass transportation.”.

REVIEWERS' COMMENTS

Reviewer #3 (Remarks to the Author):

The authors have addressed my concerns. This is an interesting study that provides greater insight into an important issue if the global community hopes to achieve climate change goals.